statistics/applied mathematics

scientific reform, reproducibility, replication, double-dipping, exploratory research

**Author for correspondence:**
Berna Devezer
e-mail: bdevezer@uidaho.edu

# The case for formal methodology in scientific reform

Berna Devezer[1], Danielle J. Navarro[3],

Joachim Vandekerckhove[4] and Erkan Ozge Buzbas[2]

[1]Department of Business, and [2]Department of Mathematics and Statistical Science, University of Idaho, Moscow, Idaho, USA
[3]School of Psychology, University of New South Wales, Sydney, Australia
[4]Department of Cognitive Sciences and Department of Statistics, University of California, Irvine, USA

BD, 0000-0002-5979-2781

Current attempts at methodological reform in sciences come in response to an overall lack of rigor in methodological and scientific practices in experimental sciences. However, most methodological reform attempts suffer from similar mistakes and over-generalizations to the ones they aim to address. We argue that this can be attributed in part to lack of formalism and first principles. Considering the costs of allowing false claims to become canonized, we argue for formal statistical rigor and scientific nuance in methodological reform. To attain this rigor and nuance, we propose a five-step formal approach for solving methodological problems. To illustrate the use and benefits of such formalism, we present a formal statistical analysis of three popular claims in the metascientific literature: (i) that reproducibility is the cornerstone of science; (ii) that data must not be used twice in any analysis; and (iii) that exploratory projects imply poor statistical practice. We show how our formal approach can inform and shape debates about such methodological claims.

## 1. Introduction

Widespread concerns about unsound research practices, lack of transparency in science and low reproducibility of empirical claims have led to calls for methodological reform across scientific disciplines [1–4]. The literature on this topic has been termed 'meta-research' [5] or 'metascience' [6] and has had policy impact on science agencies, institutions, and practitioners [7]. Perhaps surprisingly, proper evaluation of methodological claims in meta-research—understood as statements about scientific methodology that are either based on statistical arguments or affect statistical practice—has received little formal scrutiny itself. Policies are

proposed with little evidentiary backing and based on methods which are suggested with no framework for assessing their validity or evaluating their efficacy (e.g. see policy and methods proposals in [8–13]).

For example, reform methodologists have criticized empirical scientists for: (i) prematurely presenting unverified research results as facts [14]; (ii) overgeneralizing results to populations beyond the studied population [15]; (iii) misusing or abusing statistics [16,17]; and (iv) lacking rigor in the research endeavour that is exacerbated by incentives to publish fast, early, and often [11,12]. However, the methodological reform literature seems to us to be afflicted with similar issues: we see premature claims that untested methodological innovations will solve replicability/reproducibility problems; conditionally true statements about methodological tools presented as unconditional, bold claims about scientific practice; vague or misleading statistical statements touted as evidence for the validity of reforms; and we are concerned about an overall lack of rigor in method development that is exacerbated by incentives to find immediate solutions to the replication crisis (see also [7], for an overall critique of the dominant epistemology of metascience). This is a reason for concern. We expect methodological reforms to be held to standards that are *at least* as rigorous as those we expect of empirical scientists. Should we fail to do so, we run the risk of repeating the mistakes of the past and creating new scientific processes that are no better than those they replace. There is an uncomfortable symmetry to this, but also an opportunity: reformers are in an opportune position to take criticism and self-correct before allowing false claims to be canonized as methodological facts [18].

In this paper, we advocate for the necessity of statistically rigorous and nuanced arguments to make proper methodological claims in the reform literature. Because methodological claims are either based on statistical arguments or affect statistical practice, they need to be statistically correct. Statistics is a *formal* science whose methods follow from probability calculus to be valid, and this validity is established either by mathematical proofs or by simulation proofs before being advanced for the use of scientists. Formalization allows us to subject our verbal intuitions to scrutiny, revealing holes, inconsistencies, and undeclared assumptions and to make precise, transparent claims that hold under well-specified assumptions [19]. As such, by statistical rigor, we mean doing and showing the necessary formal work to establish *how* we know a methodological claim is valid, in a way that does not leave room for idle speculation. Scientific and statistical nuance, on the other hand, is about clearly specifying *when* (i.e. under what assumptions and conditions) a claim should apply, which should result in measured and contextual statements while preventing over-generalizations.

The emphasis and novelty of our current work is in demonstrating by example how formal rigor can be achieved when proposing methods in metascience: by motivating them from first principles, and using fundamental mathematical statistics machinery to provide their proofs. Herfeld & Ivanova [20, p. 1] talk about first principles in science as fundamental building blocks and define them as follows: 'Depending on the case, they can be formal axioms, theoretical postulations, basic propositions, or general principles that have a special status and role to play in the theory in which they are embedded.' Methodological reform and metascience currently lack a theoretical foundation [7], are ambiguous about their first principles, and may benefit from formalism in establishing these building blocks. A formal approach to solving methodological problems can be summarized as follows.

**Formal approach to solving methodological problems.**
0. **Conception.** An informal problem statement and a proposed solution to that problem, often expressed non-technically.
1. **Definitions.** Identification of variables, population parameters, and constants involved in the problem, and statistical model building using these quantities, with explicitly stated model assumptions.
2. **Formal problem statement.** Mathematical propositions or algorithms positing methodological claims.
3. **Formal result.** Mathematical or simulation-based proofs that interrogate the validity of the statements in step 2.
4a. **Demonstrations.** If the statements are valid, examples showing their relevance in application.
4b. **Extensions and limitations.** Assessing methodological claims' computational feasibility, robustness, and theoretical boundaries in domain-specific applications.
5. **Policy making.** Recommendations on how methods newly established through steps 2-4 can be useful in practice.

Regardless of which claims they support or oppose to, most popular methodological proposals in the reform literature start with step 0, and jump to step 5 without formal results or much evidence of work on intermediate steps. This is in stark contrast with proper formal approach in statistical method development. Practical value of a method established by steps 4b and 5 may require domain-specific

knowledge and might not be tackled well until after a method is introduced. However, in proper method development, these steps are undertaken only if steps 1-4a can actually provide justifications for or against a methodological proposal at the onset.

To show why formalism is essential in establishing the validity of methodological proposals and how informal approaches making the jump from step 0 to 5 might misinform scientific practice, we evaluate three specific examples of methodological claims from the reform literature:

— reproducibility is the cornerstone of, or a demarcation criterion for, science;
— using data more than once invalidates statistical inference; and
— exploratory research uses 'wonky' statistics.

We focus on these claims as case studies to illustrate our approach because all three are methodological claims with statistical implications that have been impactful[1] in the metascience literature as well as on post-replication crisis practices of empirical scientists while also receiving considerable but informal criticism. In an attempt to demonstrate how to formally resolve such disagreements, we evaluate each of these claims using statistical theory, against a broad philosophical and scientific background.

The results from our call for formal statistical rigor and nuance can reach further: a formal statistical approach establishes a framework for broader understanding of a methodological problem by a careful mathematical statement and consideration of model assumptions under which it is valid. Most valid methodological advances are incremental, because they can only be shown formally to be valid under a strong set of assumptions. These advances rarely ever provide simple prescriptions to complex inference problems. Norms issued on the basis of bold claims about new methods might be quickly adopted by empirical scientists as heuristics and might alter scientific practices. However, advancing such reforms in the absence of formal proofs is sacrificing rigor for boldness and can lead to unforeseeable scientific consequences. We believe that hasty revolution may hold science back more than it helps move it forward. We hope that our approach may facilitate scientific progress that stands on firm ground—supported by theory or evidence.

## 2. Claim 1: reproducibility is the cornerstone of, or a demarcation criterion for, science

A common assertion in the methodological reform literature is that reproducibility[2] is a core scientific virtue and should be used as a standard to evaluate the value of research findings [1,4,23,25,27–30]. This assertion is typically presented without explicit justification, but implicitly relies on two assumptions: first, that science aims to discover regularities about nature and, second, that reproducible empirical findings are indicators of true regularities. This view implies that if we cannot reproduce findings, we are failing to discover these regularities and hence, we are not practising science.

The focus on reproducibility of empirical findings has been traced back to the influence of falsificationism and the hypothetico-deductive model of science [31]. Philosophical critiques highlight limitations of this model [32,33]. For example, there can be true results that are by definition not reproducible. Some fields aim to obtain contextually situated results that are subject to multiple interpretations. Examples include clinical case reports and participant observation studies in hermeneutical social sciences and humanities [33]. Other fields perform inference on random populations resulting from path-dependent stochastic processes, where it is often not possible to obtain two statistically independent samples from the population of interest. Examples are inference on parameters in evolutionary systems or event studies in economics. There are also cases where observing or measuring a variable's value changes its probability distribution—a phenomenon akin to the observer effect. True replication may not be possible in these cases. In short, science does—rather often, in fact—make claims about non-reproducible phenomena and deems such claims to be true in

---

[1]As an indication of impact on scientific literature, we looked up Google Scholar citation counts for some of the key articles from which these claims originate, the oldest of which was published 8 years ago. By the time, the current manuscript was last revised, [1] had 686; [12] had 1045; [21] had 473; [22] had 574; [23] had 529; [4] had 4807; [24] had 1182; [13] had 704; and [25] had 244 citations.

[2]Here, we use reproducibility as in: 'the extent to which consistent results are observed when scientific studies are repeated' ([23], p. 657). In appendix A, we provide a technical definition of reproducibility which we use in obtaining our results. We limit our discussion to *statistical reproducibility of results* only (similar to results reproducibility in [26]), and exclude other types such as computational or methods reproducibility—whether the materials, methods, procedures, algorithms, analyses used in an original study are reported in a sufficiently detailed and transparent way that enables others to carry it out again.

spite of the non-reproducibility. In these instances what scientists do is to define and implement appropriate criteria for assessing the rigor and the validity of the results [32], without making a reference to replication or reproduction of an experimental result. Indeed, many scientific fields have developed their own qualitative and quantitative methods such as ethnography or event study methodology to study non-reproducible phenomena.

We argue that even in scientific fields that possess the ability to reproduce their findings in principle, reproducibility cannot be reliably used as a demarcation criterion for science because it is not necessarily a good proxy for the discovery of true regularities. This counterpoint has informally been brought up in metascience literature before [26,34–39]. Our goal is to further advance this argument by providing a formal, quantitative evaluation of statistical reproducibility of results as a demarcation criterion for science. We consider the following two unconditional propositions: (i) reproducible results are true results, and (ii) non-reproducible results are false results. If reproducibility serves as a demarcation criterion for science, we expect these propositions to be true: we should be able to reproduce all true results and fail to reproduce all false results with reasonable regularity. In this section, we provide statistical arguments to probe the unconditional veracity of these propositions and we challenge the role of reproducibility as a key epistemic value in science. We also list some *necessary* statistical conditions for true results to be reproducible and false results to be non-reproducible. We conclude that methodological reform first needs a mature theory of reproducibility to be able to identify whether *sufficient* conditions exist that may justify labelling reproducibility as a measure of true regularities.

## 2.1. Reproducibility rate is a parameter of the population of studies

To examine the suitability of reproducibility as a demarcation criterion, a precise definition of reproducibility of results is necessary. While many definitions have been offered for replication and results reproducibility (see [44], for a partial list), most are informal and not sufficiently precise or general for our purposes[3]. In this paper, we use our own definitions based on first principles to facilitate the derivation of our theoretical results. In assessing the reproducibility of research results, literature refers to 'independent replications' of a given study. Therefore, it is necessary to define the notion of a study mathematically, before referring to replications of that study. We provide a precise mathematical definition of an *idealized study* in appendix A. Briefly, its components involve an assumed probability model generating the data involving the random variable and parameters of interest, a dataset of fixed sample size, the statistical method employed in analysing the data, the background knowledge about the variable of interest, and a decision rule to deliver the result of the analysis. We note that it is not sufficient to lay out the higher-level assumptions to provide formal results. Lower-level assumptions such as mathematical regularity conditions about variables must also be specified as outlined in step 1 of our formal approach. We also note that the definition given in appendix A is sufficiently broad to investigate reproducibility of results for any mode of statistical inference including estimation, model selection, and prediction, and not just hypothesis testing.

Strictly, we cannot speak of statistical independence between an original study and its replications. If study B is a replication of study A, then many aspects of study B depend on study A. Rather, sequential replication studies should be assumed statistically *exchangeable*, conditional on the results and the assumptions of the original study, in the sense that the group of results obtained from a sequence of replication studies are probabilistically equivalent to each other irrespective of the order in which these studies are performed. Assuming that exchangeability holds, probability theory shows that the results from replication studies become independent of each other, but *only* conditional on the background information about the system under investigation, model assumed, methods employed, and the decision process used in obtaining the result. The commonly used phrase 'independent replications' thus has little value in developing a theory of reproducibility unless one takes sufficient care to consider all these conditionals.

This conditional independence of sequence of results immediately implies that irrespective of whether a result is true or false, there is a true reproducibility rate of *any given result*, conditional on

---

[3]Some exceptions are as follows: Patil *et al.* [38] use the overlap in prediction intervals from original and replication studies to define a statistical measure of reproducibility. Gorroochurn *et al.* [45] investigates the relationship between reproducibility and *p*-values and in the context of association between variables. Pauli [46] develops a Bayesian model to evaluate the results of replication studies and estimate a reproducibility rate. Hedges & Schauer [47] offer a principled way of evaluating replication studies within a meta-analytic framework. Different from purely statistical approaches, Fanelli [48] takes a meta-analytic approach to study reproducibility and uses an information theoretical framework to quantify it. We acknowledge and endorse the formal approach undertaken by these articles to address practical problems of evaluating and quantifying the results of replication experiments.

**Box 1.** Some necessary conditions to obtain true results that are reproducible and false results that are non-reproducible.

— True values of the unknown and unobservable quantities for which inference is desired must be in the decision space (appendix B).

Examples: (i) in model selection, selecting the true model depends on having an M-closed model space, which means the true model must be in the candidate set [40]; and (ii) in Bayesian inference, converging on the true parameter value depends on the true parameter value being included in the prior distribution, as stated by Cromwell's rule ([41], p. 90).

— If inference is performed under one assumed model, that model should correctly specify the true mechanism generating the data.

Example: a simple linear regression model with measurement error misspecified as a simple linear regression model yields biased estimates of regression coefficients, which will affect reproducibility of true and false results (figures 1 and 2).

— The quantities that methods use to perform inference on unknown and unobservable components of the model must contain enough information about those components: if they are statistics, they cannot be only ancillary. If they are pivots that are a function of nuisance parameters, then the true value of those nuisance parameters should permit reproducibility of results (appendix B).

Example: in a one sample $z$-test where the population mean is not equal to the hypothesized value under the null hypothesis, the test incorrectly fails to reject with large probability owing to large population variance.

— If inference is about parameters, observables must carry enough discernible information about these parameters. That is, model parameters should be identifiable structurally and informationally. Even weak unidentifiability will reduce the reproducibility of true results.

Example: the requirement that the Fisher information ([42], p. 115) about unknown parameters should be sufficiently large in likelihoodist frameworks.

— Free parameters of methods should be compatible with our research goals.

Example: a hypothesis test in Neyman–Pearson framework with Type I error rate $\alpha \approx 1$ is a valid statistical procedure that rejects the null hypothesis almost always when it is true.

— Methods should be free of unknown bias.

Example: Heisenbug is a special case of observer effect—where mere observation changes the system we study, potentially leading to false results that are reproducible—found in computer programming, that refers to a software bug that alters its behaviour or even disappears during debugging.

— The sample on which inference is performed is representative of the population from which it is drawn.

Example: statistical methods assume probabilistic sampling and do not make any claims in a non-probabilistic sampling framework [43].

the properties of the study. This true reproducibility rate is determined by three components: the true model generating the data, the assumed model under which the inference is performed, and the methods with which the inference is performed. In this sense, the true reproducibility rate is a parameter of the population of studies and we have the following result which satisfies step 2 of our formal approach.

**Proposition 2.1.** *Let $R_o$ be a result and $R^{(i)}$ be the result in ith attempted replication of the idealized study from which $R_o$ is obtained. If $\mathbf{I}_{\{R^{(i)}=R_o|R_o\}} = 1$ we say that $R_o$ is reproduced by $R^{(i)}$. Else, we say that $R_o$ failed to reproduce by $R^{(i)}$. Conditional on $R_o$, the relative frequency of reproduced results $\phi_N \to \phi \in [0, 1]$, as $N \to \infty$. Further, $\phi = 1$ only in highly specific problems. Proof is provided in appendix B, per step 3 of our formal approach.)*

To show the value of the formal approach, we now briefly interpret what proposition 2.1 establishes and contributes to our understanding about reproducibility of results. Just like a statistic (e.g. sample

mean) has a sampling distribution, and it converges to its population counterpart (i.e. the population mean) as the sample size increases, the sample reproducibility rate of a sequence of idealized studies has a sampling distribution, and it will converge to its population counterpart as the number of studies increases. Therefore, the true reproducibility rate for an idealized study *must be* a fixed population quantity and it is independent of our efforts given the idealized study. Furthermore, this rate of reproducibility can take any value between 0 and 1. The actual value depends on the properties of the idealized study but it can be high or low, so that we should not expect it to be high all the time. Finally, we note that this holds for any result, true or false. Stepping back, now we see the advantage of the formal approach as follows. Given the definitions in appendix A, if the proof in appendix B is correct, then our result is a mathematical fact and it *must be* correct. Therefore, a formal statement like proposition 2.1 has taken us one step further to understand the properties of reproducibility of results.

## 2.2. True results are not necessarily reproducible

Much of the reform literature claims non-reproducible results are necessarily false. For example, Wagenmakers *et al.* ([13], p. 633) assert that 'Research findings that do not replicate are worse than fairy tales; with fairy tales the reader is at least aware that the work is fictional.' It is implied that true results must necessarily be reproducible, and therefore non-reproducible results must be 'fictional.' More mildly, Zwaan *et al.* ([25], p. 13) state: 'A finding is arguably not scientifically meaningful until it can be replicated with the same procedures that produced it in the first place.' Others have taken issue with this claim (e.g. [36,38,45]), pointing to reasons why replication attempts may fail to reproduce the original result other than its truth value. We now take our formal approach again and find that an evaluation of the claim provides support for this criticism.

The fact that the true reproducibility rate is a parameter of the population of studies matters: this parameter is a probability and therefore, it takes values on the interval [0, 1]. This implies that for finite sample studies involving uncertainty, the true reproducibility rate must necessarily be smaller than one for any result and in fact, we have the following result (step 2 of our formal approach).

**Proposition 2.2.** There exists true results $R_o = R_T$, whose true reproducibility rate $\phi_T$ is arbitrarily close to 0. (Per step 3 of our formal approach, proof is provided in appendix B.)

Before looking into some examples for proposition 2.2, we discuss it to make an important point about the formal approach: proposition 2.2 may seem perplexing because intuitively, we might expect that if a result is true, we should be able to reproduce it. If this is in fact our (wrong) intuition, we should revisit and re-hone it studying the proof of proposition 2.2. The reason is that, in a formal approach as long as the proof is correct, the result must be correct, and therefore our intuition must be wrong. Most importantly, all this evaluation is made possible by motivating the issue of reproducibility from the first principles and proceeding formally from that point into a next by stating and proving the results that help us to build knowledge on the subject. We already argued that first principles on evaluating the reproducibility of results required a definition of idealized study, together with all its assumptions and mathematical regularity conditions (appendix A). Given these, we were able to show that reproducibility rate is a parameter of the population of studies (proposition 2.1). Given this, we showed that the relationship between true results and their reproducibility rate might be complex (proposition 2.2). Therefore, moving in this formal way builds a solid body of knowledge, mathematically supported under well-defined and delineated models.

As an example of step 4a of the formal approach to solving methodological problems, we discuss two statistical scenarios to illustrate the counterintuitive result provided by proposition 2.2. A well-known example is a data generating model where the sampling error (the uncertainty) is large with respect to the model expectation (the signal). This is rather an informal statement of the kind we make in step 0 (i.e. no statistical model is specified) of the formal approach to solving methodological problems. If we want to check whether the statement is true, it *should and can* be precisely formulated mathematically starting from our definition of idealized study.

In contrast to statements ubiquitous in metascience literature (e.g. [25,37]), large sampling error is not the only reason why true results might not be reproducible. Falling back to the definition of an idealized study given in appendix B, we see that its components are the assumed model and its parameters, data, method, background knowledge of the system and the decision function to obtain a result. Because the reproducibility rate is a parameter of population of studies, components of an idealized study other than large sampling error can affect the reproducibility of a true result. For example, the model might be

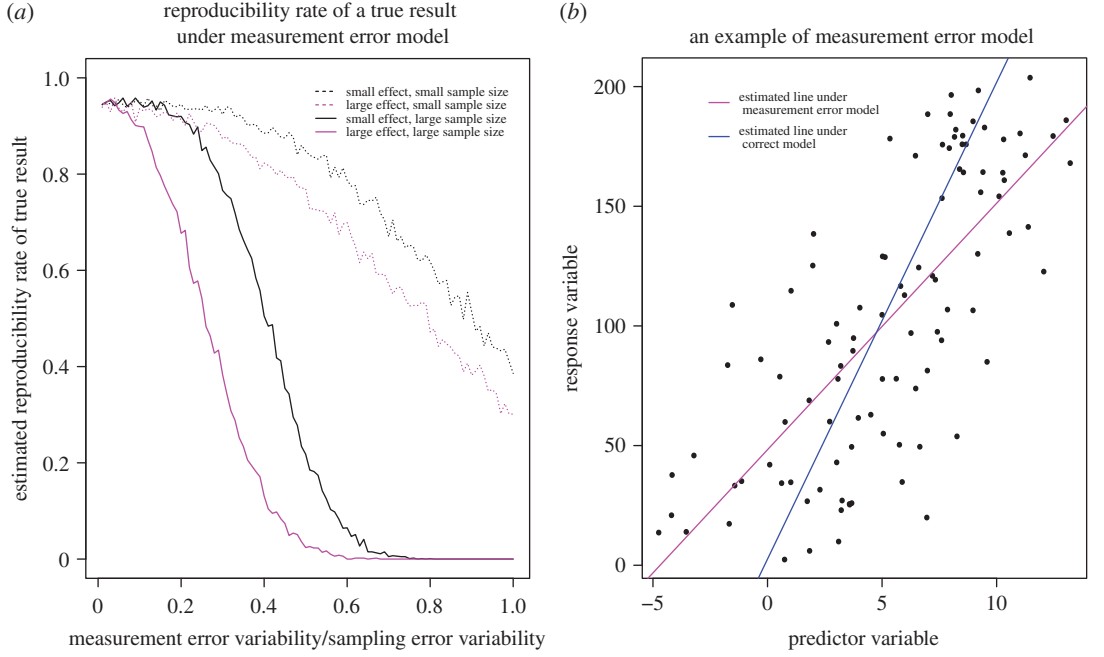

**Figure 1.** (*a*) Reproducibility rate of a true result decreases with measurement error in a misspecified simple linear regression model. Reproducibility rate is estimated by the proportion of times the 95% confidence interval captures the true effect. Sample sizes are 50 (small) and 500 (large). The true regression coefficient of the predictor variable is 2 (small effect) and 20 (large effect). Model details are given in appendix D. (*b*) Example data (black points) generated under simple linear regression model $E(Y) = 2 + 20X$. Measurement and sampling error are normally distributed with standard deviations equal 3. Regression lines are fit under measurement error model (magenta line) and the correct model (blue line) with a sample size of 100. The 95% confidence interval for the regression coefficient obtained under the measurement error model is (7.94, 12.37), which does not include the true value 20. By contrast, the 95% confidence interval for the regression coefficient obtained under the correct model, (19.86, 20.21), includes the true value. For the code generating all simulations and figures in the article, please see the electronic supplementary material.

misspecified, or the model might be correctly specified and sampling error small, but the method might have large error, or the decision function might not be optimal. Again, having defined an idealized study formally helps us to investigate and prove any one of these cases if we wish so.

We can also evaluate the opposite of the large sampling error case: small sampling error is not a guarantee that true results will be highly reproducible. It turns out that there are mathematically necessary conditions other than the truth value of a result, that need to be met for true results to be highly reproducible. Some of these conditions which are related to the components of idealized study are listed in box 1 informally. Thus, the formal approach has also the advantage of motivating and evaluating other cases such as complements, reverses, or counters, and therefore it enriches our understanding of reproducibility of results.

Another well-known statistical scenario illustrating proposition 2.2 is when the data are analysed under a misspecified model (per step 4a). Here, we take a simple linear regression *measurement error model* in which the measurement error is unaccounted for (figure 1). We are interested in the effect of measurement error on the reproducibility rate of a true effect. As the ratio of the measurement error variability in predictor to sampling error variability increases, the probability that an interval estimator of the regression coefficient (i.e. the effect size) at a fixed nominal coverage contains the true effect decreases. This is not simply an artefact of small sample sizes or small effects: the same pattern is obtained for large sample sizes and large true effects. In fact, for large sample sizes, the reproducibility rate drops to zero at *lower* measurement error variability than for small sample sizes (also see [49], for a similarly counter-intuitive effect of measurement error). Furthermore, the negative effect of measurement error on reproducibility rate of a true result actually grows with effect size, as figure 1*a* illustrates. Even in this relatively simple setting it is by no means a given that a true result will be reproducible. Measurement error is only one type of model misspecification. Other sources of misspecification and types of human error (e.g. questionable research practices) might further impair the reproducibility of true results.

When true reproducibility rate of a true result is low, the proportion of studies that fail to reproduce a true result will be high, even when methods being used have excellent statistical properties and the

model is correctly specified. However, a true low reproducibility rate does not necessarily indicate a problem in the scientific process. As Heesen [50] notes, low reproducibility in a given field or literature may be the result of there being few discoveries to be made in a given scientific system. When that is the case, a reasonable path to making scientific progress is to learn from non-reproducible results. Indeed, the history of science is full of examples of fields going through an arduous sequence of experiments yielding failures such as non-reproducible results to eventually arrive at scientific regularities [51–53].

In an article that makes practical recommendations to improve the methodology of psychological science, Lakens & Evers [54, p. 278] argue that 'One of the goals of psychological science is to differentiate among all possible truths' and suggest that one way to achieve this goal is to improve the statistical tools employed by scientists. Some care is needed when interpreting this claim. Statistical methods might indeed help us get close to the true data generating mechanism, if their modelling assumptions are met (thereby removing some of the reasons why true results can be non-reproducible). However, statistics' ability to quantify uncertainty and inform decision making does not guarantee that we will be able to correctly specify our scientific model. Irrespective of reproducibility rates of results obtained with statistical methods, scientists attempting to model truth use theories developed based on their domain knowledge. Some of the problems raised in box 1, including model misspecification and decision spaces that exclude the true value of the unknown components, can only be addressed using a theoretical understanding of the phenomenon of interest. Without this understanding, there is no theoretical reason to believe that reproducibility rates will inform us about our proximity to truth.

It would be beneficial for reform narratives to steer clear of overly generalized sloganeering regarding reproducibility as a proxy for truth (e.g. reproducibility is a demarcation criterion or non-reproducible results are fairy tales). A nuanced view of reproducibility might help us understand why and when it is or is not desirable, and what its limitations are as a performance criterion.

## 2.3. False results might be reproducible

Contrary to proposition 2.2, the next proposition considers false results and the respects in which these can sometimes be highly reproducible (per step 2 of our formal approach).

**Proposition 2.3.** *There exists false results $R_o = R_F$, whose true reproducibility rate $\phi_F$ is arbitrarily close to 1. (Per step 3 of our formal approach, proof is provided in appendix B.)*

In well-cited articles in methodological reform literature, high reproducibility of a result is often interpreted as evidence that the result is true [4,12,24,25]. A milder version of this claim is also invoked, such as 'Replication is a means of increasing the confidence in the truth value of a claim.' ([12], p. 617). The rationale is that if a result is independently reproduced many times, it must be a true result.[4] This claim is not always true [36,57]. To formally establish this, it is sufficient to note that the true reproducibility rate of any result depends on the true model *and* the methods used to investigate the claim. We follow with two examples (step 4a).

First, consider a valid hypothesis test in which the researcher unreasonably chooses to set $\alpha = 1$. Then, a true null hypothesis will be rejected with probability 1 and this decision will be 100% reproducible, assuming that replication studies also set the significance criterion ($\alpha$) to 1. While we know better than to set our significance criterion so high, this example shows how reproducibility rate is not only a function of the truth but also our methods. Second, consider estimators that exploit the bias-variance trade-off by introducing a bias in the estimator to reduce its variance. These estimators have a higher reproducibility rate but for a false result by design. In this case, researchers deliberately choose false results that are reproducible when they prefer a biased estimator over a noisy one for usefulness. Next, we give a realistic example, in which we describe a *mechanism* for why reproducibility cannot serve as a demarcation criterion for truth.

We consider model misspecification under a measurement error model in simple linear regression. Simple linear regression involves one predictor and one response variable, where the predictor variable values are assumed to be fixed and known. The measurement error model incorporates unobservable random error on predictor values. The blue belt in figure 2 shows that as measurement

---

[4]An epistemic claim that well-confirmed scientific theories and models capture (approximate) truths about the world is an example of *scientific realism*. The arguments for and against scientific realism (e.g. positivism) are beyond the scope of this paper. Interested readers may follow up on discussions in the philosophical literature (e.g. [55,56]).

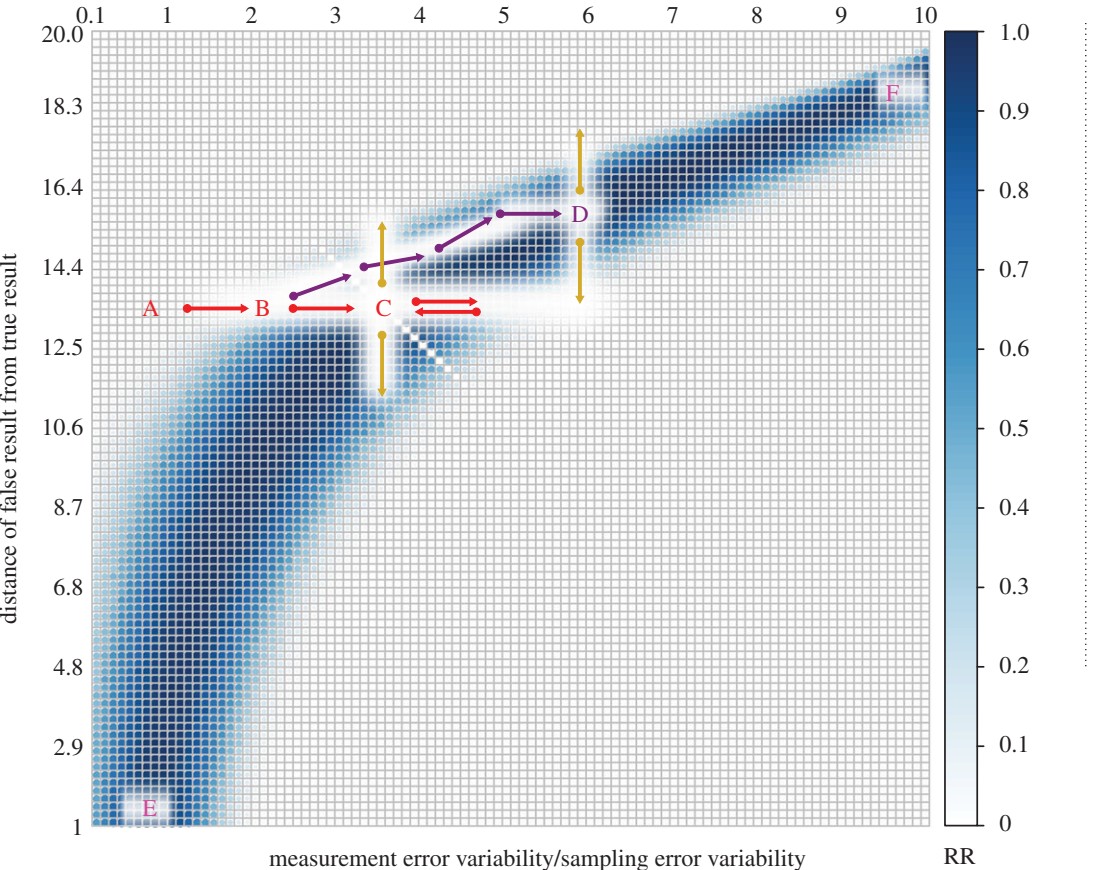

**Figure 2.** An example of almost perfectly reproducible false results in a misspecified simple linear regression model with measurement error. Colour map shows reproducibility rate (RR). Darkest blue cells indicate perfect reproducibility rate (almost 100%) of false results at appropriate measurement error for each false effect size, shown by its distance from the true effect size on the vertical axis. The true regression coefficient of predictor variable (effect size) is 20. Details are given in appendix D. For description of letters and arrows, refer to the text.

error variability grows with respect to sampling error variability, effects farther away from the true effect size become perfectly reproducible. At point F in figure 2, the measurement error variability is ten times as large as the sampling error variability, and we have perfect reproducibility of a null effect when the true underlying effect size is in fact large.

Now consider a scientist who takes reproducibility rate as a demarcation criterion. Assume she starts at point A and she performs a study which lands her at point B—which might happen by knowingly or unknowingly choosing noisier measures or by reducing sampling error variability. The reproducibility of her results has increased (from white to inside the blue belt) and to increase it further, she performs another study by further tweaking the design, which then lands her at point C. If she were to move horizontally to the right with her future studies, the reproducibility of results will decrease, and she will turn back to C, which ultimately will be a stable equilibrium of maximal reproducibility. Further, this is just one of the possible paths that she could take to achieve maximal reproducibility. When at point B, she might perform a study that follows the purple path, always increasing the reproducibility of her results ending up at point D, which is another stable equilibrium point of maximal reproducibility. In fact, any sequence of studies that increases reproducibility will end at one of the points that corresponds to the darkest blue colour in the belt. At this point, however, we note that going from point A to point C, our researcher started with a false result where the estimated slope was some ≈13 units off the true value (y-axis, point A) and arrived at the same false result (y-axis, point C), even though she has maximized the reproducibility of her results. Worse, when she arrived at point D, the estimated slope is now some ≈15 (y-axis, point D) units away from the true value, even though she still maximized the reproducibility of her results.

Taking a step back, we note that to approach the true result, one needs to move to the origin in this plot. However, that approach is controlled by the vertical axis, and not the horizontal. Unless we know

that we are committing a model misspecification error, we get no feedback when we perform studies that move us randomly on the vertical axis (yellow arrows). For example, points C and D have similar reproducibility of results but at C we are closer to truth then D. In fact, consider points E and F: we get high reproducibility of results at both points, but estimates obtained at point E are much closer to the true value than estimates obtained at point F. The mechanistic explanation of this process is that reproducibility-as-a-criterion can be optimized by the researcher *independently of the underlying truth of their hypothesis*. That is, optimizing reproducibility can be achieved without getting any closer to the true result. This is not to say that reproducibility is not useful, but it means that it cannot be used as a demarcation criterion for science.

While we advance a statistical argument for the reproducibility of false results, the truth value of reproducible results from laboratory experiments has also been challenged for non-statistical reasons [58, p. 30]. Hacking notes that mature laboratory sciences sometimes construct an irrefutable system by developing theories and methods that are 'mutually adjusted to each other'. As a result, these sciences become what Hacking calls 'self-vindicating'. That is:

> 'The theories of the laboratory sciences are not directly compared to 'the world'; they persist because they are true to phenomena produced or even created by apparatus in the laboratory and are measured by instruments we have engineered.' [58, p. 30]

Hacking concludes that '[h]igh level theories are not 'true' at all.' They can be viewed as a summary of the collection of laboratory operations to which they are adapted, but if that set of operations is selected to match a particular theory, its evidentiary value may be limited. Hacking's description of what makes mature laboratory sciences highly reproducible is consistent with our definition of reproducibility rate as a function of true model, assumed model, and methods.

An example of a theory from laboratory sciences that is not directly compared to 'the world' comes from cognitive science. One high level theory that has become prominent in this field over the last two decades is the 'probabilistic' or 'Bayesian' approach to describing human learning and reasoning [59,60]. As the paradigm rose to prominence, questions were raised as to whether claims of the Bayesian theory of the mind held any truth value at all, in either a theoretical or empirical sense [61].

Within a specific framework, a particular experimental result may have value in connection to a theoretical claim without being tied to the world. For instance, Hayes *et al.* [62] presented several experiments that appear to elicit the 'same' phenomenon in different contexts, and an accompanying Bayesian cognitive model that renders these results interpretable within that framework. On the other hand, rational Bayesian models of cognition have been criticized for not taking into account process-level data and making unrealistic environmental assumptions [63]. These models function at the computational rather than algorithmic level (per Marr's levels of analysis, [64]) and do not aim to explain the true mechanisms underpinning human reasoning [65]. Hence these robust empirical results from experiments that were designed from and adapted to the Bayesian framework do not necessarily imply normative claims about mechanisms underlying human cognition (see the discussion in [62], pp. 40–44).

As this example illustrates, Hacking's observations about the 'mutual tuning' between theoretical claims and laboratory manipulations are observed in practice, in cognitive science and potentially in other disciplines. Our measurement error example shown in figure 2 provides just one possible realization for Hacking's conjecture (see also [66], for a detailed discussion on measurement practices that might exacerbate measurement error). Other forms of inference under model misspecification might present different scenarios under which this mutual tuning may take place—for example, the inadvertent introduction of an experimental confound or an error in a statistical computation have the potential to create and reinforce perfectly reproducible *phantom* effects. The possibility of such tuning renders suspect the idea that reproducibility is a good proxy for assessing the truth potential of a result.

The reform movement began as a response to the proliferation of false results in scientific literature. Our formal analysis suggests that if we were to treat observed reproducibility of a given result as a heuristic to establish its truth value, we might incentivize research that achieves high levels of reproducibility for the wrong reasons (per Goodhart's law) and end up canonizing a subset of false results that satisfy specific criteria without facilitating any true discoveries. Hence we believe that turning reproducibility into a new false idol goes against the essence of the ongoing efforts to reform scientific practice.

# 3. Claim 2: using data more than once invalidates statistical inference

A well-known claim in the methodological reform literature regards the (in)validity of using data more than once, which is sometimes colloquially referred to as *double-dipping* or *data peeking*. For instance,

Wagenmakers *et al.* ([13], p. 633) decry this practice with the following rationale: 'Whenever a researcher uses double-dipping strategies, Type I error rates will be inflated and *p* values can no longer be trusted.' The authors further argue that 'At the heart of the problem lies the statistical law that, for the purpose of hypothesis testing, the data may be used only once.' Similarly, Kriegeskorte *et al.* ([67], p. 535) define double-dipping as 'the use of the same data for selection and selective analysis' and add the qualification that it would invalidate statistical inference 'whenever the test statistics are not inherently independent of the selection criteria under the null hypothesis.' This rationale has been used in reform literature to establish the necessity of preregistration for 'confirmatory' statistical inference [13,22].

In this section, we provide examples to show that it is incorrect to make these claims in overly general terms. The reform literature is not very clear on the distinction between 'exploratory' and 'confirmatory' inference. We will revisit these concepts in the next claim but for now, we evaluate the claim that using data multiple times invalidates statistical inference. For that, we will steer away from the exploratory-confirmatory dichotomy and focus on the validity of statistical inference specifically.

The phrases *double-dipping*, *data peeking*, and *using data more than once* do not have formal definitions and thus they cannot be the basis of any *statistical law*. These verbally stated terms are ambiguous and create a confusion that is non-existent in statistical theory.

A correct probability theory approach to establish the effect of using the data—in any way—is to derive the distributions of interest that will make the procedure valid under that usage. In fact, many well-known valid statistical procedures use data more than once (see [68], for a detailed analysis in the context of data dependent priors). In these procedures, the conditioning is already taken into account while deriving the correct probability distribution of the quantity of interest. The consumers of statistical procedures are often not exposed to steps involved in derivations and it might be surprising to find that some of the well-known statistical procedures actually use the data more than once. Colloquially, phrases such as double-dipping, data peeking, and using data more than once might be associated with practices such as model selection followed by inference and sequential testing. However, here, we pick a somewhat unusual example to make our point clear. Our main message is that one has to think carefully and formally what these phrases actually might mean.

We consider testing whether the population mean $\mu$, of a normally distributed random variable $X$ is equal to a fixed value $\mu_o$. We assume that we have a simple random sample of size $n$ from $X \sim \text{Nor}(\mu, \sigma)$ where $\sigma$ is the population standard deviation.

If we start to develop a test using the sample mean $\bar{X}$, a reasonable development towards obtaining a test statistic would be as follows: under $H_o$ we have $X \sim \text{Nor}(\mu_o, \sigma)$, and thus $\bar{X} \sim \text{Nor}(\mu_o, \sigma/\sqrt{n})$, and so we must have

$$\frac{(\bar{X} - \mu_o)}{(\sigma/\sqrt{n})} \sim \text{Nor}(0, 1). \tag{3.1}$$

The test statistic in equation (3.1) is distributed as standard normal and therefore the test is a *z*-test. This is all good, however, the test requires knowing $\sigma$, which we often do not. To surpass this issue, we now think of extracting the sample standard deviation from the data (using the data once more) and substitute it as an estimate of $\sigma$ in equation (3.1) so that we can perform the test. However, because we use the sample quantity $s$, the distribution of the new statistic is not standard normal anymore. What we can do, however, is to derive the correct probability distribution of the new statistic and still have a valid test. Indeed the quantity

$$\frac{(\bar{X} - \mu_o)}{(s/\sqrt{n})}, \tag{3.2}$$

is *t*-distributed and results in a *t*-test. Technically, the quantity in equation (3.2) uses the data at least three times, specifically to obtain $n$, $\bar{X}$, and $s$. Although this example is simplistic, its main point is instructive: irrespective of how many times the data is used or whether it is used in a single-step or a multi-step fashion, if the correct distribution of a test statistic can be derived via appropriate conditioning or from scratch, then it must yield a valid statistical procedure.

The principle of deriving the correct distribution of statistics to obtain a valid statistical procedure also applies when we perform a variety of statistical activities on the data prior to an inferential procedure of specific interest. These activities can be of any type, including exploration of the data by graphical or tabular summaries, or performing other formal procedures such as tests for assumption checks (see [69], for a formal approach for testing model assumptions). In fact, one can even build a valid statistical test by using the data to obtain almost all aspects of a hypothesis test that are not

specifically user-defined, including the hypotheses themselves. The key to validity is not how many times the data are used and for which type of activity, but appropriate application of the correct conditioning as dictated by probability calculus as information from the data is extracted with these activities [70]. When deriving valid statistical procedures, these rules must invariably hold for all cases of manipulations of random variables, whether it is a *t*-pivot, or a multi-step analysis. This is a mathematical fact and the validity of statistical procedures depend only on mathematical facts. Furthermore, under many cases, the conditioning does not affect the validity of the test of interest, and therefore can be dropped, freeing the data from its prison for use prior to test of interest [71].

When conditioning on prior activity on the data is indeed needed to make a test valid, overlooking that a procedure should be modified to accommodate this prior activity might lead to an erroneous test. However, this situation only arises if we disregard the elementary principles of statistical inference such as correct conditioning, sufficiency, completeness and ancillarity. Conditional inferences are statistically valid when their interpretation is properly conditioned on the information extracted from the observed data, which are sufficient for model parameters. Therefore, unconditionally stating that *double-dipping*, *data peeking*, or *using data more than once* invalidates inference does not make statistical sense. In contrast with common reform narratives, one can use the data many times in a valid statistical procedure. Below, we describe the conditions under which this validity is satisfied. We also discuss why preregistration cannot be a prerequisite for valid statistical inference, confirmatory or otherwise.

## 3.1. Valid conditional inference is well established

Imagine we aim to confirm a scientific hypothesis of interest which can be formulated as a statistical hypothesis and be tested using a chosen a test of interest. We suppose that we perform some statistical activity on the data as described in the previous section, until we begin the test of interest. We aim to assess the effect of information gained by this activity on the validity of the test of interest to be performed. To be useful in establishing results, it is necessary to assume that such information can be summarized by a statistic, as in *a statistic obtained from prior analyses.*

First, we categorize the amount of information contained in the test statistic of interest. This statistic may contain anywhere from *no information* to *all information* in the data about the parameter of interest. Furthermore, it can satisfy some statistical optimality criterion, in which case it is identified as the best statistic with respect to this criterion. The case of no information is trivial and not interesting. The case of all information is well known.[5] For many commonly used models, an optimal statistic is also well known[6] (first column in left and right blocks, box 2). Other cases include partial information (second column in left and right blocks, box 2).

Second, the statistic that summarizes the analyses performed on the same data prior to the test of interest may also contain anywhere from no information to all information in the data (rows in left and right blocks, box 2). However, here the case of no information is *also* of interest.[7]

If the statistic summarizing the prior analysis is used in a subsequent analysis for the test of interest, the validity of the test is guaranteed by conditioning the subsequent analysis on this statistic, using probability calculus. A relatively simple case may involve only conditioning on the statistic obtained from prior analysis (left block, box 2). In this case, no quantity exogenous to the model generating the data is introduced into the test of interest. If the test of interest uses an optimal statistic (which is the case for many well-known models), the conditioning is irrelevant because the validity of the test is not affected by the prior information (left block first column in box 2). The same result with the same validity is obtained *as if* we did not perform any activity on the data, previous to the test of interest. Hence, one can freely use information prior to performing the test of interest without any modification in the test of interest. If the test of interest does not use an optimal statistic, then conditioning will maintain the validity and often improve the performance of the test (left block second column in box 2). This is a manifestation of Rao–Blackwellization of the test statistic to reduce its variance. We reproduce an example by Mukhopadhyay [72] of estimating the parameter of a normal distribution whose mean and standard deviation are equal using a randomly sampled single observation in figure 3. Therefore, claim 2 is false for this case. Furthermore, results showing this falsity can be generalized beyond hypothesis testing into other modes of inference such as

---

[5]Sufficient statistic.

[6]Complete sufficient statistic.

[7]Ancillary statistic.

**Box 2.** Valid inference using data multiple times.

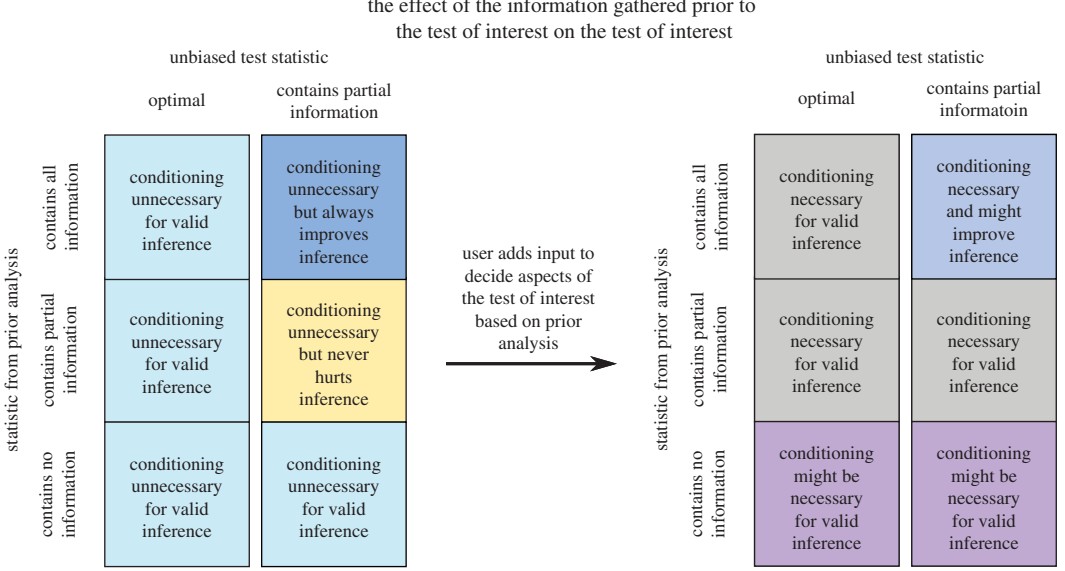

We assume a test based on an unbiased test statistic generates valid inference, in the sense of achieving its nominal Type I error probability, under its assumptions within the Neyman–Pearson hypothesis testing paradigm. Information extracted from the data prior to the test of interest is represented by a statistic from prior analysis. Cells describe the necessity and/or the outcome of conditioning the test of interest on this statistic from prior analysis, for varying levels of information captured. Some technical clarifications for special cases are discussed in appendix C.

**Left**. The statistic from prior analysis is not used in decision making, for example, by combining it with a user-defined criterion which might affect aspects of the test of interest. Many commonly used linear models fall in the first column where procedures are based on an optimal test statistic and therefore, using the information from prior analysis does not affect the validity of the test of interest. However, even if the statistic for the test of interest is not optimal, conditioning on the statistic from prior analysis is not necessary for validity of inference. Furthermore, conditioning never hurts the validity of inference and improves the performance in most cases. Details of the conditional analyses in this block are provided in propositions 3.1 and 3.2.

**Right**. The statistic from prior analysis is combined with a user-defined criterion to affect aspects of the test of interest through a decision. An example is using the data to determine which subsamples to compare. The validity of the test of interest is maintained when inference is conditioned on this decision if the statistic from prior analysis contains at least some information about the parameter to be tested.

The change in corresponding cells between the left block and right block shows the effect of using this user-defined criterion on conditional statistical inference.

estimation. Formally, we have the following definition and results (per steps 1 and 2 of our formal approach, respectively).

**Definition.** Let $S_n \sim \mathbb{P}(S_n|\theta)$ be a test statistic such that it is: (1) a function of an unbiased estimator of $\theta$, and (2) fixed prior to seeing the data. Let $U \sim \mathbb{P}(U|\theta)$ be a statistic obtained from the data, after seeing the data. If $U$ is complete sufficient for $\theta$, it is denoted by $U_s$, and if $U$ is ancillary for $\theta$, it is denoted by $U_a$.

**Proposition 3.1.** Let $S'_n = \mathbb{E}(S_n|U_s)$. For an upper tail test, define $\alpha = \mathbb{P}(S_n \geq s_\alpha|H_o) = \mathbb{P}(S'_n \geq s'_\alpha|H_o)$. Then, $s_\alpha \geq s'_\alpha$ and $\mathbb{P}(S'_n \geq s_\alpha|H_o) < \alpha$. Parallel arguments hold for lower and two tail tests.

Rao–Blackwellization and power

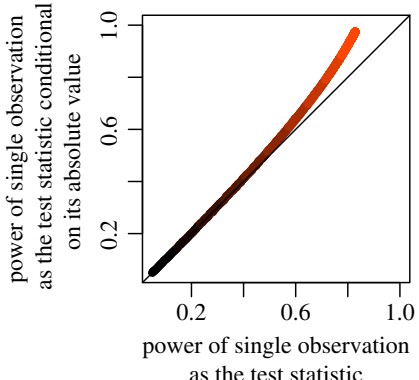

**Figure 3.** For a normally distributed variable with equal mean and variance, we randomly sample a single observation from the population. We plan to use this observation as a test statistic for the common parameter. However, prior to this test we observe the absolute value of the sample and we decide to perform the test using the information in both the observation and its absolute value, therefore, using the unsigned part twice. The plot compares power of the test based on the single observation and on the single observation conditioned on its absolute value. Conditioning improves inference by reducing the variance of the test statistic. This case corresponds to the left block, first row, second column in box 2. Lighter shades represent larger true parameter values. Technical details are given in appendix D.

**Proposition 3.2.** *Let* $H_o : \theta \in \Theta_o$ *such that* $\Theta_o = g(U_a)$, *where g is a known function and* $U_a$ *is a function of the data. Then, the upper tail test* $\mathbb{P}(\mathbf{S_n} \geq s | H_o) \leq \alpha$ *is a valid level* $\alpha$ *test. Parallel arguments hold for lower and two-tailed tests.*

(See appendix C for proofs per step 3 of our formal approach).

An intuitive interpretation of these formal statements is as follows. Assume a hypothesis test where a statistical procedure is pre-planned in the sense that its elements are determined before seeing the data. We then imagine using the data to obtain other statistics (necessarily after seeing the data). The propositions consider two scenarios regarding these statistics. In the first scenario, we consider a statistic that captures all the information in the data about the parameter being tested in a most efficient manner. Then, conditioning on this statistic results not only in a valid procedure, but also an equally good or improved one with respect to the pre-planned procedure. In the second scenario, we consider a statistic that contains no information about the parameter being tested. The null hypothesis is built using this statistic obtained from the data, and the test based on the pre-planned procedure still remains valid.

A more complicated case occurs when one not only obtains a statistic from prior analysis but also makes a decision to redefine the test of interest based on the observed value of that statistic—a decision that depends on an exogenous criterion and alters the set of values the test statistic of interest is allowed to take (right block, box 2). For example, an exogenous criterion might be to perform the test only if the statistic from prior analysis satisfies some condition. Subgroup analyses or determining new hypotheses based on the results of prior analysis (HARKing) are other examples [73]. Conditional quantities which make the test of interest valid are now altered because conditioning on *a statistic* and conditioning on *whether a statistic obeys an exogenous criterion* have different statistical consequences. If this criterion affects the distribution of the test statistic of interest, then conditioning is necessary. The correct conditioning will modify the test in such a way that the distribution of the test statistic under the null hypothesis is derived, critical values for the test are re-adjusted, and desired nominal error rates are achieved. A general algorithm to perform statistically valid conditional analysis in this sense is provided in appendix E. Adhering to correct conditioning, then, guarantees the validity of the test, making claim 2 false again.

Figure 4 provides an example of how conditioning can be used to ensure that nominal error rates are achieved (step 4a). We aim to test whether the mean of population 1 is greater than the mean of population 2, where both populations are normally distributed with known variances. An appropriate test is an upper-tail two-sample $z$-test. For a desired level of test, we fix the critical value at $z$, and the test is performed without performing any prior analysis on the data. The sum of black and dark orange areas under the black curve is the nominal Type I error rate for this test. Now, imagine that we perform some prior analysis on the data and use it only if it obeys an exogenous criterion: we do not

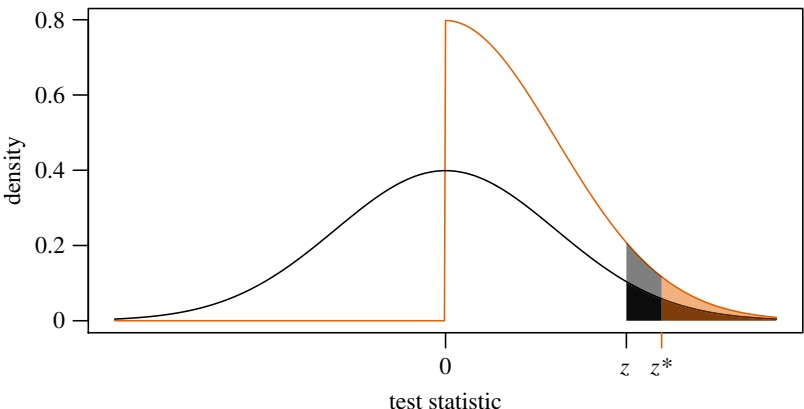

**Figure 4.** For a two sample $z$-test, we display rejection regions for an unconditional test and a conditional test, setting the alternative hypothesis in the direction of the observed effect. The black curve shows the distribution of the unconditional test statistic, with the critical value given by $z$. The orange curve shows the distribution of the conditional test statistic, with the adjusted critical value given by $z^*$.

perform our test unless 'the mean of the sample from population 1 is larger than the mean of the sample from population 2.' This is an example of us deriving our alternative hypothesis from the data. The test can still be made valid, but proper conditioning is required. If we do not condition on the information given within quotes and we still use $z$ as the critical value, we have inflated the observed Type I error rate by the sum of the light grey and light orange areas because the distribution of the test statistic is now given by the orange curve. We can, however, adjust the critical value from $z$ to $z^*$ such that the sum of the light and dark orange areas is equal to the nominal Type I error rate, and the conditional test will be valid. This case corresponds to the right block, first row, first column in box 2. Technical details are provided in appendix D.

Caution with regard to double-dipping might sometimes be justified. However, the claim that it invariably invalidates statistical inference is unsupported. In fact, the opposite is true because all cells in box 2 yield valid tests. Following steps 1-4a of our formal approach, we established some foundations for claims regarding double-dipping. These are summarized in box 2. Furthermore, we provide a fairly generic algorithm (appendix E) to obtain the sampling distribution of *any* statistic conditional on using some information in the data. Statistically and computationally nimble readers should find it straightforward to apply this algorithm to specific double-dipping problems they encounter. On the other hand, extending theoretical results from steps 1-4a of the formal approach to its applied part of steps 4b and 5 typically takes intensive work. This, for example, involves developing user friendly and well-tested tools of analysis, ready for mass consumption to perform conditional inference in a specific class of statistical models.

Clearly, proper conditioning solves a statistical problem. However, the garden of forking paths applies to problems of scientific importance as well, because our conclusions become dependent on decisions we make in our analysis. Statistical rigor is the prerequisite of a successful solution, but we should ask: solution to which problem? Statistical validity does not necessarily imply scientific validity [74]. The connection between statistical and scientific models might be weak—a problem that cannot be fixed by statistical rigor.[8] Furthermore, valid inference by proper conditioning entails maintaining the same conditioning for correct interpretation of scientific inference.

Conditioning is not the only statistically viable way to address double-dipping related problems. Alternatives to conditioning include but are not limited to multilevel modelling [16,85], multiverse analysis [86], simultaneous inference for valid data-driven variable selection [87], sequential or stepwise model selection procedures for optimal post-selection inference [88,89], and iterative Bayesian workflow [90]. The key to successfully implement these solutions is a good understanding of statistical theory and a careful interpretation of results under clearly stated assumptions.

---

[8]Testing hypotheses with no theory to motivate them is a fishing expedition regardless of methodological rigor. See [44,75–84] for discussions on scientific theory.

## 3.2. Preregistration is not necessary for valid statistical inference

Nosek *et al.* [22, p. 2601] claim that 'Standard tools of statistical inference assume prediction'[9]. Nosek *et al.* [22] intend to convey that in hypothesis testing, the analytical plan needs to be determined (i.e. preregistered) prior to data collection or observing the data for statistical inference to have diagnostic value, that is, to be valid. Wagenmakers *et al.* [13] suggest that preregistration would allow for confirmatory conclusions by clearly separating exploratory analyses from confirmatory ones and preventing researchers from fooling themselves or their readers. According to the methodological reform, any inferential procedure that is not preplanned or preregistered should better be categorized as *postdiction* or *exploratory* analysis, and should not be used to arrive at *confirmatory* conclusions [21,91].

In this section, we first clarify the *statistical* problem which preregistration aims to address. Then we assess what preregistration cannot statistically achieve under its strict and flexible interpretation. We discuss how preregistration can harm statistical inference while trying to solve its intended problem. After showing that preregistration is not necessary for valid statistical inference, we describe what it can achieve statistically.

*What is the statistical problem that preregistration aims to address?* Statistically, preregistration is offered as a solution to the problem of using data more than once and issues of validity of statistical procedures resulting from this usage [13,22,92,93]. Once a hypothesis and an analytical plan is preregistered, the idea is that researchers would be prevented from performing analyses that were not preregistered and subsequently, from presenting them as 'confirmatory'. We have shown that using data multiple times per se does not present a statistical problem. The problem arises if proper conditioning on prior information or decisions is skipped. The reform literature misdiagnoses the problem as an ordinal issue regarding the order of: hypothesis setting, decisions on statistical procedures, data collection, and performing inference. Preregistration locks this order down for an analysis to be called 'confirmatory'. Our examples of valid tests in box 3 (per step 4a of our formal approach) show that the problem is not ordinal but one of statistical rigor. Prediction and postdiction—as proposed by Nosek *et al.* [22]—do not have technical definitions in their intended meaning that reflects on statistical procedures. Furthermore, the reform literature does not present any theoretical results to show the effects of this dichotomy on statistical inference. All well-established statistical procedures deliver their claims when their assumptions are satisfied. Other non-mathematical considerations are irrelevant for the validity of a statistical procedure. A valid statistical procedure can be built either before or after observing the data, in fact, even after using the data if proper conditioning is followed. Therefore, the validity of statistical inference procedures cannot depend on whether they were preregistered.

*How can preregistration (strict or flexible) harm statistical inference?* Preregistration may interfere with valid inference because nothing prevents a researcher from preregistering a poor analytical plan. Preregistering invalid statistical procedures does not on its own ensure the validity of inference (see also [73]), while it does add a superficial veneer of rigor.

Assume hypotheses, study design, and an analysis plan are preregistered, and the researchers follow their preregistration to a T. Many hypothesis tests make parametric assumptions and not all are robust to model misspecification. Dennis *et al.* [94] show that under model misspecification, the Neyman–Pearson hypothesis testing paradigm might lead to Type I error probabilities approaching 1 asymptotically with increasing sample sizes. Model misspecification is suspected to be common in scientific practice [74,81,95]. Because the validity of a statistical inference procedure depends on the validity of its assumptions, performing assumption checks—where it is possible and sensible to do so—to choose and proceed with the model and method whose assumptions hold is sound practice. Assumption checks are performed *after* data collection and on the data, but *before specifying a model and a method for analysis*. To accommodate assumption checks under preregistration philosophy, an exception would need to be made to the core principle because they necessitate using data multiple times. Indeed such exceptions are often made [22,92] and it has been suggested that assumption checks and contingency

[9]Prediction here is not used in statistical sense but refers to 'the acquisition of data to test ideas about what will occur' ([22], p. 2600). To clarify, statistics uses sample quantities (observables) to perform inference on population quantities (unobservables). Inference, therefore, is about unobservables. Statistical prediction, on the other hand, is defined as predicting a yet unobserved value of an observable and therefore, is about observables. The quote refers to a procedure about unobservables and hence 'prediction' is not used in a statistical sense. Instead, it is used to demarcate the timing of hypothesis setting and analytical planning with regard to data collection or observation. The authors also specifically refer to the null hypothesis significance testing procedure as *the standard tool for statistical inference* referenced in this quote. While the statement itself can be misleading because of these local definitions and assumptions, our aim is to critique the intended meaning not the idiosyncratic use of statistical terminology.

plans should be preregistered. However, no statistical reasoning is provided to define the boundaries of such deviations from preregistration.

A common reform slogan states that 'preregistration is a plan, not a prison,[10]' offering an escape route from undesirable consequences of rigidity. Nosek *et al.* ([22], p. 2602) suggest that compared to a researcher who did not preregister their hypotheses or analyses, 'preregistration with reported deviations provides substantially greater confidence in the resulting statistical inferences.' This statement has no support from statistical theory. On the other hand, the claim may make researchers feel justified in changing their preregistered analyses as a result of practical problems in data collection or analysis, without accounting for the conditionality in their decisions, leading to invalid statistical inference.

A study of 16 *Psychological Science* papers with open preregistrations shows that research often deviated from preregistration plans [96]. Hence, in practice, preregistration fails to lock researchers in an analytical plan. Deviating from a preregistered plan might prevent a statistically flawed procedure from being implemented, and hence, might improve statistical validity of conclusions. On the other hand, it is possible to deviate from a plan by introducing more sequential decisions and contingency to data analysis, which if not accounted for, would invalidate the statistical inference. A strict interpretation of preregistration may also lead to invalid inference by locking researchers in a faulty plan. As such, preregistration or deviations from preregistration have little say over the diagnosticity of *p*-values or error control. Statistical rigor can neither be ensured by preregistration nor would be compromised by not preregistering a plan.

*What can preregistration achieve statistically?* Strict preregistration might work as a behavioural sanction that prevents researchers from doing any statistical analysis that involves conditioning on data, valid or invalid. This way, preregistration can prevent using data multiple times without proper conditioning by preventing proper conditioning procedures along with it. Nevertheless, as we show in box 2, conditioning on data may improve inference. On the other hand, a flexible interpretation of preregistration that allows for deviations in the plan so long as they are labelled as 'exploratory' rather than 'confirmatory' has no bearing on statistical outcomes. If proper conditioning is performed, analyses that are referred to as 'exploratory' in the reform literature might observe strict error control and if it is not, analyses currently being labelled 'confirmatory' might be statistically uninterpretable.

There exist other social advantages to preregistration of empirical studies, such as the creation of a reference database for systematic reviews and meta-analysis that is relatively free from publication bias. While these represent genuine advantages and good reasons to practise preregistration, they do not affect the interpretation or validity of the statistical tests in a particular study. We demonstrate some of the points discussed in this section with examples in box 3. Our exposition and illustration in this section have policy implications, primarily suggesting caution when proceeding to step 5 of our formal approach in this context. The statistical theory behind these examples show that the benefits of preregistration—in promoting systematic documentation and transparent reporting of hypotheses, research design and analytical procedures—should not be mistaken for a technical capacity for ensuring statistical validity. If and only if a statistically appropriate analytical plan has been preregistered and performed, would preregistration have a chance of ensuring the meaningfulness of statistical results. Yet a well-established statistical procedure always returns valid inference, preregistered or not.

# 4. Claim 3: exploratory research uses 'wonky' statistics

A large body of reform literature advances the exploratory-confirmatory research dichotomy from an exclusively statistical perspective. Wagenmakers *et al.* [13, p. 634] argue that purely exploratory research is one that finds hypotheses in the data by *post hoc* theorizing and using inferential statistics in a 'wonky' manner (borrowing Wagenmakers *et al.*'s [13] terminology) where *p*-values and error rates lose their meaning: 'In the grey area of exploration, data are tortured to some extent, and the corresponding statistics is somewhat wonky.' The reform movement seems to have embraced Wagenmakers *et al.* [13]'s distinction and definitions, and this dichotomy has been emphasized in required documentation for preregistrations [97], registered reports [21], and exploratory reports [98].

We start by discussing why the exploratory-confirmatory dichotomy is not tenable from a purely statistical perspective. The reform literature does not provide an unambiguous definition for what is

---

[10]While not part of our core argument this particular slogan is underspecified. It is not clear how the argument for the necessity of preregistration for statistically valid inference should be reconciled with the proposed flexibility of preregistrations. In any case, this line of thinking is moot from our perspective because the underlying premise itself does not hold.

**Box 3.** Validity of statistical analyses under strict, flexible and no preregistration.

We show how a strict interpretation of preregistration and a failure to use proper statistical conditioning may hinder valid statistical inference with a simulation example. Our simulations consist of $10^6$ replications of hypothesis tests for the difference in the location parameter between two populations. We build the distribution of $p$-values under the null hypothesis of no difference for three cases and four true data generating models. In addition to the normal distribution with exponentially bounded tail, we use Cauchy and T distributions for heavy tail, and Gumbel distribution for light tail. By a well-known result, the distribution of $p$-values under the null hypothesis is standard uniform for a valid statistical test.

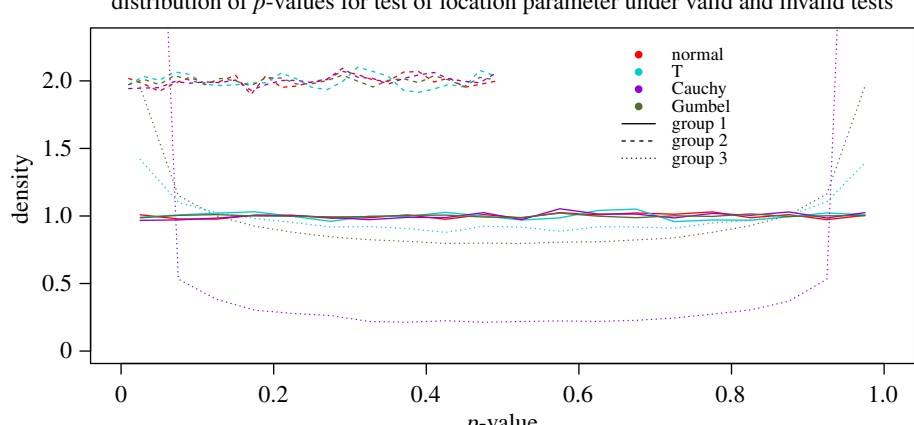

distribution of $p$-values for test of location parameter under valid and invalid tests

— Hypothesis tests in group 1 (solid lines) were performed using the following procedure:
 (i) collect data with no specification of hypothesis, model, or method (no preregistration);
 (ii) calculate the sample medians. Set the alternative hypothesis so that the median of the population corresponding to the larger sample median is larger than the median of the other population (using the data to determine the hypotheses);
 (iii) build the *conditional* reference distribution of the test statistic by permuting the data (reusing the data to determine the method); and
 (iv) calculate the test statistic from the data to compare with the reference distribution (reusing the data to calculate observed value of the test statistic).

The tests in group 1 derive almost all their components from the data by reusing them multiple times. The distribution of the $p$-values show that these tests are valid because they follow the standard uniform distribution (solid lines).

— Hypothesis tests in group 2 (dashed lines) demonstrate a situation that may arise under either flexible preregistration (assumption checks allowed) or no preregistration, when proper statistical conditioning is not performed in step 3. This is akin to HARKing without statistical controls. In this case, the distribution of $p$-values is uniform on $(0, 0.5)$. These tests are not valid, because $\mathbb{P}(p \leq \alpha | H_0) = 2\alpha$ for some significance thresholds $\alpha$.

— Hypothesis tests in group 3 (dotted lines) demonstrate a situation that may arise under a strict preregistration protocol (altering the preregistered model or methods not allowed) when there is model misspecification. The preregistered model is normal, but the data are generated under other models. These tests are not valid, because $\mathbb{P}(p \leq \alpha | H_0) > \alpha$ for some significance thresholds $\alpha$.

considered 'confirmatory' or 'exploratory'. There are many possible interpretations including: (i) formal statistical procedures such as null hypothesis significance testing are confirmatory, informal ones are exploratory; (ii) only preregistered hypothesis tests are confirmatory, non-preregistered ones are exploratory; and (iii) only statistical procedures that deliver their theoretical claims (e.g. error control) are confirmatory, invalid ones are exploratory. These three dichotomies are not consistent with each other and lead to confusing uses of terminology. One can speak of formal statistical procedures such as significance tests, and informal procedures such as data visualization, or valid and invalid

statistical inference, but there is no mathematical mapping from these to exploratory or confirmatory research, especially when clear technical definitions for the latter are not provided, in clear violation of step 1 of our formal approach. Moreover, the general usefulness and relevance of this dichotomy has also been challenged for theoretical reasons [79,80]. In this section, we sidestep issues with the dichotomy but argue against the core claim presented by [13] regarding the nature of exploratory research specifically, advancing the following points:

— exploratory research aims to facilitate scientific discovery, which requires a broader approach than statistical analysis alone and cannot be evaluated formally to derive meaningful methodological claims;
— exploratory data analysis (EDA) is a tool for performing exploratory research and uses methods that only answer to their assumptions to be valid. When making claims about EDA specifically, we should follow the steps of our formal approach;
— using 'wonky' inferential statistics does not facilitate and probably hinders exploration, because statistical theory only provides guarantees for statistical inference when its assumptions are met; and
— exploratory research needs rigor to serve its intended aim of facilitating scientific discovery.

Scientific exploration is the process of attempting to discover new phenomena [99]. Outside of the methodological reform literature, exploratory research is typically associated with hypothesis generation and is contrasted with hypothesis testing—sometimes referred to as confirmatory research. Exploratory research may lead to serendipitous discoveries. However, it is not synonymous with serendipity but is a deliberate and systematic attempt at discovering generalizations that help us describe and understand an area about which we have little or no knowledge [100]. In this sense, it is analogous to topographically mapping an unknown geographical region. The purpose is to create a complete map until we are convinced that there is no element within the region being explored that remains undiscovered. This process may take many forms from exploration of theoretical spaces (i.e. theory development; [82,83]) and exploration of model spaces [77,101] to conducting qualitative exploratory studies [102] and designing exploratory experiments [103,104], and finally to exploratory data analysis [105–108].

This process of hypothesis generation is notoriously hard to formalize, as Russel ([109], p. 544) so clearly laid out:

> As a rule, the framing of hypotheses is the most difficult part of scientific work, and the part where great ability is indispensable. So far, no method has been found which would make it possible to invent hypotheses by rule. Usually, some hypothesis is a necessary preliminary to the collection of facts, since the selection of facts demands some way of determining relevance. Without something of this kind, the mere multiplicity of facts is baffling.

Therefore, without further work on formal approaches it is not easy to implement a formal approach to make methodological claims about exploration, since we will fail at step 1. At least in our current knowledge state, we are not able to formally define exploration as a research activity. Informally, hypothesis generation requires creativity, flexibility, and open-mindedness to allow for ideas to emerge [99,100]. The inferential approach employed during exploration cannot be described as deduction or induction as it requires adding something new to known facts. This process of generating explanatory hypotheses is known as *abduction proper*[11] [112], which involves studying the facts and generating a theory to explain them ([112], p. 90). Abduction proper requires scientists to absorb and digest all known facts about a phenomenon, mull them over, use introspection and common sense [113], evaluate them against their background knowledge [83], and add something as of yet unknown, with the intention of providing new insight or understanding that would not have been possible without abduction [112]. Hypothesis generation, therefore, cannot be reduced down to formal statistical inference, whose methods are deductively derived and used inductively in application. In fact, meticulous exploration via abduction proper would improve our statistical inference by facilitating the first two conditions mentioned in box 1 by constraining our search space in a theoretically meaningful fashion.

That said, exploratory data analysis (EDA) can be instrumental in hypothesis generation. Tukey [108] suggests that EDA is not a bundle of formal inferential techniques and that it requires extensive use of

---

[11]Abductive inference involves both the process of making inference to the best explanation based on a set of candidate hypotheses [110]. and the process of generating that set of hypotheses. The latter process, which is of interest to our discussion, is specifically known as *abduction proper* [83,111]. Abduction proper is then a way to meaningfully reduce the search space for possible hypotheses. Blokpoel *et al*. [111] show that abduction proper is uncomputable when unconstrained and remains computationally intractable even when constrained. This seems to render attempts at efficiently capturing this process with rules and formalism somewhat futile.

data visualization with a flexible approach. EDA is usually an iterative process of model specification, residual analysis, examination of assumptions, and model respecification [77,105] to find patterns and reveal data structure. If inferential statistics are employed for the purposes of data exploration, we can prioritize minimizing the probability of failing to reject a false null hypothesis [114,115] as opposed to minimizing false positives because priority is given to not missing true discoveries. Nonetheless, other methods than hypothesis testing are often more closely associated with EDA owing to their flexibility in revealing patterns, such as graphical evaluation of data [105,108], exploratory factor analysis [105,107], principal components regression [116] and Bayesian methods to generate EDA graphs [106,117,118].

Whichever method is selected for EDA; however, it needs to be implemented rigorously to maximize the probability of true discoveries while minimizing the probability of false discoveries. As Behrens ([105], p. 134) observes:

> A researcher may conduct an exploratory factor analysis without examining the data for possible rogue values, outliers, or anomalies; fail to plot the multivariate data to ensure the data avoid pathological patterns; and leave all decision making up to the default computer settings. Such activity would *not* be considered EDA because the researcher may be easily misled by many aspects of the data or the computer package. Any description that would come from the factor analysis itself would rest on too many unassessed assumptions to leave the exploratory data analyst comfortable.

The implication is that using 'wonky' statistics cannot be a recommended practice for data exploration. The reason is that by repeatedly misusing statistical methods, it is possible to generate an infinite number of patterns from the same data set but most of them will be what Good ([113], p. 290) calls a *kinkus*—'a pattern that has an extremely small prior probability of being potentially explicable, given the particular context'. If the process of hypothesis generation yields too many such kinkera (plural of kinkus), it can neither be considered a proper application of abduction principle nor would serve the ultimate goal of exploratory research: making true discoveries. Relying on statistical abuse in the name of scientific discovery will easily lead to well-known statistical problems such as increasing false positives by multiple hypothesis testing [119], specifically by multiple tests of the same hypothesis [120,121], or by failing to use proper conditioning as we outlined in the previous section.

If exploratory research needs to satisfy a certain level of rigor to be effective but we are not able to formalize it, what criteria should we use to assess its quality? Because the process of exploration is elusive and informal, it may not be possible to derive some minimum standards all exploratory studies need to meet. Nonetheless, some desirable qualities can be inferred from successful implementation of exploratory approaches in different fields: (i) as suggested by Russell's quote, exploration needs to start with subject matter expertise or theoretical background, and hence, cannot be decontextualized, free of theory, or completely dictated by the data [83,102,104,105,111,113]; (ii) the key for running successful exploratory studies is the richness of data [122]. Random datasets that are uninformative about the area to be explored will probably not yield important discoveries; (iii) exploration requires robust methods that are insensitive to underlying assumptions [105]. As such, rather than misusing or abusing standard procedures for inferential statistics, using robust approaches such as multiverse analysis [86] or metastudies [123] could be more appropriate for exploration purposes; and (iv) exploratory work needs to be done in a structured, systematic, honest and transparent manner using a deliberately chosen methodology appropriate for the task [10,122].

The above discussion should make two points clear, regarding claim 3: first, exploratory research cannot be reduced to exploratory data analysis and cannot be formalized, rendering broad methodological claims about exploration unwarranted. Second, when exploratory data analysis is pursued as a preferred method for scientific exploration, it needs rigor and formal justifications. Describing exploratory research as though it were synonymous with or accepting of 'wonky' procedures that misuse or abuse statistical inference not only undermines the importance of systematic exploration in the scientific process but also severely handicaps the process of discovery.

## 5. Conclusion

Our call for statistical rigor and scientific nuance encompasses all claims regarding scientific practice and policy changes. Rigor requires attention to detail, precision, clarity in statements and methods, and transparency. Nuance necessarily means moving away from speculative, sweeping claims and not losing sight of the context of inference. Simple solutions to complex scientific problems rarely exist. Simple fixes motivated by speculative arguments, lacking rigor and proper scientific support might

appear to be legitimate and satisfactory in the short run, but may prove to be counter-productive in the long run. It is instructive to remember how taking $p < 0.05$ as a sign of scientific relevance or even truth has proved to be detrimental to scientific progress.

Recent developments in methodological reform have already been impactful in inducing behavioural and institutional changes. However, as Niiniluoto [124] suggests, impact of research 'only shows that it has successfully 'moved' the scientific community in some direction. If science is goal-directed, then we must acknowledge that movement in the wrong direction does not constitute progress.' Unfortunately, the reform literature has largely overlooked the necessity of first principles and formalism in advancing methodological tools. That is: providing mathematical definitions of fundamental concepts the methods rely on, making claims about these tools with transparency and under clearly stated assumptions, supporting these claims by and mathematical or simulation proofs, and documenting the limitations of these tools. Such a formal approach aids us in making positive contributions to scientific progress. The five-step formal approach we illustrated in this article is just an example of this formalism, showing how to encapsulate the necessary standard for methodological rigor and nuance. With this example, and its application to three proposed reform policies, we hope to contribute to laying the groundwork of *a formal methodology in scientific reform*.

Data accessibility. All code generating the data and figures are attached in the electronic supplementary files.
Authors' contributions. B.D. conceived of the idea. E.O.B. developed statistical theory and proofs. B.D. and E.O.B. conceptualized the examples and illustrations, drafted, and revised the manuscript. J.V. contributed to writing, revising, and editing. D.J.N. contributed to writing. All authors gave final approval for publication and agree to be held accountable for the work performed therein.
Competing interests. We declare we have no competing interests.
Funding. B.D. and E.O.B. were supported by NIGMS of the NIH under award no. P20GM104420. J.V. was supported by grant nos 1850849 and 1658303 from the National Science Foundation's Cognitive Neuroscience panel.
Acknowledgements. The authors thank Iris van Rooij for her generous and astute feedback on a previous draft of the manuscript as well as all the insights she has contributed via many relentless discussions with B.D., as well as Sarahanne M. Field, Daniele Fanelli and John T. Ormerod for their insightful and helpful comments.

# Appendix A

## A.1. Regularity conditions and notation

We assume some regularity conditions which are sufficient for our purposes here for all random variables:

— distribution functions $F \equiv F(w) = \mathbb{P}(W \leq w)$, are absolutely continuous and non-degenerate, endowed with the density function $f(w) = dF(w)/dw$;
— $\{\mathbb{E}(|W|^n) < \infty, \forall n\}$, $\mathbb{E}(W^2) > 0$, where $\mathbb{E}(W) = \int_{-\infty}^{\infty} f(w)dw$, and $\mathbb{V}(W) = \mathbb{E}(W^2) - [\mathbb{E}(W)]^2$; and
— we make frequent use of the indicator function: $\mathbf{I}_{\{A\}} = 1$ if $A$, and 0 otherwise.

## A.2. Assumptions of idealized study

We build on the notion of *idealized study* [101], obeying the following assumptions below:

**A1.** There exists a true probability model $M_T$, completely specified by $F_T$ of random variable $X$, which is the observable for a phenomenon of interest.
**A2.** Some known background knowledge $K$ partially specifies $M_T$ up to property $\theta \in \Theta$, which denotes unknown and unobservable components of $M_T$. For notational economy, $K$ is often dropped, with the understanding that all statements are conditional on $K$.
**A3.** A statement that is in principle testable via statistical inference using a simple random and finite sample $\mathbf{X_n} = (X_1, X_2, \ldots, X_n)$, where $X_i \sim F_T$ is made about $\theta$.
**A4.** Candidate mechanisms $M_i$, inducing distribution functions $F_i$ are formulated.
**A5.** A fixed and known function $S$ is used to extract the information in $\mathbf{X_n}$ pertinent to $M_i$. $S$ evaluated at $\mathbf{X_n}$ returns $\mathbf{S_n}$, with non-degenerate distribution function $\mathbb{P}(\mathbf{S_n} \leq s)$.
**A6.** Formal statistical inference returns a *result* $\{R = d(\mathbf{S_n}, c), R \subset \Theta\}$, where $c$ is a user-defined known quantity, and $d(\cdot, \cdot)$ is a fixed and known non-constant decision function which formalizes the statistical inference (by inducing a frequency assessment for a result).

**Definitions.**

— $\xi = (M_i, \theta, \mathbf{X_n}, S, K, d)$ is an idealized study.
— $\xi^{(i)}$ which differs from $\xi$ only in $K$ and $\mathbf{X_n}^{(i)}$ generated independently from $\mathbf{X_n}$, is a replication experiment.

# Appendix B

## B.1. Relationship between true results and reproducible results

*Proof.* Proofs of propositions 2.1, 2.2 and 2.3. $R^{(i)}$ are {0, 1} exchangeable random variables since $\xi^{(i)}$ are invariant under permutation of labels. By De Finetti's representation theorem for {0, 1} variables, there exists a $\phi$ such that $R^{(i)}$ are conditionally independent given $\phi$. For a finite subsequence $R^{(1)}$, $R^{(2)}$, ..., $R^{(N)}$, and the relative frequency of reproduced results defined by $\phi_N = N^{-1} \sum_{i=1}^{N} \mathbf{I}_{\{R^{(i)}=R_o|R_o\}}$, we have $\lim_{N\to\infty}\phi_N = \phi$, by the Law of Large Numbers.

By definition $\phi \geq 0$, because it is a probability. It follows by contradiction that $\phi = 1$ only in highly specific cases: assume $\phi = 1$. We have $\phi = \mathbb{E}(\mathbf{I}_{\{R=R_o|R_o\}}) = \mathbb{P}(R = R_o|R_o) = 1$, which implies that $\mathbf{I}_{\{R^{(i)}=R_o|R\}} = 1$ for all $i$. Therefore, $d(\mathbf{S_n}, c)$ in **A6** must return a singleton ($R_o$) for all values of $\mathbf{S_n}$. This can happen in three ways: $\mathbf{X_n}$ is non-stochastic, which contradicts **A1**, or $\mathbf{S_n}$ is non-stochastic, which contradicts **A5**, or $R_o$ is not a proper subset of $\Theta$, which contradicts **A6**.

The truth of 1.2 implies 1.3 and vice versa: if a result is not true, then it is false because $\phi_T + \phi_F = 1$. To see that $\phi_T$ can be arbitrarily close to zero (and $\phi_F$ arbitrarily close to 1), fix $R_T$. Choose $S$ such that $d(\mathbf{S_n}, c)$ does not return $R_T$ with probability $1 - \phi_T$. A simple example is a biased estimator of a parameter in a probability distribution. We also note that by proposition 1.1, $\phi_T$ must have positive probability for every point on its support for some $\xi$, which includes values arbitrarily close to 0. ∎

**Remark.** $\phi_N$ should not be misinterpreted as an estimator with less than ideal properties. Quite the opposite: By Central Limit Theorem, $(\phi_N - \phi)/[\phi(1 - \phi)]$ converges to the standard normal distribution and $\phi_N$ has excellent statistical properties as an estimator of $\phi$ [125–127].

## B.2. Remarks for some cases in box 1

**Bullet 1.** Fix $c$ such that $\epsilon(c) > 0$. Consider a model selection problem where $d(\mathbf{S_n}, c)$ returns a model between two candidate models $M_1$ and $M_2$, which are different from the true model $M_T$. The selected model $M_1$ or $M_2$ is false with probability 1 independent of how well $S$ performs. Yet, $M_1$ and $M_2$ can be chosen so that the divergence or metric on which the model selection measure $S$ is based satisfy selecting $M_1$ over $M_2$ with probability $\phi_F = 1 - \epsilon(c)$.

**Bullet 3.** Let $\theta_o$ be the parameter of interest of $F_T$ and $\theta_o'$ be nuisance parameters. Assume that the true value of $\theta_o$ is in $\Theta$. We let $d(\mathbf{S_n}, c)$ to return $\mathbf{S_n}$ as an estimator of parameter $\theta_o$ where $\mathbb{E}(\mathbf{S_n})$ is not equal to the true value. $\mathbf{S_n}$ is often a pivotal quantity. We consider two cases: if furthermore $\mathbf{S_n}$ is a statistic, then it is ancillary for $\theta_o$. Let $\mathbb{V}(\mathbf{S_n}) = \epsilon(c)^2$. By Chebychev's inequality, we have $|\mathbf{S_n} - \mathbb{E}(\mathbf{S_n})| \leq \epsilon(c)$ with probability 1. Thus, the result returned is false and $\phi_F > 1 - \epsilon(c)$. Else if, $\mathbf{S_n}$ is not a statistic, but depends on $\theta_o'$, choosing the value of $\theta_o'$ suitably yields the result.

# Appendix C

## C.1. Conditional analysis

**Proof 2.1.** By Chebychev's inequality, we have $\mathbb{P}(|\mathbf{S_n} - \theta| \leq \sqrt{\mathbb{V}(\mathbf{S_n})}/\alpha) \leq \alpha^2$ and $\mathbb{P}(|\mathbf{S_n'} - \theta| \leq \sqrt{\mathbb{V}(\mathbf{S_n'})}/\alpha) \leq \alpha^2$, where $\mathbb{V}(\mathbf{S_n})/\alpha$ and $\mathbb{V}(\mathbf{S_n'})/\alpha$ are critical values of the two tests. We have $0 \leq \mathbb{V}(\mathbf{S_n'}) \leq \mathbb{V}(\mathbf{S_n})$ by Rao–Blackwell Theorem ([128], p. 342). It follows that $s_\alpha \geq s_\alpha'$ and $\mathbb{P}(\mathbf{S_n'} \geq s_\alpha|H_o) < \alpha$.

**Proof 2.2.** By ancillarity, we have $\mathbb{P}(U_a|\theta) = \mathbb{P}(U_a)$, implying $\mathbb{P}(U_a|\mathbf{S_n}, \theta) = \mathbb{P}(U_a|\mathbf{S_n})$. The sampling distribution of $\mathbf{S_n}$ given $\theta$ can be written as:

$$\mathbb{P}(\mathbf{S_n}|\theta) = \frac{\mathbb{P}(\mathbf{S_n}|U_a, \theta)\mathbb{P}(U_a|\theta)}{\mathbb{P}(U_a|\mathbf{S_n}, \theta)} = \mathbb{P}(\mathbf{S_n}|U_a, \theta)[\mathbb{P}(U_a)/\mathbb{P}(U_a|\mathbf{S_n})],$$

where the second equality follows by substituting for $\mathbb{P}(U_a|\theta)$ and $\mathbb{P}(U_a|\mathbf{S_n}, \theta)$. The term within the brackets is independent of $\theta$, so that a test based on $\mathbf{S_n}$, and a test based on $\mathbf{S_n}|U_a$ yield the same result. Therefore, using $U_a$ to inform $H_o$ does not affect the validity of the test.

**Remarks for some cases in box 2.**

**Left block, 1st row, 1st column.** If $\mathbf{S_n}$ is not complete sufficient and $U_s$ is minimally sufficient, then for an upper tail test $\mathbb{P}(\mathbf{S_n} \geq s|U_s, H_a) \geq \mathbb{P}(\mathbf{S_n} \geq s|H_a)$ for some $s$ is possible, where $H_a$ is the alternative hypothesis. That is, the test conditional on a statistic from prior analysis can be more powerful. Parallel arguments hold for lower and two-tailed tests.

**Left block, 1st row, 2nd column.** Rao–Blackwellization guarantees that $\mathbb{V}(\mathbf{S_n}|U) \leq \mathbb{V}(\mathbf{S_n})$. See figure 3 for an example.

**Right block, 1st row, 1st column.** Conditioning on a decision based on user-defined criterion might alter the support of the sampling distribution of $\mathbf{S_n}$. In these cases, conditioning is necessary for a valid test. See figure 4 for an example.

**Right block, 3rd row.** $U_a$ and $\mathbf{S_n}$ might be dependent (see Casella & Berger ([128], pp. 284–285) for an example). Applying a decision with a user-defined criterion and $U_a$ might affect the support of the sampling distribution of $\mathbf{S_n}$. In these cases, conditioning on the decision regarding $U_a$ is necessary for a valid test.

# Appendix D

## D.1. Details of models used in figures

**Figure 1a.** The simple linear regression model is given by $y_i = \beta_0 + \beta_1 x_i + \epsilon_i$, where the errors obey Gauss–Markov conditions: $\mathbb{E}(\epsilon_i) = 0$, $\mathbb{V}(\epsilon_i) = \sigma_\epsilon^2$, $\forall i$, and $Cov(\epsilon_i, \epsilon_j) = 0$, $\forall(i, j)$. The $x_i$ are assumed fixed and known. The errors $\epsilon_i \sim \text{Nor}(0, \sigma_\epsilon)$. The measurement error model is the true model when there is stochastic measurement error in $x$ making it a random variable $X$. We assume $X_i = x_i + \eta_i$, where $\eta_i \sim \text{Nor}(0, \sigma_\eta)$. The assumed (incorrect) model under which inference is performed is the simple linear regression model, which corresponds to $\sigma_\eta = 0$. Specific values used in the plot are: $x \sim \text{Unif}(0, 10)$, $\beta_0 = 2$, $\beta_1 \in \{2, 20\}$, $\sigma_\epsilon = 1$, $\sigma_\eta \in \{0.01, 0.02, \ldots, 1.0\}$, and the sample size is 50.

**Figure 2.** The model is the same as in figure 1a, except that the values plotted are $\sigma_\eta \in \{0.01, 0.02, \ldots, 10\}$, and the true value is $\beta_1 = 20$. The vertical axis shows the distance between $\hat{\beta}_1$ and $\beta_1$.

**Figure 3.** This example is from Mukhopadhyay [72]. Let $X \sim \text{Nor}(\mu, \mu)$, $\mu > 0$. The data are a single observation $X_1$, which is an unbiased estimator of $\mu$. Using Rao–Blackwellization, $|X_1|$ is a sufficient statistic for $\mu$ and the mean of $X_1$ conditional on the value $|X_1|$ improves the power of a test while maintaining its validity.

**Figure 4.** Let $X_i \sim \text{Nor}(\mu_X, \sigma_X^2)$ and $Y_i \sim \text{Nor}(\mu_Y, \sigma_Y^2)$, $i = 1, 2, \ldots, n$ independent samples with known population variances $\sigma_X^2$ and $\sigma_Y^2$. Let the null and the alternative hypotheses be $H_o : \mu_X = \mu_Y$, $H_a : \mu_X > \mu_Y$, respectively. An appropriate test statistic for level $\alpha = \mathbb{P}(Z \geq z_\alpha|H_o)$ test is the z-score: $Z = (\bar{X} - \bar{Y})/(\sigma_X/\sqrt{n} + \sigma_Y/\sqrt{n})$, which follows a standard normal distribution under $H_o$. Assume we perform the test *only if* we observe $\bar{X} - \bar{Y} > 0$. Define: $U(c) = \bar{X} - \bar{Y}$ if $\bar{X} > \bar{Y}$, and $U(c) = 0$ otherwise. Here, $U(c)$ is the statistic $U = \bar{X} - \bar{Y}$ whose non-zero values are constrained by the user-defined criterion $c$, given by $\bar{X} > \bar{Y}$. The conclusion of the test depends on $U(c)$ because when $\bar{X} > \bar{Y}$, the larger the value of $U$, larger the value of $Z$. The distribution of the conditional test statistic $Z|U(c)$, $H_o$ is not standard normal and therefore the level of the test is not necessarily $\alpha$ for the critical value $z_\alpha$, as is with the test statistic $Z$. However, if the distribution of $Z|U(c)$, $H_o$ is available then the correct critical value, can be chosen to perform a level $\alpha$ test. We let $W = Z\mathbf{I}_{[\bar{X} > \bar{Y}]}$, the standard normal random variable with support on non-negative real line (folded at zero), properly normalized. This is known as the standard half-normal distribution.

We see that $\mathbb{P}(W > z_\alpha|H_o) = 2\alpha$. For the level of the conditional test to be $\alpha$, we adjust the critical value as $z^* = z_{\alpha/2}$ and have $\mathbb{P}(W > z^*|H_o) = \alpha$.

# Appendix E

## E.1. A simulation-based method to sample the conditional distribution of the test statistic

If the distribution of the conditional test statistic under $H_o$ is not available as a closed form solution, an appropriate simulation-based method can be used to sample it. Here, we give an example for the unconditional test statistic $\mathbf{S_n}$ with distribution $\mathbb{P}(\mathbf{S_n}|H_o)$, where $H_o : \theta = \theta_o$. We aim to sample $M$ values from the conditional distribution of $\mathbf{S_n} \,|\, U(c), H_o$ where $U(c)$ is a statistic obtained from the data constrained by a user-defined criterion $c$.

**Algorithm.**

Initialize: Set $M$ (large desired number), and $i = 0$.

Begin While $i < M$, do:

1. Simulate $X_j \sim \mathbb{P}(X_i|\theta_o)$, $j = 1, 2, \ldots, n$ independently of each other. Set $\mathbf{X_n^{(i)}} = (X_1, X_2, \cdots, X_n)$.

2. Calculate $\mathbf{S_n^{(i)}} = S(\mathbf{X_n^{(i)}})$ and $U^{(i)} = U(\mathbf{X_n^{(i)}})$.

3. If $U^{(i)}$ obeys $c$ accept $\mathbf{S_n^{(i)}}$ as a draw from the distribution of the conditional test statistic and set $i = i + 1$. Else discard $(\mathbf{X_n^{(i)}}, \mathbf{S_n^{(i)}}, U^{(i)})$.

End While

The accepted values $\mathbf{S_n^{(1)}}, \mathbf{S_n^{(2)}}, \ldots, \mathbf{S_n^{(M)}}$ is a sample from the distribution $\mathbf{S_n} \,|\, U(c), H_o$. A valid level $\alpha$ test can be built by finding the relevant sample quantile. This method is precise up to a Monte Carlo error which vanishes as $M \to \infty$.

Sometimes it may not be possible to condition on the exact value of statistic $U(c)$, for example when $c$ involves an equality (instead of inequality) and $U$ is continuous random variable. In these cases, the algorithm given above can be modified to build an approximate test using an approximate simulation method such as a likelihood free method. The error rates in approximation can be estimated by simulation.

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
