## [Peer Review File · Royal Society Open Science]

Review History

RSOS-200805.R0 (Original submission)

Review form: Reviewer 1 (Sarahanne Field)

Is the manuscript scientifically sound in its present form?

Yes

Are the interpretations and conclusions justified by the results?

Yes

Is the language acceptable?

Yes

Do you have any ethical concerns with this paper?

No

Have you any concerns about statistical analyses in this paper?

No

Recommendation?

Accept with minor revision (please list in comments)

Comments to the Author(s)

I will start by saying that this paper is a much-needed addition to the metascience and philosophy of science literatures, and would be good in a reading list for undergraduates and PhD students coming into social sciences. I enjoyed reading it.

I agree with the paper's key argument that there is not enough nuance and humility used when it comes to claims and arguments made by reformers; I also agree with the observation that this is very ironic. While I do think it's possible that these issues are marks of a field in its infancy (I'd argue that although metascience as a concept emerged as early as the debates of Neurath, Kellen and Morris in the 30's, the current wave of metascience with emphasis on reform is relatively new), it is vital that practitioners in the field stay humble and cautious in their claims and are reminded to do so.

This article is well-written and thoughtful, uses succinct phrasing and active voice, and has clearly been carefully edited. The statistics seem solid and largely nicely support the conceptual arguments. Many of the proofs in the appendix material are beyond my understanding. This said, I have some comments which I think should be considered before I recommend publication in RSOS. They largely involve language and nuance of some statements. Further, I feel that some argumentation borders on the 'straw-man' kind.

I think this paper should and will be widely read, and I think it has the potential to make impact on how people perceive initiatives and more broadly the 'reform movement'. As a result, I think its arguments should be more strongly made, and contain as much nuance and humility as the paper implores the field to use itself. This is why I am more argumentative and nitpicky in this review than I am in most reviews. My comments are intended as constructive criticism and I hope the authors can accept them in the positive spirit they are meant.

I will elaborate on the issues I perceive directly. They are presented below in chronological order. When I refer to pages and lines, I refer to those of the proof, not those of the manuscript within the proof PDF.

1. P2, L37/38: The authors express here surprise that the field has received little criticism. I think this statement is a little misleading. The field (and reforms it has birthed) certainly have received criticism on a few different levels:

For instance, Strobe and Strack (2014) were highly critical of both the replication crisis and direct replication as a means of rigorous hypothesis testing. Leonelli has been critical of using reproducibility as a yardstick for research quality. Feldman-Barrett has written about her scepticism that the benefits of preregistration/registered reports outweigh the drawbacks. Allen and Mehler discuss problems like flexibility and time when it comes to adopting open science practices. Lancaster is critical about open science principles when it comes to the large numbers of preprints flooding the internet, and Tyfield and Mirawski separately describe concerns about marketing and corporatization of science caused by the release of huge volumes of articles and data (in the form of, for instance, preprints). Makel describes his concerns with open science advances coming too fast for current institutional systems to adjust to (leading to inequality with things like hiring in academia). Bastian and Gelman both fear we are forgetting to do science in the face of having to fix its problems. Bastian in particular seems to be vocal about a few weaknesses in metascientific attempts at reform, for instance, she takes issue with the system of providing badges for things like data sharing.

There is also much criticism starting to come from STS about the lack of depth of metascientific research, among a spate of other complaints (that said, these have yet to make their way into publications from what I see, though these critiques are to be found in rather large numbers on

Twitter), and this does not cover the numerous critiques of individual reform researchers' (such as Heathers and Brown) approaches either.

This is off the top of my head. There are a number of other critiques of metascience/open science reform-focused ideas and incentives that I know I am missing, and then there's the literature I don't know about. My point is that to say that metascience has received little criticism is inaccurate in my opinion. I *would* say that its advocates can be prone to just blindly swallowing the kool-aid though (I myself have been guilty of this in the past).

I suggest to soften the statement that the field has received little criticism in favour of a more nuanced and accurate comment.

2. P4, L6/7: The authors write: "Indeed, many scientific fields have developed their own qualitative and quantitative methods such as ethnography or event study methodology to study non-reproducible phenomena." I think this statement could also do with some nuance. For one thing, ethnography was not developed in the first place to get around the reproducibility problem (not precisely what the authors said, and maybe not what they meant, but the implication is certainly there). While ethnography does have reproducibility issues that are unique to its nature, it is not accurate to say that it is always irreproducible. For instance, consider content analysis on qualitative data. If a coding system for use on a given data set (say, interview data) has been developed well enough by the original coder, it is likely that another researcher who is given the same dataset will come up with a highly comparable set of codes.

Reproducibility should be defined differently for qualitative methods than with quantitative, I would argue, but to imply that reproducibility as a goal is orthogonal to methods like ethnography (and the approaches it involves) is not accurate and will not sit well with many readers (like myself).

I suggest that that statement be adjusted to at least give a nod to the opinions of people like Ingo Rohlfing who consider reproducibility relevant for both qual and quantitative researchers.

3. P9, L50-53: The authors write "It is less clear – even to the authors of the original study – what relationship the robust empirical results have to the true mechanisms..." What is meant by this? Even to the authors of the original study – does this mean that they said something that indicated this, or are the authors inferring that the authors were unclear? Please clarify here what you mean to imply. If the authors stated this themselves, a quotation would be good, and otherwise I would like to know what the authors have inferred and why.

4. P10, L10-12: The authors write the following "If the heuristic that reproducibility is a demarcation criterion were to take hold in scientific discourse, false results might get treated as true, irreversibly altering the course of scientific progress with implications for broader society." Although I see what the authors mean and tend to agree with the point they're pushing here, this statement is a bit dramatic and ironic in my opinion. This – false results being treated as true, irreversibly altering the course of scientific progress, implications etc – is already going on, and has been the state of most fields of science for quite some time. If reproducibility were taken as a proxy for truth, the state of things would be still much better than they are now. It's not good to replace one false god with another, certainly, but to imply that that is the cost of taking reproducibility too seriously given that bullshit science already rules the literature in most fields is odd, and weakens the argument.

I think it is important to adjust this sentence to accommodate the problem that already exists (bullshit science, false results being treated as true) and that adding the idol of reproducibility to

the mix will do the opposite of what open science initiatives are attempting, which is to fix the bullshit science.

5. In Claim 2, the authors discuss the point that using the data more than once will invalidate the stats. While it's good to engage with this point, I think the illustration used with the calculation of the one-sample t-test is a bit odd, and I feel like it is a bit of a strawman argument in a sense. I find it hard to articulate exactly why this illustration does not sit well with me, but I will do my best.

I think there's a (big) difference between using the same data twice or more within a single calculation (which you also do for correlation, chi square, standard error, etc) and using it in two distinct phases (exploring/ describing vs confirming/ testing) in the scientific process. It feels to me that the authors are comparing simple arithmetic with a much more complex, and at times, philosophical issue. Is there a more compelling illustration they can come up with?

6. P11, L10/11: The authors write: "Suppose we perform some statistical activity on the data until we begin the test of interest." I do not get this. Why would someone do this? Is this meant to describe a researcher doing optional stopping, or peaking at the data before it's all collected, or...? Sorry, this part in the thought experiment totally escapes me. Perhaps it needs clarification?

7. Figure 4 – pretty graph, but will the red and green present an issue for RG colour-blind readers? Might be an idea to select a different palette for that figure if so.

8. P14, L41-43: The authors copy a section heading from a Wagenmakers paper that confirmatory conclusions need to have been preregistered. I know this paper well-ish, and think that the authors do not treat the words with the spirit in which they were written. This is not a statement Wagenmakers and colleagues make, this is a heading they chose to summarize an argument they make. Said argument in that section concludes with the following: "The only way to safeguard academics against fooling themselves, their readers, reviewers, and the general public, is to demand that confirmatory results are clearly separated from work that is exploratory."

I think it is a mischaracterization of the arguments in that paper to take the heading text without acknowledging the nuance that does exist in that section in the article. The whole thrust of this manuscript (back to the current submission now) is for people to be more nuanced. This should apply more than anything to this submission and I recommend adding some context to the quotation.

9. P16, L21/22: The authors write "It remains unclear why these labels should be preferred over more direct descriptors such as "preregistered" or "not preregistered"". I think the most accurate labels should be used, of course, but these aren't interchangeable labels. The authors seem to imply here that confirmatory/exploratory and not-preregistered/preregistered are interchangeable labels? If so, this is not accurate right? If not, perhaps some clarification is needed?

I look forward to seeing the reviewed manuscript.

Signed,
Sarahanne M. Field

Review form: Reviewer 2 (Daniele Fanelli)

Is the manuscript scientifically sound in its present form?

Yes

Are the interpretations and conclusions justified by the results?

Yes

Is the language acceptable?

Yes

Do you have any ethical concerns with this paper?

No

Have you any concerns about statistical analyses in this paper?

No

Recommendation?

Major revision is needed (please make suggestions in comments)

Comments to the Author(s)

I agree with most of the arguments advanced by the authors, but mostly fail to see how this article makes an original contribution to the literature. Evaluating this aspect is made all the more harder by the fact that the author avoid citing virtually any article other than the articles they intend to critique or very old and standard theoretical texts.

The introduction completely omits any reference to articles that have previously pointed out the problems they intend to discuss. This is unacceptable for various reasons: it suggests more originality than this article might deserve, it does not back the ideas laid out in this text with previous relevant (and maybe parallel) contributions, and it misrepresents the state of meta-science by (unwittingly, I suppose) suggesting to the reader that no scientific and methodological debate on the matter has occurred to date.

The list of publications to consider is really long, and most of the arguments made by the article seem to repeat, on a lower technical level, analyses and arguments made before.

Just on the issue of reproducibility, for example, all that the authors say has been discussed and analysed in previous publications. The list is just too long to report here, however relevant references could be found, for example, in this is a list of recent contributions, most of which should also be cited and acknowledged by the authors:

Scientific progress despite irreproducibility: A seeming paradox

Richard M. Shiffirina,¹ Katy Börnerb, and Stephen M. Stiglerc

Gelman, Andrew. 2018. "Don't Characterize Replications as Successes or Failures." *The Behavioral and Brain Sciences*. <https://doi.org/10.1017/S0140525X18000638>

Hartgerink, C. H.J., J. M. Wicherts, and M. A.L.M. Van Assen. 2017. "Too Good to Be False: Nonsignificant Results Revisited." *Collabra: Psychology* 3 (1).

<https://doi.org/10.1525/collabra.71>.

Hedges, Larry V., and Jacob M. Schauer. 2018. "Statistical Analyses for Studying Replication: Meta-Analytic Perspectives." *Psychological Methods*. <https://doi.org/10.1037/met0000189>.

Laraway, Sean, Susan Snyckerski, Sean Pradhan, and Bradley E. Huitema. 2019. "An Overview of Scientific Reproducibility: Consideration of Relevant Issues for Behavior Science/Analysis." *Perspectives on Behavior Science* 42 (1): 33–57. <https://doi.org/10.1007/s40614-019-00193-3>.

- Leek, Jeffrey T., and Leah R. Jager. 2017. "Is Most Published Research Really False?" *Annual Review of Statistics and Its Application* 4 (1): 109–22. <https://doi.org/10.1146/annurev-statistics-060116-054104>.
- Massonnet, C., D. Vile, J. Fabre, M. A. Hannah, C. Caldana, J. Lisec, G. T. S. Beemster, et al. 2010. "Probing the Reproducibility of Leaf Growth and Molecular Phenotypes: A Comparison of Three Arabidopsis Accessions Cultivated in Ten Laboratories." *PLANT PHYSIOLOGY* 152 (4): 2142–57. <https://doi.org/10.1104/pp.109.148338>.
- Mathur, M. B., & VanderWeele, T. 2017. "New Statistical Metrics for Multisite Replication Projects." <https://doi.org/10.31219/osf.io/w89s5>.
- Mathur, Maya B., and Tyler J. VanderWeele. 2019. "Challenges and Suggestions for Defining Replication 'success' when Effects May Be Heterogeneous: Comment on Hedges and Schauer (2019)." *Psychological Methods* 24 (5): 571–75. <https://doi.org/10.1037/met0000223>.
- Milcu, Alexandru, Ruben Puga-Freitas, Aaron M. Ellison, Manuel Blouin, Stefan Scheu, Grégoire T. Freschet, Laura Rose, et al. 2018. "Genotypic Variability Enhances the Reproducibility of an Ecological Study." *Nature Ecology and Evolution* 2 (2): 279–87. <https://doi.org/10.1038/s41559-017-0434-x>.
- Pauli, Francesco. 2019. "A Statistical Model to Investigate the Reproducibility Rate Based on Replication Experiments." *International Statistical Review* 87 (1): 68–79. <https://doi.org/10.1111/insr.12273>.
- Wang, Daiping, Wolfgang Forstmeier, Malika Ihle, Mehdi Khadraoui, Sofia Jerónimo, Katrin Martin, and Bart Kempenaers. 2018. "Irreproducible Text-Book 'knowledge': The Effects of Color Bands on Zebra Finch Fitness." *Evolution* 72 (4): 961–76. <https://doi.org/10.1111/evo.13459>.
- Wilson, Brent M., Christine R. Harris, and John T. Wixted. 2020. "Science Is Not a Signal Detection Problem." *Proceedings of the National Academy of Sciences of the United States of America*. <https://doi.org/10.1073/pnas.1914237117>.

Other older articles (sorry, I don't have the full reference to copy and paste for these; note how many address the exact same problems discussed by the authors):

Maxwell et al 2015 (Is Psychology Suffering From a Replication Crisis? What Does "Failure to Replicate" Really Mean?)

Patil et al 2016 ("What Should Researchers Expect When They Replicate Studies? A Statistical View of Replicability in Psychological Science")

GORROUCHURN ET AL 2007 (Non-replication of association studies: "pseudo-failures" to replicate?)

Miller 2009 (What is the probability of replicating a statistically significant effect?)

Stanley and Spence 2014 (Expectations for Replications: are yours realistic?)

Rubin 2017 (Do p Values Lose Their Meaning in Exploratory Analyses? It Depends How You Define the Familywise Error Rate)

Lieberman and Cunningham 2009 (Type I and Type II error concerns in fMRI research: re-balancing the scale, doi:10.1093/scan/nsp052 *SCAN* (2009) 4, 423^ 428)

Djulgovic and Hozo 2007 (When Should Potentially False Research Findings Be Considered Acceptable?)

Inbar 2016 ("Association between contextual dependence and replicability in psychology may be spurious")

Stroebe 2016 (Are most published social psychological findings false?)

Fielder et al 2012 (The Long Way From α -Error Control to Validity Proper: Problems With a Short-Sighted False-Positive Debate)

Although I have an inevitable self-interest in doing so, I must also mention my own work (Daniele Fanelli), which (alone or in collaboration) has repeatedly offered general analyses and critiques of the kind of problems with metascience that the authors discuss, and it has also, most recently, proposed a general framework to address some one of them, in the very same journal of this submission.

This is just a partial list, of course. And I am happy to believe that the authors might have already been familiar with most of this literature. However, no progress can be made in a field if new contributions are not compared to existing equivalent ones, and their relative merits are discussed.

Further, it is in the best interest of the authors to offer a more nuanced and contextualised analysis, to avoid the obvious third-level irony created by over-stating and over-generalizing one's own critique of meta-science.

More specific comments:

Argument 1: the discussion of reproducibility seems to effectively repeat arguments made by Goodman et al 2016, and others listed above (see also my treatment of the problem in Fanelli 2020 in the meta-arxiv). I also feel that the section portrays a straw-man version of reproducibility claims. Even Nosek & company have moved past some initial exaggeration and have offered more nuanced accounts of the problem. Note that I still agree that critical work needs to be done to improve our understanding, and I believe that there could be original elements in the characterization of the non-independence of results, but only in the sense that this formalizes concepts that other authors have expressed before. Taken as it is, the argument is not new. The argument about measurement error and the reproducibility of false results, seems to repeat conclusions of analyses that were done years ago in a more elaborate form than this article. Again, I am happy to believe that there may be elements of innovation in the argument, but these need to be made clear. As it stands, this is all *deja-vu*.

2.1 offers an argument that I think has potential value, again as a formalization of commonly expressed notions. However, it needs elaborating. At present, I find the arguments by conditioning rather too abstract to be useful. In the most concrete example, they amount to advocating for a correction of the alpha level, which is ordinary practice. I still agree that the problem of double dipping may be over-stated, but of course the concern is primarily directed at people who re-use data without correcting their statistics as necessary. So what is new here? Again, however, I notice some unfortunate omissions of key literature. The issue of Harking, for example, has been examined by Rubin in some detail, showing that there are conditions in which it is not a problem at all, precisely based on considerations of independence. Rubin is cited to refer to the problem of Harking, but his relevant theoretical arguments are not discussed. Section 3: again, very unclear what is being said here that is actually new, or even what is really at odds with what the literature being criticized. Whereas I agree that the case for exploratory analyses being problematic may have been over-stated, the literature offers deeper analyses of the problems discussed (e.g. Rubin 2017 "do P-values lose their meaning..." , and Matsunaga 2007 cited therein).

In conclusion, I wish that I could have been more positive about the article, because I agree in general with its stance and many of the arguments made in the article. However, the authors need to make a greater effort in connecting their work to the previous literature, and make a case for any specific innovation they feel they are contributing. Alternatively, there may be value in an article that just reviews and summarizes counterarguments to common claims and/or expresses an opinion. But, again, this must be made in the context of a fair and balanced review of the literature for and against the relevant claims.

Decision letter (RSOS-200805.R0)

Dear Dr Devezer,

The Editors assigned to your paper RSOS-200805 "The case for formal methodology in scientific reform" have now received comments from reviewers and would like you to revise the paper in accordance with the reviewer comments and any comments from the Editors. Please note this decision does not guarantee eventual acceptance.

Please submit your revised manuscript and required files (see below) no later than 21 days from today's (ie 07-Sep-2020) date. Note: the ScholarOne system will 'lock' if submission of the revision is attempted 21 or more days after the deadline. If you do not think you will be able to meet this deadline please contact the editorial office immediately.

Best regards,

on behalf of Dr Simone Schnall (Associate Editor) and Essi Viding (Subject Editor)
openscience@royalsociety.org

Associate Editor Comments to Author (Dr Simone Schnall):

Dear Dr. Devezer,

Please accept my apologies for the long delay in the review process, which has been due to pandemic-related issues on my part, for which I take full responsibility. Thank you for your patience.

Two highly knowledgeable reviewers who both signed their reviews (Sarahanne Field and Daniele Fanelli) provided detailed comments on your paper, which you will find below. A key

point noted by both is that you omitted to cite existing papers that already discussed related issues. Fanelli in particular questions whether there is anything new at all in your paper.

On various occasions you critique assertions made by Wagemakers and colleagues in 2012, but the field of meta-science has been evolving at a rapid pace, so it does not seem entirely fair to focus on a paper that was published quite some time ago. Both reviewers provide a long list of articles that made similar points to your own; in particular, the claim that reproducibility is the most important corner stone of science has been disputed previously, and the claim that exploratory analyses are of lower value than confirmatory analyses. For example, for the former point Stanley & Spence (Perspectives in Psychological Science) already noted in 2014 that people may not have a good understanding of how much can be inferred from replications. So, a lot has happened since the early days of methodological reform, and more nuanced discussions have been taking place for several years now.

On the one hand I see the value of continuing these discussions, but on the other hand, for your paper to have an impact you will need to show that you have something new to say. Having given this careful consideration, I will offer you an opportunity for a major revision, for which you will need to refine your argument to go beyond the existing literature.

One possibility to explore is whether the concerns raised by you, and by others before that, have in fact not been taken seriously enough. For example, is there any evidence that gatekeepers (e.g., journals or grant agencies) have implemented new requirements that might in fact be counterproductive given your argument? In other words, given years of debate, who has actually been heard when it comes to new practices adopted across the field? Perhaps you are making your argument because you have a sense that despite ongoing discussions the reform movement has pushed ahead without considering the points raised by critics?

I don't know whether this is something underlying your argument, and I certainly don't want to put words in your mouth. But in any case I would expect the revision to be a quite different paper from the one you have now, so it will be up to you to decide whether it is something you want to pursue.

Note that neither of the reviewers said much about the statistical aspects of your argument, which may, or may not, speak to the concerns about lack of novelty. If you do decide to prepare a revision I will get specialist advice on this part of the paper to ensure it gets full consideration, but this only becomes relevant once the novelty issue has been addressed.

Thanks again for considering the journal as an outlet for your work. I hope the suggestions are useful in helping you take this work to the next level.

Sincerely,
Dr. Simone Schnall

Reviewer comments to Author:

Reviewer: 1
Comments to the Author(s)

I will start by saying that this paper is a much-needed addition to the metascience and philosophy of science literatures, and would be good in a reading list for undergraduates and PhD students coming into social sciences. I enjoyed reading it.

I agree with the paper's key argument that there is not enough nuance and humility used when it comes to claims and arguments made by reformers; I also agree with the observation that this is very ironic. While I do think it's possible that these issues are marks of a field in its infancy (I'd argue that although metascience as a concept emerged as early as the debates of Neurath, Kellen and Morris in the 30's, the current wave of metascience with emphasis on reform is relatively new), it is vital that practitioners in the field stay humble and cautious in their claims and are reminded to do so.

This article is well-written and thoughtful, uses succinct phrasing and active voice, and has clearly been carefully edited. The statistics seem solid and largely nicely support the conceptual arguments. Many of the proofs in the appendix material are beyond my understanding. This said, I have some comments which I think should be considered before I recommend publication in RSOS. They largely involve language and nuance of some statements. Further, I feel that some argumentation borders on the 'straw-man' kind.

I think this paper should and will be widely read, and I think it has the potential to make impact on how people perceive initiatives and more broadly the 'reform movement'. As a result, I think its arguments should be more strongly made, and contain as much nuance and humility as the paper implores the field to use itself. This is why I am more argumentative and nitpicky in this review than I am in most reviews. My comments are intended as constructive criticism and I hope the authors can accept them in the positive spirit they are meant.

I will elaborate on the issues I perceive directly. They are presented below in chronological order. When I refer to pages and lines, I refer to those of the proof, not those of the manuscript within the proof PDF.

1. P2, L37/38: The authors express here surprise that the field has received little criticism. I think this statement is a little misleading. The field (and reforms it has birthed) certainly have received criticism on a few different levels:

For instance, Strobe and Strack (2014) were highly critical of both the replication crisis and direct replication as a means of rigorous hypothesis testing. Leonelli has been critical of using reproducibility as a yardstick for research quality. Feldman-Barrett has written about her scepticism that the benefits of preregistration/registered reports outweigh the drawbacks. Allen and Mehler discuss problems like flexibility and time when it comes to adopting open science practices. Lancaster is critical about open science principles when it comes to the large numbers of preprints flooding the internet, and Tyfield and Mirawski separately describe concerns about marketing and corporatization of science caused by the release of huge volumes of articles and data (in the form of, for instance, preprints). Makel describes his concerns with open science advances coming too fast for current institutional systems to adjust to (leading to inequality with things like hiring in academia). Bastion and Gelman both fear we are forgetting to do science in the face of having to fix its problems. Bastian in particular seems to be vocal about a few weaknesses in metascientific attempts at reform, for instance, she takes issue with the system of providing badges for things like data sharing.

There is also much criticism starting to come from STS about the lack of depth of metascientific research, among a spate of other complaints (that said, these have yet to make their way into publications from what I see, though these critiques are to be found in rather large numbers on Twitter), and this does not cover the numerous critiques of individual reform researchers' (such as Heathers and Brown) approaches either.

This is off the top of my head. There are a number of other critiques of metascience/open science reform-focused ideas and incentives that I know I am missing, and then there's the literature I

don't know about. My point is that to say that metascience has received little criticism is inaccurate in my opinion. I *would* say that its advocates can be prone to just blindly swallowing the kool-aid though (I myself have been guilty of this in the past).

I suggest to soften the statement that the field has received little criticism in favour of a more nuanced and accurate comment.

2. P4, L6/7: The authors write: "Indeed, many scientific fields have developed their own qualitative and quantitative methods such as ethnography or event study methodology to study non-reproducible phenomena." I think this statement could also do with some nuance. For one thing, ethnography was not developed in the first place to get around the reproducibility problem (not precisely what the authors said, and maybe not what they meant, but the implication is certainly there). While ethnography does have reproducibility issues that are unique to its nature, it is not accurate to say that it is always irreproducible. For instance, consider content analysis on qualitative data. If a coding system for use on a given data set (say, interview data) has been developed well enough by the original coder, it is likely that another researcher who is given the same dataset will come up with a highly comparable set of codes.

Reproducibility should be defined differently for qualitative methods than with quantitative, I would argue, but to imply that reproducibility as a goal is orthogonal to methods like ethnography (and the approaches it involves) is not accurate and will not sit well with many readers (like myself).

I suggest that that statement be adjusted to at least give a nod to the opinions of people like Ingo Rohlfling who consider reproducibility relevant for both qual and quantitative researchers.

3. P9, L50-53: The authors write "It is less clear – even to the authors of the original study – what relationship the robust empirical results have to the true mechanisms..." What is meant by this? Even to the authors of the original study – does this mean that they said something that indicated this, or are the authors inferring that the authors were unclear? Please clarify here what you mean to imply. If the authors stated this themselves, a quotation would be good, and otherwise I would like to know what the authors have inferred and why.

4. P10, L10-12: The authors write the following "If the heuristic that reproducibility is a demarcation criterion were to take hold in scientific discourse, false results might get treated as true, irreversibly altering the course of scientific progress with implications for broader society." Although I see what the authors mean and tend to agree with the point they're pushing here, this statement is a bit dramatic and ironic in my opinion. This – false results being treated as true, irreversibly altering the course of scientific progress, implications etc – is already going on, and has been the state of most fields of science for quite some time. If reproducibility were taken as a proxy for truth, the state of things would be still much better than they are now. It's not good to replace one false god with another, certainly, but to imply that that is the cost of taking reproducibility too seriously given that bullshit science already rules the literature in most fields is odd, and weakens the argument.

I think it is important to adjust this sentence to accommodate the problem that already exists (bullshit science, false results being treated as true) and that adding the idol of reproducibility to the mix will do the opposite of what open science initiatives are attempting, which is to fix the bullshit science.

5. In Claim 2, the authors discuss the point that using the data more than once will invalidate the stats. While it's good to engage with this point, I think the illustration used with the calculation of the one-sample t-test is a bit odd, and I feel like it is a bit of a strawman argument in a sense. I

find it hard to articulate exactly why this illustration does not sit well with me, but I will do my best.

I think there's a (big) difference between using the same data twice or more within a single calculation (which you also do for correlation, chi square, standard error, etc) and using it in two distinct phases (exploring/describing vs confirming/testing) in the scientific process. It feels to me that the authors are comparing simple arithmetic with a much more complex, and at times, philosophical issue. Is there a more compelling illustration they can come up with?

6. P11, L10/11: The authors write: "Suppose we perform some statistical activity on the data until we begin the test of interest." I do not get this. Why would someone do this? Is this meant to describe a researcher doing optional stopping, or peaking at the data before it's all collected, or...? Sorry, this part in the thought experiment totally escapes me. Perhaps it needs clarification?

7. Figure 4 - pretty graph, but will the red and green present an issue for RG colour-blind readers? Might be an idea to select a different palette for that figure if so.

8. P14, L41-43: The authors copy a section heading from a Wagenmakers paper that confirmatory conclusions need to have been preregistered. I know this paper well-ish, and think that the authors do not treat the words with the spirit in which they were written. This is not a statement Wagenmakers and colleagues make, this is a heading they chose to summarize an argument they make. Said argument in that section concludes with the following: "The only way to safeguard academics against fooling themselves, their readers, reviewers, and the general public, is to demand that confirmatory results are clearly separated from work that is exploratory."

I think it is a mischaracterization of the arguments in that paper to take the heading text without acknowledging the nuance that does exist in that section in the article. The whole thrust of this manuscript (back to the current submission now) is for people to be more nuanced. This should apply more than anything to this submission and I recommend adding some context to the quotation.

9. P16, L21/22: The authors write "It remains unclear why these labels should be preferred over more direct descriptors such as "preregistered" or "not preregistered"". I think the most accurate labels should be used, of course, but these aren't interchangeable labels. The authors seem to imply here that confirmatory/exploratory and not-preregistered/preregistered are interchangeable labels? If so, this is not accurate right? If not, perhaps some clarification is needed?

I look forward to seeing the reviewed manuscript.

Signed,
Sarahanne M. Field

Reviewer: 2
Comments to the Author(s)

I agree with most of the arguments advanced by the authors, but mostly fail to see how this article makes an original contribution to the literature. Evaluating this aspect is made all the more harder by the fact that the author avoid citing virtually any article other than the articles they intend to critique or very old and standard theoretical texts.

The introduction completely omits any reference to articles that have previously pointed out the problems they intend to discuss. This is unacceptable for various reasons: it suggests more

originality than this article might deserve, it does not back the ideas laid out in this text with previous relevant (and maybe parallel) contributions, and it misrepresents the state of meta-science by (unwittingly, I suppose) suggesting to the reader that no scientific and methodological debate on the matter has occurred to date.

The list of publications to consider is really long, and most of the arguments made by the article seem to repeat, on a lower technical level, analyses and arguments made before.

Just on the issue of reproducibility, for example, all that the authors say has been discussed and analysed in previous publications. The list is just too long to report here, however relevant references could be found, for example, in this is a list of recent contributions, most of which should also be cited and acknowledged by the authors:

Scientific progress despite irreproducibility: A seeming paradox

Richard M. Shiffrina,¹ Katy Börner^b, and Stephen M. Stigler^c

Gelman, Andrew. 2018. "Don't Characterize Replications as Successes or Failures." *The Behavioral and Brain Sciences*. <https://doi.org/10.1017/S0140525X18000638>

Hartgerink, C. H.J., J. M. Wicherts, and M. A.L.M. Van Assen. 2017. "Too Good to Be False: Nonsignificant Results Revisited." *Collabra: Psychology* 3 (1). <https://doi.org/10.1525/collabra.71>.

Hedges, Larry V., and Jacob M. Schauer. 2018. "Statistical Analyses for Studying Replication: Meta-Analytic Perspectives." *Psychological Methods*. <https://doi.org/10.1037/met0000189>.

Laraway, Sean, Susan Snyckerski, Sean Pradhan, and Bradley E. Huitema. 2019. "An Overview of Scientific Reproducibility: Consideration of Relevant Issues for Behavior Science/Analysis." *Perspectives on Behavior Science* 42 (1): 33–57. <https://doi.org/10.1007/s40614-019-00193-3>.

Leek, Jeffrey T., and Leah R. Jager. 2017. "Is Most Published Research Really False?" *Annual Review of Statistics and Its Application* 4 (1): 109–22. <https://doi.org/10.1146/annurev-statistics-060116-054104>.

Massonnet, C., D. Vile, J. Fabre, M. A. Hannah, C. Caldana, J. Lisec, G. T. S. Beemster, et al. 2010. "Probing the Reproducibility of Leaf Growth and Molecular Phenotypes: A Comparison of Three Arabidopsis Accessions Cultivated in Ten Laboratories." *PLANT PHYSIOLOGY* 152 (4): 2142–57. <https://doi.org/10.1104/pp.109.148338>.

Mathur, M. B., & VanderWeele, T. 2017. "New Statistical Metrics for Multisite Replication Projects." <https://doi.org/10.31219/osf.io/w89s5>.

Mathur, Maya B., and Tyler J. VanderWeele. 2019. "Challenges and Suggestions for Defining Replication 'success' when Effects May Be Heterogeneous: Comment on Hedges and Schauer (2019)." *Psychological Methods* 24 (5): 571–75. <https://doi.org/10.1037/met0000223>.

Milcu, Alexandru, Ruben Puga-Freitas, Aaron M. Ellison, Manuel Blouin, Stefan Scheu, Grégoire T. Freschet, Laura Rose, et al. 2018. "Genotypic Variability Enhances the Reproducibility of an Ecological Study." *Nature Ecology and Evolution* 2 (2): 279–87. <https://doi.org/10.1038/s41559-017-0434-x>.

Pauli, Francesco. 2019. "A Statistical Model to Investigate the Reproducibility Rate Based on Replication Experiments." *International Statistical Review* 87 (1): 68–79. <https://doi.org/10.1111/insr.12273>.

Wang, Daiping, Wolfgang Forstmeier, Malika Ihle, Mehdi Khadraoui, Sofia Jerónimo, Katrin Martin, and Bart Kempenaers. 2018. "Irreproducible Text-Book 'knowledge': The Effects of Color Bands on Zebra Finch Fitness." *Evolution* 72 (4): 961–76. <https://doi.org/10.1111/evo.13459>.

Wilson, Brent M., Christine R. Harris, and John T. Wixted. 2020. "Science Is Not a Signal Detection Problem." *Proceedings of the National Academy of Sciences of the United States of America*. <https://doi.org/10.1073/pnas.1914237117>.

Other older articles (sorry, I don't have the full reference to copy and paste for these; note how many address the exact same problems discussed by the authors):

Maxwell et al 2015 (Is Psychology Suffering From a Replication Crisis? What Does “Failure to Replicate” Really Mean?)

Patil et al 2016 (“What Should Researchers Expect When They Replicate Studies? A Statistical View of Replicability in Psychological Science”)

GORROOCHURN ET AL 2007 (Non-replication of association studies: “pseudo-failures” to replicate?)

Miller 2009 (What is the probability of replicating a statistically significant effect?)

Stanley and Spence 2014 (Expectations for Replications: are yours realistic?)

Rubin 2017 (Do p Values Lose Their Meaning in Exploratory Analyses? It Depends How You Define the Familywise Error Rate)

Lieberman and Cunningham 2009 (Type I and Type II error concerns in fMRI research: re-balancing the scale, doi:10.1093/scan/nsp052 SCAN (2009) 4, 423^ 428)

Djulgovic and Hozo 2007 (When Should Potentially False Research Findings Be Considered Acceptable?)

Inbar 2016 (“Association between contextual dependence and replicability in psychology may be spurious”)

Stroebe 2016 (Are most published social psychological findings false?)

Fielder et al 2012 (The Long Way From α -Error Control to Validity Proper: Problems With a Short-Sighted False-Positive Debate)

Although I have an inevitable self-interest in doing so, I must also mention my own work (Daniele Fanelli), which (alone or in collaboration) has repeatedly offered general analyses and critiques of the kind of problems with metascience that the authors discuss, and it has also, most recently, proposed a general framework to address some one of them, in the very same journal of this submission.

This is just a partial list, of course. And I am happy to believe that the authors might have already been familiar with most of this literature. However, no progress can be made in a field if new contributions are not compared to existing equivalent ones, and their relative merits are discussed.

Further, it is in the best interest of the authors to offer a more nuanced and contextualised analysis, to avoid the obvious third-level irony created by over-stating and over-generalizing one's own critique of meta-science.

More specific comments:

Argument 1: the discussion of reproducibility seems to effectively repeat arguments made by Goodman et al 2016, and others listed above (see also my treatment of the problem in Fanelli 2020 in the meta-arxiv). I also feel that the section portrays a straw-man version of reproducibility claims. Even Nosek & company have moved past some initial exaggeration and have offered more nuanced accounts of the problem. Note that I still agree that critical work needs to be done to improve our understanding, and I believe that there could be original elements in the characterization of the non-independence of results, but only in the sense that this formalizes concepts that other authors have expressed before. Taken as it is, the argument is not new.

The argument about measurement error and the reproducibility of false results, seems to repeat conclusions of analyses that were done years ago in a more elaborate form than this article.

Again, I am happy to believe that there may be elements of innovation in the argument, but these need to be made clear. As it stands, this is all *deja-vu*.

2.1 offers an argument that I think has potential value, again as a formalization of commonly expressed notions. However, it needs elaborating. At present, I find the arguments by conditioning rather too abstract to be useful. In the most concrete example, they amount to advocating for a correction of the alpha level, which is ordinary practice. I still agree that the problem of double dipping may be over-stated, but of course the concern is primarily directed at people who re-use data without correcting their statistics as necessary. So what is new here?

Again, however, I notice some unfortunate omissions of key literature. The issue of Harking, for

example, has been examined by Rubin in some detail, showing that there are conditions in which it is not a problem at all, precisely based on considerations of independence. Rubin is cited to refer to the problem of Harking, but his relevant theoretical arguments are not discussed. Section 3: again, very unclear what is being said here that is actually new, or even what is really at odds with what the literature being criticized. Whereas I agree that the case for exploratory analyses being problematic may have been over-stated, the literature offers deeper analyses of the problems discussed (e.g. Rubin 2017 "do P-values lose their meaning..." , and Matsunaga 2007 cited therein).

In conclusion, I wish that I could have been more positive about the article, because I agree in general with its stance and many of the arguments made in the article. However, the authors need to make a greater effort in connecting their work to the previous literature, and make a case for any specific innovation they feel they are contributing. Alternatively, there may be value in an article that just reviews and summarizes counterarguments to common claims and/or expresses an opinion. But, again, this must be made in the context of a fair and balanced review of the literature for and against the relevant claims.

===PREPARING YOUR MANUSCRIPT===

Your revised paper should include the changes requested by the referees and Editors of your manuscript. You should provide two versions of this manuscript and both versions must be provided in an editable format:
 one version identifying all the changes that have been made (for instance, in coloured highlight, in bold text, or tracked changes);
 a 'clean' version of the new manuscript that incorporates the changes made, but does not highlight them. This version will be used for typesetting if your manuscript is accepted.
 Please ensure that any equations included in the paper are editable text and not embedded images.

===PREPARING YOUR REVISION IN SCHOLARONE===

Author's Response to Decision Letter for (RSOS-200805.R0)

See Appendix A.

RSOS-200805.R1 (Revision)

Review form: Reviewer 1 (Sarahanne Field)

Is the manuscript scientifically sound in its present form?

Yes

Are the interpretations and conclusions justified by the results?

Yes

Is the language acceptable?

Yes

Do you have any ethical concerns with this paper?

No

Have you any concerns about statistical analyses in this paper?

No

Recommendation?

Accept as is

Comments to the Author(s)

Dear authors,

I have carefully read your response to reviews as well as your revised manuscript. The response to reviews alone is an impressive exercise in how we should respond to critique in the peer review process. The manuscript itself has undergone many improvements, and is even better than it already was.

I can see now, after reading your replies to my comments, that most of my concerns were down to misunderstanding some points made in the article which your revisions greatly cleared up for me. While there are still a few things we might debate, these relate to my personal 'feelings' and thoughts/philosophies. Such discussions would ideally take place over a beer (were we not in a pandemic), and are not at issue for the publication of this article for a wider readership.

All that's left for me to do now as a reviewer is to congratulate you all on producing what is, in my mind, an important, well-written and well-argued contribution to the literature.

Well done!

Warm regards,
Sarahanne M. Field

Review form: Reviewer 3

Is the manuscript scientifically sound in its present form?

Yes

Are the interpretations and conclusions justified by the results?

No

Is the language acceptable?

Yes

Do you have any ethical concerns with this paper?

No

Have you any concerns about statistical analyses in this paper?

Yes

Recommendation?

Major revision is needed (please make suggestions in comments)

Comments to the Author(s)

The case for formal methodology in scientific reform

- Statistical formality is an important idea routinely misunderstood in the meta-science literature. This is an extremely important point that needs to be published.

- The specifics of this paper seem to me to be pretty idiosyncratic and much more arguable than I'd like with a paper like this.

- I don't get what the point is of adding to this rigorous statement the idea of adding "nuance". I don't see the point here or the necessity of talking about this.

- I don't know that anyone seriously claims that results that can be replicated are true (or more likely to be true) and those which cannot be replicated are more likely to be false. That's just not the point of the literature. Instead, the problem with nonreplicable results is that they are not even false. That is, they are merely ill posed claims that have no standing as either true or false. If I say that I did A, B, and C and I get D, then who really cares about my paper at all if someone else cannot also do A, B and C and also get D. The whole point of science is that the scientific results are public, not inside the author; in this sense, the author is irrelevant and must be irrelevant. If you need to talk to the author or see the author rerun his or her analyses to get the same results, then the study conducted is not part of public scientific knowledge. This is absolutely critical, but it has nothing to do with whether the author did the right thing, only that the thing the author did is known by all and verifiably knowable.

- I don't think the points the authors make about replicability are so wrong, but I do think they are somewhat besides the point, relative to this crucial point.

- Why would model misspecification affect the replicability rate? If both the original study and the replication uses the wrong specification, they should still yield the same answer.

- In a paper about well defined, clear, formal statistical approaches, you resort to calling descriptive approaches "wonky"? Really?

- There's a whole pre-test literature that uses a small sample of data to describe and discover and then another sample or samples to draw inferences. There's no reason you can't portray description in formal ways.

- I'm afraid I have similar views about most of the points in this paper: interesting, not wrong, but idiosyncratic and a bit off point.

Review form: Reviewer 4

Is the manuscript scientifically sound in its present form?

Yes

Are the interpretations and conclusions justified by the results?

Yes

Is the language acceptable?

Yes

Do you have any ethical concerns with this paper?

No

Have you any concerns about statistical analyses in this paper?

No

Recommendation?

Accept with minor revision (please list in comments)

Comments to the Author(s)

First I apologize for the lateness of the review, having read the preprint before I thought I could complete the review quickly, but substantial changes were made to the revision. Further, I felt rather burnt out at the end of last year and it took longer than I anticipated.

Having read the original article I can see that an enormous amount of work has gone into the revision. Sorry in advance if I cause too much more work and if I misunderstand or misrepresent anything.

Decision letter (RSOS-200805.R1)

Dear Dr Devezer

On behalf of the Editors, we are pleased to inform you that your Manuscript RSOS-200805.R1 "The case for formal methodology in scientific reform" has been accepted for publication in Royal Society Open Science subject to minor revision in accordance with the referees' reports. Please find the referees' comments along with any feedback from the Editors below my signature.

We invite you to respond to the comments and revise your manuscript one final time. Please pay particular attention to reviewer 3 comments. Below the referees' comments we provide additional requirements. Final acceptance of your manuscript is dependent on these requirements being met. We provide guidance below to help you prepare your revision.

Please submit your revised manuscript and required files (see below) no later than 7 days from today's (ie 10-Feb-2021) date. Note: the ScholarOne system will 'lock' if submission of the revision

is attempted 7 or more days after the deadline. If you do not think you will be able to meet this deadline please contact the editorial office immediately.

on behalf of Dr Simone Schnall (Associate Editor) and Essi Viding (Subject Editor)
openscience@royalsociety.org

Reviewer comments to Author:
Reviewer: 3

Comments to the Author(s)
The case for formal methodology in scientific reform

- Statistical formality is an important idea routinely misunderstood in the meta-science literature. This is an extremely important point that needs to be published.

- The specifics of this paper seem to me to be pretty idiosyncratic and much more arguable than I'd like with a paper like this.

- I don't get what the point is of adding to this rigorous statement the idea of adding "nuance". I don't see the point here or the necessity of talking about this.

- I don't know that anyone seriously claims that results that can be replicated are true (or more likely to be true) and those which cannot be replicated are more likely to be false. That's just not the point of the literature. Instead, the problem with nonreplicable results is that they are not even false. That is, they are merely ill posed claims that have no standing as either true or false.

If I say that I did A, B, and C and I get D, then who really cares about my paper at all if someone else cannot also do A, B and C and also get D. The whole point of science is that the scientific results are public, not inside the author; in this sense, the author is irrelevant and must be irrelevant. If you need to talk to the author or see the author rerun his or her analyses to get the same results, then the study conducted is not part of public scientific knowledge. This is absolutely critical, but it has nothing to do with whether the author did the right thing, only that the thing the author did is known by all and verifiably knowable.

- I don't think the points the authors make about replicability are so wrong, but I do think they are somewhat besides the point, relative to this crucial point.

- Why would model misspecification affect the replicability rate? If both the original study and the replication uses the wrong specification, they should still yield the same answer.

- In a paper about well defined, clear, formal statistical approaches, you resort to calling descriptive approaches "wonky"? Really?

- There's a whole pre-test literature that uses a small sample of data to describe and discover and then another sample or samples to draw inferences. There's no reason you can't portray description in formal ways.
- I'm afraid I have similar views about most of the points in this paper: interesting, not wrong, but idiosyncratic and a bit off point.

Reviewer: 1

Comments to the Author(s)

Dear authors,

I have carefully read your response to reviews as well as your revised manuscript. The response to reviews alone is an impressive exercise in how we should respond to critique in the peer review process. The manuscript itself has undergone many improvements, and is even better than it already was.

I can see now, after reading your replies to my comments, that most of my concerns were down to misunderstanding some points made in the article which your revisions greatly cleared up for me. While there are still a few things we might debate, these relate to my personal 'feelings' and thoughts/philosophies. Such discussions would ideally take place over a beer (were we not in a pandemic), and are not at issue for the publication of this article for a wider readership.

All that's left for me to do now as a reviewer is to congratulate you all on producing what is, in my mind, an important, well-written and well-argued contribution to the literature.

Well done!

Warm regards,
Sarahanne M. Field

Reviewer: 4

Comments to the Author(s)

First I apologize for the lateness of the review, having read the preprint before I thought I could complete the review quickly, but substantial changes were made to the revision. Further, I felt rather burnt out at the end of last year and it took longer than I anticipated.

Having read the original article I can see that an enormous amount of work has gone into the revision. Sorry in advance if I cause too much more work and if I misunderstand or misrepresent anything.

===PREPARING YOUR MANUSCRIPT===

Your revised paper should include the changes requested by the referees and Editors of your manuscript. You should provide two versions of this manuscript and both versions must be provided in an editable format:
 one version identifying all the changes that have been made (for instance, in coloured highlight, in bold text, or tracked changes);
 a 'clean' version of the new manuscript that incorporates the changes made, but does not highlight them. This version will be used for typesetting.
 Please ensure that any equations included in the paper are editable text and not embedded images.

===PREPARING YOUR REVISION IN SCHOLARONE===

-- Ensure that your data access statement meets the requirements at <https://royalsociety.org/journals/authors/author-guidelines/#data>. You should ensure that you cite the dataset in your reference list. If you have deposited data etc in the Dryad repository, please only include the 'For publication' link at this stage. You should remove the 'For review' link.

-- If you have uploaded ESM files, please ensure you follow the guidance at <https://royalsociety.org/journals/authors/author-guidelines/#supplementary-material> to include a suitable title and informative caption. An example of appropriate titling and captioning may be found at https://figshare.com/articles/Table_S2_from_Is_there_a_trade-off_between_peak_performance_and_performance_breadth_across_temperatures_for_aerobic_sc_ope_in_teleost_fishes_/3843624.

Author's Response to Decision Letter for (RSOS-200805.R1)

See Appendix B.

Decision letter (RSOS-200805.R2)

Dear Dr Devezer,

It is a pleasure to accept your manuscript entitled "The case for formal methodology in scientific reform" in its current form for publication in Royal Society Open Science. The comments of the reviewer(s) who reviewed your manuscript are included at the foot of this letter.

You can expect to receive a proof of your article in the near future. Please contact the editorial office (openscience@royalsociety.org) and the production office (openscience_proofs@royalsociety.org) to let us know if you are likely to be away from e-mail

contact – if you are going to be away, please nominate a co-author (if available) to manage the proofing process, and ensure they are copied into your email to the journal.

on behalf of Dr Simone Schnall (Associate Editor) and Essi Viding (Subject Editor)
openscience@royalsociety.org

Associate Editor Comments to Author (Dr Simone Schnall):

Comments to the Author:

Dear Dr. Devezer and Colleagues,

Thank you for your thorough attention to the second round of reviews. It is my pleasure to accept this version of the paper. It exemplifies the rigor and nuance that you describe as essential for attempts to improve the process of doing science. The paper makes a much-needed and provocative contribution to the reform discourse and I'm pleased you chose our journal to share this view.

Best wishes,
Simone Schnall

Appendix A

Letter to the Editor

College of Business and Economics
University of Idaho
Moscow, ID

November 10, 2020

Dear Dr. Schnall,

Please find enclosed a revised version of our manuscript titled “The case for formal methodology in scientific reform”.

Thank you so much for the extended timeframe for getting this revision done. We have benefited greatly from this extension and completed a major revision that, we believe, has strengthened our work.

As we suggested in our earlier correspondence, we pursued the following three-step strategy for our revision:

1. We clarified our main point about statistical formalism and made it center-stage of the paper starting with the introduction and throughout the manuscript. We believe that the framework for formalism which we now outline in the introduction and illustrate throughout the paper has brought much clarity and focus, and it highlights a key missing piece in current reform discussions.
2. We situated our results better in the context of the current literature, properly acknowledging the existing disagreements and confusion. We noticed that some of the references suggested by both reviewers were very relevant to our work, yet others were tangential to irrelevant. We cited only the ones directly relevant and added more, recent references of our own choosing to provide a broader context. Overall we ended up adding over 30 new citations. They will allow curious readers to read up more on specific topics they are interested in. Also, to address a shared comment by both reviewers, we represent the current status of the debates in metascience literature more accurately in the revised manuscript.
3. We addressed most specific comments by the reviewers, adding clarifications and explanations as necessary. We disagree with a few of the comments, especially from Dr. Fanelli, and state our disagreements and give justifications for these. We believe that Dr. Fanelli has largely missed the contribution and the main goal of our paper, and his review at times is dismissive (of the contributions of our work) and hasty (in recommending references not directly related to our work and in unjustified criticisms). He has criticized the lack of novelty in our results. We believe that the revised manuscript preemptively addresses the root of his concerns. We are happy to stipulate that there exist a large number of publications in the literature that discuss various

shortcomings of their respective fields, and the revised manuscript recognizes as much. We maintain, however, that these publications fail to undergird their arguments and recommendations with the sort of formal reasoning that we argue should be the gold standard. We believe the overall negative and condemning tone of Dr. Fanelli's review does not seem warranted and we hope the revised manuscript will help assuage most of his concerns if not all.

Further, to address one of your direct comments in our email correspondence, we have included citation counts for major target articles from which the core claims we evaluate originate both in our response to reviewers and as a footnote in the paper (footnote 1). While we also acknowledge the impact these claims have had on science policy and practice in the manuscript and the response letter, we are not making these points center-stage since our (now clarified) focus is not on criticizing specific claims but on presenting a rigorous way of making them.

Thank you for an opportunity to revise our work. We hope the revised manuscript will meet your expectations. We will be happy to address any remaining concerns and incorporate further suggestions.

Please address all correspondence concerning this manuscript to me at University of Idaho and feel free to correspond with me by e-mail (bdevezer@uidaho.edu). Thank you for your consideration of our revised work.

Sincerely,

Berna Devezer

Response To Reviewers

Major comments in both reviews led us to believe that the main goal of our work was not clear. It also seems to us that we use formalism and nuance in a different sense from what is understood by the reviewers. Below, we first clarify the major goal of our work in a common responses to reviewers section, before addressing each reviewer's specific comments. We believe that clarifications in our common response and the revised manuscript incorporating these clarifications address most of the reviewers' concerns. We believe our manuscript has benefited greatly from making these clarifications and this version makes a stronger case for the value and necessity of formalism and reaches a broader audience. We thank the reviewers for their helpful comments.

Common Response to the Reviewers

1. The main goal and novelty of our work, usage of formalism and nuance.

Our main goal seems to have come across as to critique some isolated problems/ideas (the *claims* in the manuscript) in the methodological reform literature. This was not our intention. Instead we are mainly interested in the proposal and evaluation of methodological recommendations in the metascience literature—methodological understood as related to statistical methodology. The claims we address in the manuscript are just influential examples of methodological proposals from the reform literature, which lack formal rigor. We believe that the reform movement so far has been generous with its critique and recommendations on pre-reform statistical methods but skimpy with formal justifications to support their own recommendations. By formal we mean mathematical formalism of statistical methods. By nuance, we mean technical justifications done properly and with precision, that is, according to rules of probability calculus (not as nuance in discussing a point), and under well-specified assumptions and definitions.

This state of affairs –not making proper statistical statements or failing to prove them (or to at least illustrating them with examples)– is in stark contrast with method development in statistics. Statistics is a formal science and it does not sanction its methods haphazardly, based on opinions. Proposed methods must provide mathematical definitions for quantities involved, and proofs for their validity, either mathematical or simulation-based. Our main goal in the manuscript is: to convey the idea that there is a need for statistical rigor and formal (mathematical) arguments when making methodological claims, to emphasize that a field of science exists where claims about methodology can be rigorously evaluated, to show how this rigor can be achieved, and what kind of claims can or cannot be made based on what's afforded by formalism. This is also why chose to title our manuscript "The case for formal methodology in scientific reform."

The benefits of a formal approach have been laid out clearly in recent work by van Rooij and Blokpoel (2020) and the formal approach we propose is no exception:

- Formalization as a dialogical process helps us express our verbal intuitions about the phenomena of interest formally to iteratively arrive at well-specified definitions and expressions.
- Formalization reveals invisible holes, inconsistencies, and hidden assumptions in underlying our intuitions.
- Formalizations are transparent specifications that can be communicated and understood without reliance on any code, as they specify the theory independently of implementational details.
- Formalization allows us to make transparent and reproducible predictions.

The novelty of our work is in showing how formal rigor can be achieved by motivating methodological propositions from the first principles of mathematical statistics, and using statistical thinking, and probabilistic proofs or simulation-based proofs to establish the truth of these propositions. In contrast with most work in the reform literature (including a large number of references provided by Dr. Fanelli and regardless of which claims they support or oppose to) that do not advance formal statistical theory, we present mathematical definitions, propositions, their proofs, and interpretations as we formally examine each of the claims we focus on.

This is exemplified by the technical work in our five appendices. The generality of our results, and hence their novelty, seems to be lost in communication. All the results presented in appendices are novel in a statistical sense in the context of science reform. Some of these results do not assume normality (whereas a large body of metascientific work does), and others are not restricted to hypothesis testing mode of inference or p-value discussions (whereas a large body of metascientific work is). We provide clear mathematical definitions of parameters and variables under the assumed model (Appendix A), present probabilistic models and algorithms (Appendices B and E), provide proofs for methodological claims (Appendices B and C), make precise probability statements with minimal assumptions, and provide a number of simulation examples to illustrate our results. Most importantly, we explicitly document all of this work so other researchers can follow our example, build on this foundation and advance theory further. We hope the reviewers and the readers can appreciate the depth of mathematical statistical machinery used to prove these results in their presented generality: Law of Large Numbers, Central Limit Theorem, Chebychev's inequality, conditional probability and expectation, sufficiency, minimal sufficiency, and ancillarity are some of the concepts that we use to show how general results can be established starting from first principles. Most claims in the reform literature do not even attempt to distinguish an unconditional probability statement from a conditional one, let alone working with more sophisticated statistical concepts. Almost none of this fundamental machinery is used in impactful works proposing methodological reforms, but this is the established standard for statistical claims. We believe this is a critical point that is well worth making.

2. Literature citations and their relevance to our work.

Both reviewers commented that we failed to cite meta-scientific literature that has been critical of the reforms exemplified in our manuscript. We believe that the clarification we made above about the goal of our work largely resolves this issue. We are only interested in methodological claims made in the literature and consequently cite only relevant work. There were over 90 citations in our original submission, where we drew from a broad literature including metascience, statistics, social sciences, and philosophy; both recent and classic works; providing arguments counter to and in support of the specific points discussed in our manuscript.

In any case, we went through the list of recommended articles as per the points raised by the reviewers, and have incorporated some key references from metascience and statistics literatures (both from their list and from our own reading of new literature) in our write-up (we are now up to 128 references) to help contextualize our paper properly and to clarify our focus and novelty.

After reviewing the reference list suggested by the reviewers, our core issue with majority of the metascience literature has strengthened: Formal analyses for the methodological claims are indeed scarce (although not entirely nonexistent). This aspect makes many articles in the list, especially the opinion pieces that advance methodological proposals in the literature, only tangentially relevant to our work. For a methodology that is proposed or a methodological claim that is forwarded, we advocate that formal justification should be provided, not just what the opinion of the author is. On the other hand, whenever there has been formal work on the subjects we discuss, we now acknowledge them properly.

3. Counterpoints for claims investigated in our manuscript having been already made in the literature.

Naturally, the outcomes of our investigation overlap with some claims already made in the literature and oppose others. However, the goal of this manuscript is not so much as to what these specific methodological claims are, but rather –being in the domain of statistics– how these claims should be made properly and then examined formally to show whether statistical theory justifies them. The difference in our approach to arrive at methodological claims marks the original contribution of our work.

Major reasons for focusing on claims endorsed in Wagenmakers et al. (2012), Pashler and Wagenmakers (2012), Nosek et al. (2012, 2018), Nosek and Lakens (2014), OSF (2012,2015) are that these articles 1) have been very impactful in the scientific literature as indicated by high citation counts (mostly not for critical purposes), leading to wide-scale circulation and acceptance of some of the ideas proposed¹, 2) have already had meta-level impact on science policy and practice via practical reforms (e.g., reproducibility initiatives; different standards for reporting exploratory vs. confirmatory analyses), and 3) serve as useful examples to illustrate our point of need for rigor since they do not advance any formal arguments to support their statistical claims. There are many other articles making the same claims (and we cite more of them now). We agree with the reviewers that yet many other articles have advanced arguments completely contradicting these (and we now acknowledge more of them in the paper, explicitly stating that the target claims have been criticized in the literature). In the revised manuscript, we more clearly document and acknowledge ongoing debates and different arguments that have been raised in the literature in support of or criticizing the methodological claims we examine.

However, even most of the critical work lacks a formal approach and content. Since statistics is a formal discipline, the only way to settle debates about methodological claims is to provide formal arguments, mathematical or computational. This approach moves the discussion away from opinions and preferences to what can be theoretically shown. Otherwise, all sides of the arguments remain “just-so” stories, making it seem like all conclusions are equally valid. When such verbal arguments are proposed back and forth, some of these stories will be favored over others for statistically irrelevant reasons (e.g., popularity of the arguments proposed, reputation and in-group status of the authors). And sometimes it becomes

¹ Per Google Scholar as of November 7,2020, Wagenmakers et al. (2012) has 704; Pashler and Wagenmakers (2012) has 1182; Nosek et al. (2012) has 1045; Nosek et al. (2018) has 574; Nosek and Lakens (2014) has 473; OSF (2012) has 529; OSF(2015) has 4807; and Zwaan et al. (2018) has 244 citations.

impossible to introduce statistical nuance where inaccurate or overgeneralized heuristics have already grown roots.

4. Major revisions based on these three responses.

We understand that our points described so far did not come across clearly in our original submission. To more clearly articulate them early in the paper, we have revised the Introduction extensively. Particularly, we have rewritten **the section in lines 26-82** to clarify our motivation and our major goal. We have then made changes throughout the main text of the manuscript to clarify our focus and highlight our contributions. The revised parts in the text are marked by red color. A summary of these revisions is as follows:

- We highlighted that our novelty is in showing how to formally evaluate methodological claims, which we now clearly define as *statements about scientific methodology that are either based on statistical arguments or affect statistical practice*, rather than focusing on the truth value of specific claims we chose as case studies (**lines 3-8, 26-43, and 73-82**).
- We clearly outlined the steps of the formal approach we propose and discussed how the formal approach is different from the ongoing debates around certain methodological claims. (See: Formal approach to solving methodological problems) (**lines 42-82**)
- Throughout the manuscript, we referred back to the steps of the formal approach we propose and showed how we implement these steps for specific claims or explained where and why we are not able to make formal progress (**e.g., lines 148, 169, 173, 201, and so on**).
- We moved some of the results (e.g., Propositions 1.1, 1.2, 1.3, 2.1, 2.2) originally presented in the appendices into the main text and we added a non-technical interpretation and discussion of each result to highlight what a formal approach to methodological claims has to provide over a verbal one (**lines 167-187, 201-245, 288-291, and 501-524**). This maintains the emphasis on the formal approach we advocate for throughout the paper.
- Having established that the paper is about the approach we promote rather than the specific conclusions or claims, we now explicitly acknowledge the existence of counter-claims in the literature drawing from the references suggested by the reviewers (**lines 19-20, 40-41, 73-76, 119-122, 136-138, 195-197, 296, 576-580, and 780-781. See also footnotes 3 and 8.**).
- We also expanded the reference list to highlight other formal, statistically rigorous work on the topics we study. Here is the list of new citations (complete references included in the paper: Berk et al. (2013); Fanelli (2020); Fithian et al. (2015); Fried (2020); Gelman et al. (2020); Gorroochurn et al. (2007); Hacking (1983); Haig (2005; 2009; 2020); Hedges and Schauer (2019); Herfeld (2020); Jones and Love (2011); Klein et al. (2018); Laraway et al. (2019); Leek and Jaeger (2017); Loftus and Taylor (2014); Marr(1983); Matsunaga (2007); Patil, Peng, and Leek (2016); Paul (2020); Pauli (2019); Peterson and Panofsky (2020); van Rooij and Baggio (2020b); van Rooij and Blokpoel (2020); Rubin (2017a); Shamsudheen (2020); Srivastava (2018); Strack and Stroebe (2018); Stroebe and Strack (2014); Tauber et al. (2017); Williams and Mulder (2020); Zwaan (2018).

Please find our response to more specific reviewer comments below. We will refer to the points we made above as necessary.

Reviewer: 1 Comments to the Author(s)

Comment: I will start by saying that this paper is a much-needed addition to the metascience and philosophy of science literatures, and would be good in a reading list for undergraduates and PhD students coming into social sciences. I enjoyed reading it.

I agree with the paper's key argument that there is not enough nuance and humility used when it comes to claims and arguments made by reformers; I also agree with the observation that this is very ironic. While I do think it's possible that these issues are marks of a field in its infancy (I'd argue that although metascience as a concept emerged as early as the debates of Neurath, Kellen and Morris in the 30's, the current wave of metascience with emphasis on reform is relatively new), it is vital that practitioners in the field stay humble and cautious in their claims and are reminded to do so.

This article is well-written and thoughtful, uses succinct phrasing and active voice, and has clearly been carefully edited. The statistics seem solid and largely nicely support the conceptual arguments. Many of the proofs in the appendix material are beyond my understanding. This said, I have some comments which I think should be considered before I recommend publication in RSOS. They largely involve language and nuance of some statements. Further, I feel that some argumentation borders on the 'straw-man' kind.

I think this paper should and will be widely read, and I think it has the potential to make impact on how people perceive initiatives and more broadly the 'reform movement'. As a result, I think its arguments should be more strongly made, and contain as much nuance and humility as the paper implores the field to use itself. This is why I am more argumentative and nitpicky in this review than I am in most reviews. My comments are intended as constructive criticism and I hope the authors can accept them in the positive spirit they are meant.

I will elaborate on the issues I perceive directly. They are presented below in chronological order. When I refer to pages and lines, I refer to those of the proof, not those of the manuscript within the proof PDF.

Response: Thank you. We appreciate the acknowledgement of the strengths of our work and will do our best to address the critical comments. Specifically, given the critical position of the paper, we had tried our best to stay away from straw man arguments and focus on factual statements advanced in the literature. We appreciate the chance to address remaining concerns in this regard.

Comment: 1. P2, L37/38: The authors express here surprise that the field has received little criticism. I think this statement is a little misleading. The field (and reforms it has birthed) certainly have received criticism on a few different levels:

For instance, Strobe and Strack (2014) were highly critical of both the replication crisis and direct replication as a means of rigorous hypothesis testing. Leonelli has been critical of using reproducibility as a yardstick for research quality. Feldman-Barrett has written about her scepticism that the benefits of preregistration/registered reports outweigh the drawbacks. Allen and Mehler discuss problems like flexibility and time when it comes to adopting open science practices. Lancaster is critical about open science principles when it comes to the large numbers of preprints flooding the internet, and Tyfield and Mirawski separately describe concerns about marketing and corporatization of science caused by the release of huge volumes of articles and data (in the form of, for instance, preprints). Makel describes his concerns with open science advances coming too fast for current institutional systems to adjust to (leading to inequality with things like hiring in academia). Bastion and Gelman both fear we are forgetting to do science in the face of having to fix its problems. Bastian in particular seems to be vocal about a few

weaknesses in metascientific attempts at reform, for instance, she takes issue with the system of providing badges for things like data sharing.

There is also much criticism starting to come from STS about the lack of depth of metascientific research, among a spate of other complaints (that said, these have yet to make their way into publications from what I see, though these critiques are to be found in rather large numbers on Twitter), and this does not cover the numerous critiques of individual reform researchers' (such as Heathers and Brown) approaches either.

This is off the top of my head. There are a number of other critiques of metascience/open science reform-focused ideas and incentives that I know I am missing, and then there's the literature I don't know about. My point is that to say that metascience has received little criticism is inaccurate in my opinion. I *would* say that its advocates can be prone to just blindly swallowing the kool-aid though (I myself have been guilty of this in the past).

I suggest to soften the statement that the field has received little criticism in favour of a more nuanced and accurate comment.

Response: We agree with the reviewer that our statement here was confusing. We actually meant to highlight the lack of *formal scrutiny*, as in *formal statistical scrutiny* for methodological claims, as explained in **item 1 of our common response**. We now have clarified this in the introduction of the revised manuscript (**lines 4-6**). While the specific references which the reviewer refers to are not directly relevant to the claims we present (with the exception of Gelman whose work we cite multiple times), nor do they present formal arguments, for each claim we study, we have now incorporated other articles from the metascience literature that have been critical of it (**see common response #2 and #4**). We have also cited critical pieces that are technically relevant to our work from a variety of perspectives, from STS, statistics, cognitive psychology, and other disciplines throughout the paper.

Comment: 2. P4, L6/7: The authors write: "Indeed, many scientific fields have developed their own qualitative and quantitative methods such as ethnography or event study methodology to study non-reproducible phenomena." I think this statement could also do with some nuance. For one thing, ethnography was not developed in the first place to get around the reproducibility problem (not precisely what the authors said, and maybe not what they meant, but the implication is certainly there). While ethnography does have reproducibility issues that are unique to its nature, it is not accurate to say that it is always irreproducible. For instance, consider content analysis on qualitative data. If a coding system for use on a given data set (say, interview data) has been developed well enough by the original coder, it is likely that another researcher who is given the same dataset will come up with a highly comparable set of codes.

Reproducibility should be defined differently for qualitative methods than with quantitative, I would argue, but to imply that reproducibility as a goal is orthogonal to methods like ethnography (and the approaches it involves) is not accurate and will not sit well with many readers (like myself).

I suggest that that statement be adjusted to at least give a nod to the opinions of people like Ingo Rohlfing who consider reproducibility relevant for both qual and quantitative researchers.

Response: We believe that there was a confusion as to how we define reproducibility locally in our work. We focus on the statistical reproducibility of results and not the reproducibility of methods (see "results reproducibility" versus "methods reproducibility" as defined in Goodman et al., 2016). We make this explicit in the definition of the idealized experiment in Appendix A and also addressed in footnote 2

(see also lines 135-151). We have now extended this footnote to prevent any further confusions by adding the necessary clarification. It should now be clear that the coding reproducibility in ethnographic research falls outside the scope of our claims. We do not intend to suggest that methods in qualitative research cannot be reproducible. Instead we emphasize that the focus of these approaches, by definition, is not on generating reproducible results (i.e., results reproducibility) based on probability sampling from a well-defined statistical population. This is also the point made by Leonelli (2018) and Penders et al. (2019) who reflect on qualitative methodologies as qualitative researchers themselves.

Comment: 3. P9, L50-53: The authors write “It is less clear — even to the authors of the original study — what relationship the robust empirical results have to the true mechanisms...” What is meant by this? Even to the authors of the original study – does this mean that they said something that indicated this, or are the authors inferring that the authors were unclear? Please clarify here what you mean to imply. If the authors stated this themselves, a quotation would be good, and otherwise I would like to know what the authors have inferred and why.

Response: We understand the confusion our previous phrasing created. One of the authors of the Hayes et al. (2019) study (Danielle Navarro) is also an author on the current manuscript and that was the reason for our initial choice of wording. Now we have rewritten part of this example and cite another work by Dr. Navarro and colleagues to briefly explain why robust empirical results about Bayesian cognition do not necessarily capture the true underlying mechanism (see also lines 370-376). We also cite page numbers from the original Hayes et al. (2019) paper to the section where they discuss why these results do not necessarily mean that Bayesian cognition is normatively true.

Comment: 4. P10, L10-12: The authors write the following “If the heuristic that reproducibility is a demarcation criterion were to take hold in scientific discourse, false results might get treated as true, irreversibly altering the course of scientific progress with implications for broader society.” Although I see what the authors mean and tend to agree with the point they’re pushing here, this statement is a bit dramatic and ironic in my opinion. This – false results being treated as true, irreversibly altering the course of scientific progress, implications etc – is already going on, and has been the state of most fields of science for quite some time. If reproducibility were taken as a proxy for truth, the state of things would be still much better than they are now. It’s not good to replace one false god with another, certainly, but to imply that that is the cost of taking reproducibility too seriously given that bullshit science already rules the literature in most fields is odd, and weakens the argument.

I think it is important to adjust this sentence to accommodate the problem that already exists (bullshit science, false results being treated as true) and that adding the idol of reproducibility to the mix will do the opposite of what open science initiatives are attempting, which is to fix the bullshit science.

Response: We appreciate the reviewer’s point and agree that the problem is replacing one false god with another. Hence we deleted that statement and added a new paragraph instead (lines 387-393).

Having made that change, we want to emphasize that the following point made by the reviewer needs theoretical or empirical support: “If reproducibility were taken as a proxy for truth, the state of things would be still much better than they are now.” Currently, we do not have such support and cannot assess its truth but formal work like ours is a step toward that direction. A major point of our manuscript is that as researchers, we should not accept such statements on faith, without appropriate and sufficient evidence. Otherwise all we have is our unjustified intuitions. We, the authors, all practice open science

and transparency. We do work toward improving scientific practice. But we also believe this cannot be done based on reasoning, intuitions, or slogans alone; we first need to do the work, then apply reasoning and follow the evidence; hence our five-step approach.

Comment: 5. In Claim 2, the authors discuss the point that using the data more than once will invalidate the stats. While it's good to engage with this point, I think the illustration used with the calculation of the one-sample t -test is a bit odd, and I feel like it is a bit of a strawman argument in a sense. I find it hard to articulate exactly why this illustration does not sit well with me, but I will do my best.

I think there's a (big) difference between using the same data twice or more within a single calculation (which you also do for correlation, chi square, standard error, etc) and using it in two distinct phases (exploring/describing vs confirming/testing) in the scientific process. It feels to me that the authors are comparing simple arithmetic with a much more complex, and at times, philosophical issue. Is there a more compelling illustration they can come up with?

Response: We would like to point out that the section where the t -test example is given is specifically on using data more than once (or double-dipping) and it deliberately tries to stay away from exploratory/confirmatory discussion in detail. (E.g., see the paragraph before the example: "The reform literature is not very clear on the distinction between "exploratory" and "confirmatory" inference. We will revisit these concepts in the next claim but for now, we evaluate the claim that using data multiple times invalidates statistical inference. For that, we will steer away from the exploratory-confirmatory dichotomy and focus on the validity of statistical inference specifically.")

On the other hand, the principle behind the example of the t -test applies equally well to the exploratory/confirmatory distinction. We have now added a sentence to emphasize this point in the discussion of the revised example (using single-step and multi-step analysis terminology instead of exploratory/confirmatory to not complicate matters further in this section).

As for the illustration itself: We understand that it was not clear. However, we do believe that the misunderstanding of this example lies exactly at the core of our argument. It is an example we choose to articulate and therefore we have revised and expanded the illustration (**lines 416-446**).

The main point of the t -test example is as follows: When deriving valid statistical procedures, probability rules must invariably hold for all cases of manipulations of random variables, whether it is a t -pivot, or a multi-step analysis. There is no straw man here. The argument is a provable argument based on probability calculus and not a philosophical one. Proper conditioning on the information used from the data must yield valid procedures in both cases. The perceived difference between the case of t -pivot and a more complex multistep analysis might be due to the black box treatment of statistical procedures scientists employ on a regular basis without questioning the mathematics behind statistics. The t -pivot or other commonly used quantities such as correlation coefficient, chi-square statistic, standard error are *readily derived* for consumers of statistics, who only have to use the formula which may create an illusion of simplicity regarding the underlying math (which does not involve simple arithmetic but employs algebra of random variables in all cases). In the revised version this is described clearly.

Comment: 6. P11, L10/11: The authors write: "Suppose we perform some statistical activity on the data until we begin the test of interest." I do not get this. Why would someone do this? Is this meant to describe a researcher doing optional stopping, or peaking at the data before it's all collected, or...? Sorry, this part in the thought experiment totally escapes me. Perhaps it needs clarification?

Response: Here, "some statistical activity" means any information gathered from the data after it is collected but before the test of interest is performed.

To clarify "statistical activity", we added the following: "The principle of deriving the correct distribution of statistics to obtain a valid statistical procedure also applies when we perform a variety of statistical activities on the data prior to an inferential procedure of specific interest. These activities can be of any type, including exploration of the data by graphical or tabular summaries, performing other formal procedures such as tests for assumption checks (see Shamsudheen, 2020 for a formal approach for testing model assumptions)." (lines 442-446)

Comment: 7. Figure 4 – pretty graph, but will the red and green present an issue for RG colour-blind readers? Might be an idea to select a different palette for that figure if so.

Response: Thank you for this note. We now use a colorblind-friendly palette.

Comment: 8. P14, L41-43: The authors copy a section heading from a Wagenmakers paper that confirmatory conclusions need to have been preregistered. I know this paper well-ish, and think that the authors do not treat the words with the spirit in which they were written. This is not a statement Wagenmakers and colleagues make, this is a heading they chose to summarize an argument they make. Said argument in that section concludes with the following: "The only way to safeguard academics against fooling themselves, their readers, reviewers, and the general public, is to demand that confirmatory results are clearly separated from work that is exploratory."

I think it is a mischaracterization of the arguments in that paper to take the heading text without acknowledging the nuance that does exist in that section in the article. The whole thrust of this manuscript (back to the current submission now) is for people to be more nuanced. This should apply more than anything to this submission and I recommend adding some context to the quotation.

Response: We address this comment in two parts.

1. The quote "Confirmatory conclusions require preregistration" was indeed used to summarize the argument presented by Wagenmakers et al. (2012) and we think it is in the same spirit we use it for the following reason. Many articles citing Wagenmakers et al. (2012) make similar statements to distinguish between exploratory and confirmatory research based on whether activities were planned or not. Such as: "The central purpose of pre-registration is transparency with respect to which aspects of the study were pre-planned (confirmatory) and which were not (exploratory)." (Klein et al. 2018) and "Confirmatory results follow a preregistered analysis plan and thereby ensure interpretability of the reported p-values (Wagenmakers et al., 2012). In exploratory analysis, p-values lose their meaning due to an unknown inflation of the alpha-level. That does not mean that exploratory analysis is not valuable; it is just more tentative." (Nosek and Lakens (2014). In a very recent November 2020 preprint, Paul et al. (2020) state: "The advantages of preregistering research plans are manyfold. First, preregistration allows a clear separation between confirmatory (hypothesis-driven) and exploratory (data-driven) analysis". Even Wagenmakers himself continues to make this point elsewhere: "Statistical tools such as p-values and confidence intervals are meaningful only for strictly confirmatory analyses. In turn, pre-registration is one of very few ways to check and confirm that the presented analyses were indeed confirmatory." (Wagenmakers 2016, retrieved from Psychological Science blog).

While we are unsure why the reviewer thought we misrepresented the intended meaning of the quote, we did our best to rephrase our framing as follows (lines 587-592):

“Wagenmakers et al.(2012) suggest that preregistration would allow for confirmatory conclusions by clearly separating exploratory analyses from confirmatory ones and preventing researchers from fooling themselves or their readers. According to the methodological reform, any inferential procedure that is not preplanned or preregistered should better be categorized as *postdiction* or *exploratory* analysis, and should not be used to arrive at *confirmatory* conclusions (Klein et al. 2018, Nosek and Lakens 2014).”

2. It is also important to revisit the issue of formalism (or lack thereof) that we hope we clarified in our common response. For example, the argument by Wagenmakers et al. (2012) quoted by the reviewer has no statistical foundation. There are no clearly stated models, assumptions, probabilistic arguments, or proofs to establish the veracity of claims in Wagenmakers et al. (2012). As such, assessing the validity of statements regarding why unplanned analyses cannot be used to confirm a given statistical hypothesis is impossible. And it becomes an argument made from authority that the audience has no way of validating but is expected to take at the face value. Formal assumptions, definitions, arguments minimize the need for relying on the author’s credibility and rhetorical skills, and allow the readers to evaluate the soundness of the arguments using formal methodology themselves. From our perspective, nuance that would have made a difference in this claim would have to be offered based on mathematical foundations.

Comment: 9. P16, L21/22: The authors write “It remains unclear why these labels should be preferred over more direct descriptors such as “preregistered” or “not preregistered””. I think the most accurate labels should be used, of course, but these aren’t interchangeable labels. The authors seem to imply here that confirmatory/exploratory and not-preregistered/preregistered are interchangeable labels? If so, this is not accurate right? If not, perhaps some clarification is needed?

Response: We agree that these are not interchangeable labels and it was not our intention to suggest so. We now clarify this point by taking out the sentence:

“It remains unclear why these labels should be preferred over more direct descriptors such as “preregistered” or “not preregistered””.

Comment: I look forward to seeing the reviewed manuscript.

Signed,

Sarahanne M. Field

Response: Thank you! We hope the revised manuscript will alleviate the reviewer’s concerns.

Reviewer: 2 Comments to the Author(s)

Comment: I agree with most of the arguments advanced by the authors, but mostly fail to see how this article makes an original contribution to the literature. Evaluating this aspect is made all the more harder by the fact that the author avoid citing virtually any article other than the articles they intend to critique or very old and standard theoretical texts.

Response: We believe that Dr. Fanelli captures two main themes in this first comment. These themes are:

1. Questions about the novelty of our work and our conclusions
2. Insufficient contextualization of our work in the metascience literature

To prevent repeating ourselves throughout this response letter, we address these two issues upfront and then address only the remaining issues regarding specific comments.

1. The main response to the point of novelty/originality, is in **items 1 and 3 in our common response**, in which we clarify both the focus of our manuscript and the novelty of our work. For a quick overview, below is a list of results and illustrations in the manuscript that are novel:
 - Formal statement (Appendix A) and mathematical proof of propositions 1.1, 1.2, 1.3, 2.1, 2.2 (Appendices B and C)
 - Feedback mechanism described in Figure 2 as well simulation examples in Figures 1, Figure 4, and Box 3
 - Metascience application of the example in Figure 3
 - All conditioning cases in Box 2
 - Sampling algorithm in Appendix E

We show how statistical justification can be given for all of the results we present and provide clear illustrations, following steps 1 through 4a of our formal approach, which is our main point as emphasized in common response to reviewers.

We would also like to push back against Dr. Fanelli's insistence on the point of novelty of our conclusions (which is not our focus). Not because our work here lacks novelty, as we detailed in the list above but because the search for novel conclusions has been identified as one of the reasons for replication crisis so we believe it is counterproductive to insist on this point. Moreover, making conjectures and providing support for these conjectures in the form of formal proofs are completely separate ordeals. In no way should the fact that the claims we evaluate in the literature have already been challenged by counter-claims make our work toward proper formal validation redundant or unnecessary. If anything, this kind of foundational work becomes even more important in resolving these conflicts. Case in point, very recently, in 2019, *The American Statistician* published a special issue on statistical significance and a statement on the use of p -values. Neither these concepts nor the recent discussions are novel; yet, as the problems recur, the topic remains as relevant as ever and there are always significant contributions to be made, even when they may seem incremental. Similarly, we believe that the problems we are flagging

here are very relevant to the ongoing methodological reform and there is a lot of room to advance a formal understanding of meta-scientific problems.

While we do not take impact metrics as an indication of our paper's value or novelty, that our preprint has so far been downloaded over 4,000 times, already been cited 10 times (per Google scholar), discussed in multiple blog posts, appeared in a Wikipedia entry (for preregistration) is a testament that it is positively contributing to ongoing reform conversations regardless of the novelty of our conclusions.

2. For the issue regarding contextualization, **items 2 and 4 of our common response** should largely address the Dr. Fanelli's concerns. We would further like to remind the reviewer that we originally did cite a wide range of papers from recent literature in support of our arguments as well as those we critique. E.g., Heesen (2018), Barwich (2019), Shiffrin et al.(2018), Flake and Fried (2019) for Claim 1; Rubin (2017), Navarro (2019), Gelman et al. (2012), Gelman and Loken (2013), Gervais (2020), Guest and Martin(2020), MacEachern and VanZandt(2019), Muthukrishna and Henrich (2019), Szollosi et al.(2019), van Rooij(2019), van Rooij and Baggio(2020) for Claim 2; Oberauer and Lewandowsky(2019), Szollosi and Donkin(2019), Blokpoel et al. (2018), Lee et al.(2019), Goeman et al. (2011). We appreciate that the reviewer has pointed us to a number of articles that we have inadvertently omitted and have now expanded our reference list (see common response #4 for a complete list), aiming to acknowledge relevant previous work and help contextualize our contributions to the literature.

We address the remaining points to Dr. Fanelli's specific comments below.

Comment: The introduction completely omits any reference to articles that have previously pointed out the problems they intend to discuss. This is unacceptable for various reasons: it suggests more originality than this article might deserve, it does not back the ideas laid out in this text with previous relevant (and maybe parallel) contributions, and it misrepresents the state of meta-science by (unwittingly, I suppose) suggesting to the reader that no scientific and methodological debate on the matter has occurred to date.

Response: We re-wrote the introduction to clarify and emphasize our original contribution to the meta-science literature which is about formal rigor in methodology rather than making novel claims (see the common response). We now explicitly acknowledge the ongoing debates and clarify the gap we observe in the literature.

Comment: The list of publications to consider is really long, and most of the arguments made by the article seem to repeat, on a lower technical level, analyses and arguments made before. Just on the issue of reproducibility, for example, all that the authors say has been discussed and analysed in previous publications. The list is just too long to report here, however relevant references could be found, for example, in this is a list of recent contributions, most of which should also be cited and acknowledged by the authors: [...List of references provided by Dr. Fanelli here in original review...(removed for flow)]

Other older articles (sorry, I don't have the full reference to copy and paste for these; note how many address the exact same problems discussed by the authors): [...List of references provided by Dr. Fanelli here in original review...(removed for flow)]

Although I have an inevitable self-interest in doing so, I must also mention my own work (Daniele Fanelli), which (alone or in collaboration) has repeatedly offered general analyses and critiques of the kind of problems with metascience that the authors discuss, and it has also, most recently, proposed a general framework to address some one of them, in the very same journal of this submission.

Response: We have addressed the main points in this comment in our common response. On the specifics, we firmly disagree with the reviewer's following assessment "most of the arguments made by the article seem to repeat, on a lower technical level, analyses and arguments made before". The technicality of our work is in our formal approach for obtaining statistical results and method development. Formal statistical statements on methodological issues and their proofs to support them are very rare in the reform literature including in most of the references provided by the reviewer. In contrast, all of our formal statements are original. Their proofs are firmly founded in mathematical statistics and are our own work. If the reform literature provided statistical development at the level of our work, we would not have thought it necessary to write this manuscript in the first place. Please see the last part of item 1 in our common response about the (statistical) level of our work.

Our manuscript may have come across as less technical than it is because we buried the mathematical details in the appendices and chose to lead with examples that are accessible to a broader audience. To clarify that our emphasis is on providing formal justifications for methodological claims, we now have included some of these results in the main text and explicitly discuss the advantage they provide in advancing an understanding on methodological issues (**lines 167-187, 201-245, 288-291, and 501-524**).

Comment: This is just a partial list, of course. And I am happy to believe that the authors might have already been familiar with most of this literature. However, no progress can be made in a field if new contributions are not compared to existing equivalent ones, and their relative merits are discussed. Further, it is in the best interest of the authors to offer a more nuanced and contextualised analysis, to avoid the obvious third-level irony created by over-stating and over-generalizing one's own critique of meta-science.

Response: We made an honest effort to incorporate literature suggested by both reviewers in the revision. However, this is not a review article and instead of providing a comprehensive overview of the literature, we aimed to include key references that are directly relevant to our work (**see common response #2 and #4**).

We also believe that it is important to push back against the following comment as it involves a misrepresentation of our work: "obvious third-level irony created by over-stating and over-generalizing one's own critique of meta-science". Such irony does not exist in our formal statements. We are able to and do show the validity of all our results that we present in the paper. This is precisely the point of formalization we advocate in this manuscript. Anybody interested in the veracity of our claims can examine our proofs and if they find mistakes we would be happy to discuss specific points. Theory of statistics allows us to make strict statements without needless speculation. We state exactly under which assumptions our claims hold (which most of the reform literature misses) and exactly where they do or don't generalize (which most of the reform literature misses) so there isn't any over-generalization. Further, the precision we provide allows others to work toward generalizing our claims by relaxing some of our assumptions and observing the implications. One partial exception to this precision is Claim 3 and there the challenge is embedded in the problem because there are no clear mathematical definitions for scientific exploration or for the distinction between exploratory and confirmatory research. We point out

that unlike reproducibility or conditioning, exploration is a concept difficult to formalize. Therefore, our write-up in this section is necessarily different. We only aim to demonstrate why we cannot speak of a strict exploration-confirmation dichotomy when we cannot even formally define exploration. The only strict claims we make in this section (i.e., that exploration cannot be based on wonky statistics and that there is no logical link that can be made from misuse of statistics to true scientific discoveries) is a simple statistical fact and not an over-generalization either.

Comment: More specific comments: Argument 1: the discussion of reproducibility seems to effectively repeat arguments made by Goodman et al 2016, and others listed above (see also my treatment of the problem in Fanelli 2020 in the meta-arxiv). I also feel that the section portrays a straw-man version of reproducibility claims. Even Nosek & company have moved past some initial exaggeration and have offered more nuanced accounts of the problem. Note that I still agree that critical work needs to be done to improve our understanding, and I believe that there could be original elements in the characterization of the non-independence of results, but only in the sense that this formalizes concepts that other authors have expressed before. Taken as it is, the argument is not new.

Response: The focus of our work is on the form of argumentation; that is, formalism or lack thereof rather than the arguments themselves. We dissect and formally evaluate existing claims, rather than claiming novelty in our conclusions. As the reviewer has suggested, we do formalize the arguments brought up by others. We now make this clarification in the revised manuscript and acknowledge the articles suggested by the reviewer (**see footnotes 2 and 3, and lines 119-122**).

With regards to the straw man claim, we certainly do not intend to blame individuals or misrepresent their work. That is why we take care to use multiple references and use verbatim quotes to represent some of the claims that have been published. Of course, it is true that individual authors may have made exaggerated claims and changed their minds/softened their perspective over time. We do not claim or imply otherwise. However, that point is tangential to the aims of our manuscript. We suggest that there is no need to fall back on such overgeneralized and underspecified claims to begin with, especially since we know that the impact of false claims in the literature is very difficult to undo even after the authors have changed their minds. For clarity, we now make this point in the manuscript as well. If we follow the formal approach and rigorously study toward a deeper understanding of our problems before rushing to policy implementation, our claims will be measured, our assumptions will be clearly and transparently documented, our uncertainty will be incorporated into our statements from the get-go (see also van Rooij and Blokpoel, 2020). That is the approach we endorse.

Comment: The argument about measurement error and the reproducibility of false results, seems to repeat conclusions of analyses that were done years ago in a more elaborate form than this article. Again, I am happy to believe that there may be elements of innovation in the argument, but these need to be made clear. As it stands, this is all deja-vu.

Response: While related results with regards to measurement error have been published before (such as Loken and Gelman (2017) which we cited), we do not know of any papers that made our point with regard to the relationship between measurement error and statistical reproducibility in a formal framework. Regardless, however, note that this is not an argument we make but merely an example to make our formal results easier to communicate. Our simulations are original but more importantly, they are insightful in that they serve to illustrate the implications of the underlying original work (i.e., the

propositions we prove) we advance in this paper (per step 4a our our steps of formalism). We highlight the purpose and importance of our illustrations in the revised manuscript.

Comment: 2.1 offers an argument that I think has potential value, again as a formalization of commonly expressed notions. However, it needs elaborating. At present, I find the arguments by conditioning rather too abstract to be useful. In the most concrete example, they amount to advocating for a correction of the alpha level, which is ordinary practice.

I still agree that the problem of double dipping may be over-stated, but of course the concern is primarily directed at people who re-use data without correcting their statistics as necessary. So what is new here? Again, however, I notice some unfortunate omissions of key literature. The issue of Harking, for example, has been examined by Rubin in some detail, showing that there are conditions in which it is not a problem at all, precisely based on considerations of independence. Rubin is cited to refer to the problem of Harking, but his relevant theoretical arguments are not discussed.

Response: We understand that our contribution did not come across in the paper and have now expanded this section to clearly highlight the originality and importance of our formal work (**lines 416-446, 471-475, 501-524**). As implied by the title of section 2.1 “Valid conditional inference is well-established,” we naturally do not claim to have discovered conditional inference. However, the emphasis on the importance of conditionality of inference to understand what double dipping means statistically is both new and different from the practice of multiple testing corrections. Another thing that is new here is all of Box 1. There has not been a clear, formal elaboration of when multiple inference can be OK and when conditioning is necessary for valid inference. Moreover, the metascience literature largely ignores the possibility of “double dipping” improving statistical inference altogether. We have cited Rubin’s conceptual work with regards to HARKing but must emphasize that his work, too, lacks the formalism we endorse in this paper. We included further citations to recent formal work in the statistics literature such as Berk et al. (2013), Fithian et al. (2015), Gelman et al. (2020), Loftus and Taylor(2014), and Shamsudheen and Hennig (2020). These works approach a similar problem of data-driven inference from a different angle—one of simultaneous inference, rather than conditional. In that sense, our work presents a novel approach based on known results in mathematical statistics.

To address the reviewer’s point regarding the practical usefulness of our conditional inference approach, we have now added a new paragraph (**lines 559-568**).

Comment: Section 3: again, very unclear what is being said here that is actually new, or even what is really at odds with what the literature being criticized. Whereas I agree that the case for exploratory analyses being problematic may have been over-stated, the literature offers deeper analyses of the problems discussed (e.g. Rubin 2017 “do P-values lose their meaning...” , and Matsunaga 2007 cited therein).

Response: We have incorporated the suggested references in this section (**see lines 780-781**) and more importantly, clarified what our formal evaluation contributes to the discussion (**see lines 699, 704-713, 734-737, 798-801, and footnote 11**). Specifically, we discuss why we cannot implement our 5-step formal approach to the problem of exploration and hence, why we should not make broad methodological claims about exploratory research.

Comment: In conclusion, I wish that I could have been more positive about the article, because I agree in general with its stance and many of the arguments made in the article. However, the authors need to make a greater effort in connecting their work to the previous literature, and make a case for any specific innovation they feel they are contributing. Alternatively, there may be value in an article that just reviews and summarizes counterarguments to common claims and/or expresses an opinion. But, again, this must be made in the context of a fair and balanced review of the literature for and against the relevant claims.

Response: We believe that in the revised manuscript, we have clarified our contribution and provided sufficient literature-based context to compare and contrast with our work. Our goal is not to write a review article and our work should not be evaluated as one. To highlight the need for formalism in methodological claims and illustrate a path toward achieving the standard levels of formal rigor is our primary interest. While all the statistical results presented in our appendices are novel, our focus is not on the results themselves but on the methodology we endorse.

Appendix B

Letter to the Editor

College of Business and Economics
University of Idaho
Moscow, ID

February 15, 2021

Dear Dr. Schnall and Dr. Viding,

Please find enclosed a revised version of our manuscript titled “The case for formal methodology in scientific reform”.

We have addressed all reviewer comments either by providing an explanation directly to the reviewer or by making minor changes in the manuscript as necessary. Most of Reviewer 3’s comments appear to have been misunderstandings about the paper and few revisions were needed to address them. Reviewer 4 has made several very astute, technical comments that helped us refine our statements in the paper and make them more accurate and/or measured.

We appreciate your decision to accept the manuscript and hope that this final revision satisfies the criteria for publication.

Sincerely,
Berna Devezer

Response To Reviewers

Reviewer: 3

Comment: Statistical formality is an important idea routinely misunderstood in the meta-science literature. This is an extremely important point that needs to be published.

Response: We thank the reviewer for their comments.

Comment: The specifics of this paper seem to me to be pretty idiosyncratic and much more arguable than I'd like with a paper like this.

Response: We agree. We picked the popular metascientific claims and accordingly some idiosyncratic but useful examples to make a clear case. Our main message is that one has to think carefully and formally before making any methodological claims. The specific cases only serve to illustrate this general message.

Comment: I don't get what the point is of adding to this rigorous statement the idea of adding "nuance". I don't see the point here or the necessity of talking about this.

Response: It is not entirely clear to which statement the reviewer is referring but overall the reason we separately refer to both rigor and nuance in the paper is because they refer to different aspects of the problem at hand. We use rigor to mean "doing and showing the necessary work to justify the claims being made". We use nuance, on the other hand, to mean "the claims being valid only under well specified conditions". Our paper is intended for a broad audience and we believe that making the distinction will be helpful in navigating our claims. Lacking such specification necessarily leads to overgeneralized claims that lack nuance. We now make this distinction clear in the paper (See line 33-38. Note that the line numbers we refer to correspond to the version of the manuscript with changes tracked.).

Comment: I don't know that anyone seriously claims that results that can be replicated are true (or more likely to be true) and those which cannot be replicated are more likely to be false.

Response: As we show with several citations, yes, many papers/authors in the literature have made exactly this claim. For example, see references 1, 13, 23, 25, 27, 28, 29, and 54, which partly inspired the current paper.

Comment: That's just not the point of the literature. Instead, the problem with nonreplicable results is that they are not even false. That is, they are merely ill posed claims that have no standing as either true or false. If I say that I did A, B, and C and I get D, then who really cares about my paper at all if someone else cannot also do A, B and C and also get D. The whole point of science is that the scientific results are public, not inside the author; in this sense, the author is irrelevant and must be irrelevant. If you need to talk to the author or see the author rerun his or her analyses to get the same results, then the study conducted is not part of public scientific knowledge. This is absolutely critical, but it has nothing to do with whether the author did the right thing, only that the thing the author did is known by all and verifiably knowable.

Response: We understand that the reviewer finds transparency and methods replicability to be more pressing issues in the metascience literature, which is a plausible position. Our paper, however, is largely agnostic about this issue. We believe the reviewer has mixed up our use of “results reproducibility” with “methods reproducibility,” the latter of which is about transparent sharing of materials and methods. Please see footnote 2 for clarification on what we study. Our focus in Section 2 (Claim 1) is on whether the results of given empirical studies tend to occur again in independent attempts to replicate the same study -- and what statistical claims we can or cannot make about reproducibility.

That aside, the following statement by the reviewer states an opinion but is not substantiated with evidence: “the problem with nonreplicable results is that they are not even false. That is, they are merely ill posed claims that have no standing as either true or false.” Our paper makes the case that researchers should raise the bar for making such metascientific claims without proper evidence.

Comment: I don’t think the points the authors make about replicability are so wrong, but I do think they are somewhat besides the point, relative to this crucial point.

Response: As we explain above, we believe the reviewer is mistaken about which concept we study in the paper. It is correct that our points are not about replicability of methods but about reproducibility of results. Replicability (in the sense of transparency and openness) is important, but not the topic of our paper.

Comment: Why would model misspecification affect the replicability rate? If both the original study and the replication uses the wrong specification, they should still yield the same answer.

Response: Reproducibility rate of a result is a property of the population of idealized studies and one of the elements of an idealized study is the assumed model. Let us assume the true model generating the data is M_T . If the original study and the replication studies use the same misspecified model, say, M_1 , then the true reproducibility rate of a result, say will be ϕ_1 . If the original study and the replication studies use the same misspecified model but a different one than the first misspecified model, say M_2 , then the true reproducibility rate of a result, say will be ϕ_2 . In general ϕ_1 and ϕ_2 are going to be different, unless the divergence between M_1 and M_T is the same as the divergence between M_2 and M_T . Therefore, model misspecification affects the reproducibility rate of a result.

Besides, as we mentioned in the paper “All well-established statistical procedures deliver their claims when their assumptions are satisfied.” Model misspecification is a form of assumption violation. Statistical procedures will not be able to provide any guarantees about error control or replication rate when their assumptions are violated. Each specific case of assumption violation, including each case of model misspecification, should be individually studied to understand what statistical claims can or cannot be made.

Comment: In a paper about well defined, clear, formal statistical approaches, you resort to calling descriptive approaches “wonky”? Really?

There’s a whole pre-test literature that uses a small sample of data to describe and discover and then another sample or samples to draw inferences. There’s no reason you can’t portray description in formal ways.

Response: The use of “wonky” was not our choice, but part of a direct quote (Wagenmakers et al. 2012). We now clearly state this in line 695 of the revised manuscript. They do not use the term to refer to descriptive statistics; instead what they call “wonky” is situations where p-values and error rates lose their meaning--that is, where statistical inference is misused. Our paper does not talk about descriptive statistics at all so the pretest literature remains unrelated to our focus. Otherwise, we agree with the reviewer that descriptive statistics are formally well defined and we make no claim to the contrary.

Comment: I’m afraid I have similar views about most of the points in this paper: interesting, not wrong, but idiosyncratic and a bit off point.

Response: We thank the reviewer for their comments. We agree that ours is an idiosyncratic paper that is interesting and not wrong. The reviewer seems to have a particular point in mind that our paper does not make.

Reviewer: 1

Comment:

Dear authors,

I have carefully read your response to reviews as well as your revised manuscript. The response to reviews alone is an impressive exercise in how we should respond to critique in the peer review process. The manuscript itself has undergone many improvements, and is even better than it already was.

I can see now, after reading your replies to my comments, that most of my concerns were down to misunderstanding some points made in the article which your revisions greatly cleared up for me. While there are still a few things we might debate, these relate to my personal 'feelings' and thoughts/philosophies. Such discussions would ideally take place over a beer (were we not in a pandemic), and are not at issue for the publication of this article for a wider readership.

All that's left for me to do now as a reviewer is to congratulate you all on producing what is, in my mind, an important, well-written and well-argued contribution to the literature.

Well done!

Warm regards,
Sarahanne M. Field

Response: Thank you so much for allowing us to articulate the obscure or ambiguous parts of the paper with your pointed review. And thank you for reading and appreciating the revision we have submitted.

We hope the published article will be received well by the broader scientific community.

Kind regards,
The authors

Reviewer: 4

Comments to the Author(s)

Comment: First I apologize for the lateness of the review, having read the preprint before I thought I could complete the review quickly, but substantial changes were made to the revision. Further, I felt rather burnt out at the end of last year and it took longer than I anticipated.

Having read the original article I can see that an enormous amount of work has gone into the revision. Sorry in advance if I cause too much more work and if I misunderstand or misrepresent anything.

Response: We hope the reviewer has had a better start to 2021 than how last year ended. It has been a tough year for all of us. We thank the reviewer for thoughtful comments that we were happy to address. All very relevant points that needed clarification so the paper is better for it.

Summary

Comment: I think that this is a fantastic paper. It considers a relatively deep examination of three popular claims made in the scientific reform movement and elsewhere: “(a) that reproducibility is the cornerstone of science; (b) that data must not be used twice in any analysis; and (c) that exploratory projects imply poor statistical practice.” They demonstrate their arguments, rather convincingly, with a nuanced blend of statistical arguments, clearly laid out assumptions, and simulation studies. The article itself demonstrates an impressive blend of knowledge covering the history and philosophy of science and mathematical statistics and experimental experience. I find the arguments laid out in the manuscript very convincing and aligned with my intuitions before I read the manuscript. The article is brilliant and should be an exemplar for how we argue about various meta-scientific statements while acknowledging the limitations of such statements.

Response: Thank you so much for such generous praise. We hope our work takes metascience conversations a step forward in the right direction.

Comments

Comment: I really like the amendments made to Section 2a), 2b) and appendices that formalize the reproducibility problem. It makes sense, that if a result is based on a random process where the outcome is binary, that there is a rate or probability that the outcome is 1 (indicating a reproduced result). I think this has made the paper stronger as a whole and that this framing adds subtly and nuance to the discussion surrounding reproducibility. Perhaps instead of talking about whether a result is reproducible that questions should circle around the reproducibility rate, i.e., whether a result has a high reproducibility rate or not, whether it is important to have a high rate or whether it is possible, etc, rather than if a method is reproducible or not.

Response: This is a great observation. Indeed, given potential sources of error that impact our judgment on the reproducibility of specific results, it would be a more productive approach to consider reproducibility rate and sharpen our understanding of it in specific scientific contexts.

Comment: The idea of a reproducibility rate bears some relation to the statistical concept of statistical power. A low powered statistical procedure will presumably have a low reproducibility rate depending on

the signal size and sample size. The reproducibility rate is certainly however, a broader concept and I don't think there is a type I error analogue. If power is a theoretical property, is the reproducibility result a practical property?

Response: We agree with the intuition in this comment. Here is some more description of the reproducibility rate as defined in the paper: 1) It is a parameter of population of idealized studies and therefore it should depend on at least one and possibly all the components of the idealized study. This makes it a theoretical property of the population of studies. 2) It is a broader concept than power of a statistical test because it applies to all modes of statistical inference (i.e., hypothesis testing, parameter estimation, model selection), albeit in a different manner for each mode. For hypothesis testing, the relationship between the reproducibility rate of a result, power, and Type I error rate can be seen as follows. There are two key points: The first is that, when we say a “result” in the paper, it can be a true or a false result and one would have the 1- the reproducibility rate of the other. The second is that, unlike other modes of inference (e.g., parameter estimation) a hypothesis test further conditions on what is true. For example, Type I error and power are conditioned on H_0 and H_a . This delineates four well-known possibilities: 1) If we assume that H_0 is true and the result is false, then we rejected H_0 and the reproducibility rate for this result is fixed to the user defined value alpha (Type I error rate) by design. 2) If we assume that H_0 is true and the result is true, then we failed to reject H_0 and the reproducibility rate for this result is fixed as 1-alpha by design. 3) If we assume that H_a is true and the result is true, then we rejected H_0 and the reproducibility rate for this result is the power of the test---which depend on the data X_n and method S_n components of the idealized study, as well as the model M and the true value of the parameter (e.g., the mean of the distribution generating the data under H_a). Finally, 4) If we assume that H_a is true and the result is false, then we failed to reject H_0 and the reproducibility rate for this result is Type II error rate.

Comment: Page 6, line 170. Consider also defining $R^{(i)}$, i.e., “Proposition 1.1 Let R_0 be a result and let $R^{(i)}$ be the i th attempted replication of result R_0 .” While I think the reader will know what is meant it is better to define it explicitly.

Response: Thanks, this is a very good point. We included an explicit definition. Page 6, line 175-176 (Note that the line numbers we refer to correspond to the version of the manuscript with changes tracked.) now reads: Let R_0 be a result and $R^{(i)}$ be the result in i th attempted replication of the idealized study from which R_0 is obtained.

Comment: Page 6, line 172, “ $\phi = 1$ only trivially.” I think that this statement is context specific. In some fields a $\phi=1$ might be expected, e.g., classical physics, or chemistry. I might suggest “ $\phi = 1$ only in highly specific problems” or something similar.

Note that it may require hundreds of attempts to replicate to accurately estimate ϕ (within a few percent with high confidence).

Response: We agree and made the suggested change accordingly on Page 6, line 173 (and in Appendix B on line 1140). It now reads:

Further, $\phi=1$ only in highly specific problems.

Comment: Page 8. This is a fantastic example of how a true result may not be reproducible. A further example of a procedure where the truth is no reproducible is a model selection procedure where the true model is extremely complex (p , the number of parameters is large), while the sample size, n , is small (and

smaller than p , say). Almost all model selection criteria trade off goodness of fit against model complexity. In this case models simpler than the true model will be preferred with high probability – models with potentially little proximity to the truth. This suggests the reproducibility rate will also often depend on the sample size n . Also, letting p diverge with n fixed the reproducibility rate of the true model will tend to zero! This might be interpreted as the M-closed case (in the sense of Clarke et al, 2013). Are M-closed models inherently non-reproducible? I don't think further changes are required here.

Response: Thanks for this stimulating comment. There is much to brainstorm on it. We will confine ourselves to the points we made in our paper. Indeed, the effect of the sample size on reproducibility rate is also in our definition of the idealized experiment where X_n the data and $S(X_n)$ the method evaluated at the data of sample size n are components. Another simple way to see that the reproducibility rate of a result will change with the sample size is to notice that the variance of the sampling distribution of the sample mean is σ/\sqrt{n} which is a function of the sample size and will appear in confidence intervals. For the M-closed models, we agree with the given example of number of parameters/sample size issue for the reproducibility rate. Similar examples arise in nonparametric curve fitting. However, we think that a general statement about M-closed models is difficult to make. Some methods such as AIC are statistically inconsistent. That is, they do not converge to the true model as the sample size increases. So, the reproducibility rate is highly dependent on the properties of the method as well as whether the true model is an element of the model space. We are not aware of any study in this direction.

Comment: Page 13, line 401. Please don't call the advice to avoid double dipping a statistical "law". I would like to think that most statisticians would know better! This advice comes out of the fact that the theory associated with correcting for double-dipping is either difficult (and not usually built in to statistical software) and/or is computationally difficult to correct. Some of the advanced classes where I work teach about the adjustments that need to be made to procedures that use "double dipping" (examples include articles cited at the top of page 19). For example, for iid problems with sufficiently large sample size and for specific ways double dipping is used, simply using bootstrapping procedures can often be used to "fix" the p-values, however few analysts do this because it is computationally hundreds of times more expensive. Most statisticians would say that extra care needs to be taken when using the data twice, but not that they lead to inherently untrustworthy procedures. This is the kind of statistical rule of thumb or statistical advice that is given to 1st year undergraduate students because the mathematics is difficult and/or that the concepts are subtle to introduce to a student's first experience of statistics. Double dipping would also lead to excluding data cleaning, calculating summary statistics, visualizing raw data, performing data quality and assumption checks, all of which is silly not to do (as is argued in the paper).

Response: We agree and find the rule of thumb comment useful and to the point. Indeed, it is our feeling that this is how the recommendation about double dipping should be stated and that sophisticated methods to handle multiple use of data exist under certain conditions. Another rule of thumb, that $n=30$ is large enough to rely on the Central Limit Theorem is similar. Our critique is that such statements should be made with care so as not to mislead scientists applying these methods who want to make the most of their data.

Comment: Page 13. The arguments of the new material in this section don't sit well with me, but this is possibly a matter of personal taste, not that I think that anything said is wrong or necessarily needs to be changed. I think that the original version of this section was better: (1) this is not what people would refer to when referring to double dipping (e.g., model selection followed by inference, or sequential testing); (2) the number of times used depends on the implementation of the t-test statistic (e.g. calculating sigma

alone would be 3 times, calculating the mean once would use the data once and n once – 5 times in total. Perhaps it might be fairer to say that it uses the data “at least 3 times”); (3) the t-test is a special case of the likelihood ratio test with all of the inherited optimality properties and not plucked out of thin air; and (4) at a different level of abstraction statistical tests are different functions of the data, i.e., $T(X_1, \dots, X_n)$ for some non-linear function $T(\cdot)$ where in a particular sense the data is only used once.

Response: Thanks for this useful comment. We agree with the points made. Double-dipping is a challenging subject to describe even in theoretical statistics, and we tried to give a simple example that users of statistics may relate to, perhaps at the cost of oversimplification. To clarify our purpose further and to prevent potential misunderstandings of the point that the t-test example tries to convey about double-dipping, we added the following Page13, lines 430-434:

Colloquially, phrases such as double-dipping, data peeking, and using data more than once might be associated with practices such as model selection followed by inference and sequential testing. However, here, we pick a somewhat unusual example to make our point clear. Our main message is that one has to think carefully and formally what these phrases actually might mean.

Comment: Page 18, line 561 states that an Approximate Bayesian Computation (ABC) algorithm is being used. It isn't clear to me that this algorithm belongs to the class of ABC methods. Consider citing an article/book on background to ABC methods. The method stated in the paper requires the parameter space to be completely constrained to a point with no free parameters to be fit - a niche case. Most ABC methods simulate from a prior which is not the case here. What is meant by U obeys c ? Please define. The algorithm does appear to work however. Perhaps it would be simply easier to not call it an ABC algorithm.

Response: Thanks for this thoughtful comment. After some reflection, it makes sense to us that the reference to ABC or to a closely related class of algorithms (explained below) complicates the narrative. We agree with the reviewer that ABC, having a Bayesian component, requires a prior and is often used in estimating free parameters of a model. Our goal was not to refer to the Bayesian aspect of ABC, but rather to the likelihood-free aspect. Therefore, we decided to take out the reference to ABC in the paper. Our original rationale was that in all likelihood-free inference methods (including ABC), the model generating the data is sufficiently complex so that the likelihood of the data given a parameter value is unfeasible to evaluate. This makes any likelihood based inference method (including Maximum likelihood and Bayesian) challenging to implement to perform a data analysis (since both Bayesian and likelihood inference requires this evaluation). The idea behind the likelihood-free inference methods is that one can circumvent this difficulty by simulating data under the assumed model with a variety of parameter values and then comparing the simulated data to the observed data to find which parameters are more likely to have generated the observed data. Essentially, likelihood-free methods substitute the evaluation of the likelihood of the data, by simulation of hypothetical data to obtain an approximate likelihood value of the observed data under a variety of parameter values. Our algorithm has a resemblance to a likelihood-free approach because we are interested in obtaining a sampling distribution that obeys a data-dependent condition: that the value of a specified statistic obtained from the observed data is conditioned upon for the sampling distribution, but it is too cumbersome to mathematically derive what this distribution is. Thus, conditioning on the statistic obtained from the observed data to match the statistic obtained from the simulated data, has a resemblance to a likelihood-free approach.

Comment: Page 20, line 617. Preregistration is a misguided attempt to reduce overfitting or cherry picking results, both of which can be addressed in other ways – but perhaps aids in making analyses more transparent. Preregistration can lead to overfitting or underfitting the data in the replicated dataset. Overfitting can lead to superfluous variables in the model, larger standard errors, and inflated type I error, while underfitting can lead to bias in the parameter estimates... also potentially inflating type I error. Fitting only one model to your dataset doesn't tell you whether the model is any good for the replicated dataset and blindly using a single preregistered model feels like poor statistical practice to me since there is no attempt to verify model appropriateness. If in addition we can't check whether the single model is appropriate (because that's double dipping), following the precepts of the reform movement digs ourselves into a hole where we are either using an inappropriate model, or where an analysis is criticised for double dipping.

During the process of cleaning some datasets one needs to make judgement calls the type of judgements that one can't make without seeing the data. Judgement calls that can't be made in advance of seeing the data. Preregistration does not allow for this though. It almost presupposes extraordinarily nice data, which rarely happens in practice. It is clear to me that preregistration is not free of potentially negative consequences. I strongly agree with the arguments made in the paper regarding preregistration.

Response: Thank you. These are all very good points that we agree with. We particularly feel that more work is needed to statistically understand the relationship between reproducibility of results in the context of model selection. Considering that theoretical statistics is still charting the properties of model selection methods, developing a theory of reproducibility is doubly challenging. With regards to preregistration, we absolutely agree that it is important to acknowledge potential harms of strict adherence to this practice. We're pleased to hear that our caution comes through clearly in the paper.

Comment: Page 20, line 627. "performing assumption checks (if possible)." I don't like the rigid advice of checking every assumption for every method. Many assumption checks are next to pointless to check and others are impossible to verify. Take a t-test. The asymptotic involved to check for tests of normality require that sample sizes need to be relatively large, large enough for asymptotic normality to hold, making the p-values from a z-test approximately correct. However, p-values from a t-test are slightly more conservative when compared to the z-test, making normality checks a waste of time. Bigger problems arise because of the iid assumption, which usually doesn't hold exactly, and requires domain knowledge to argue whether it holds approximately or not. Consider changing to "performing assumption checks (where it is possible and sensible to do so)." If, however, the underlying null model is wildly wrong, the p-values will be meaningless.'

Response: We agree. We made the suggested change, which is now on page 20 line 638-639.

Comment: P-values/hypothesis testing procedures are never exact in practice simply because the underlying assumptions never hold exactly in practice. Much of the advice coming from the reform movement comes about because people are doubling down on p-values/hypothesis testing, as if we get p-values right science will be fixed, and as if the purpose of a scientific analysis was to calculate p-values as accurately as possible (spending enormous efforts to do so) rather than to aid in understanding an underlying phenomenon.

Response: We agree. This also reminds us of the statistical debates of 1930s between Fisher and Neyman-Pearson schools about how a hypothesis test should be conducted. Adhering to one or the other school might have certain advantages in specific situations (for example, the need to specify an alternative

hypothesis or not) but there is no a catch-all statistical solution here because the decision involves subject matter expertise.

Comment: Page 30, Appendix A. Requiring all absolute moments to exist is a very strong assumption. Most asymptotic theory uses the first four absolutely moments or less to exist. Maybe simply state that these are sufficient conditions.

Response: We agree. We added that these are sufficient conditions on line 1104. We devised these regularity conditions so that it works for the generic purpose of importing results from mathematical statistical theory to apply on the notion of the idealized study.

Comment: Page 30, A6 states the R is in the same parameter space as θ , is this right?

Response: Yes. In an earlier version of the work we formulated R to take a specifically defined result space, but since there must be a one-to-one mapping between this space and the unknowns space Θ , we specify R as taking values on Θ . For example, in a frequentist hypothesis testing context, we can define the result space as $\{\text{Reject}, \text{Fail to Reject}\}$ but since these map to a unique partition of the parameter space as defined by the null hypothesis and its complement, one can define R living on Θ . Similar arguments hold for parameter estimation and model selection. It actually would have been easier to formulate a “result” from a Bayesian perspective since Bayesian statistical inference can be naturally motivated as a special case of decision theoretic framework, where the relationship between decisions under uncertainty and the space of unknowns is easier to see.

Minor comments

Perhaps these are a matter of taste.

Comment: Page 2. Line 16. I’d use “claim” rather than “facts” here.

Response: Done.

Comment: Page 2. Line 52. I’d use “arguments” rather than “proofs” here.

Response: We specifically wanted to focus on proofs here exact (i.e., mathematical) or approximate (i.e., simulation-based). So we left this as it is.

Thank you kindly for your careful read and insightful comments.